



Phanerozoic paleoenvironmental and paleoclimatic evolution in Svalbard

Aleksandra Smyrak-Sikora[1,2], Peter Betlem[3], Victoria S. Engelschiøn[4], William J. Foster[5], Sten-Andreas Grundvåg[6], Mads E. Jelby[7], Morgan T. Jones[8,9], Grace E. Shephard[10,11], Kasia K. Śliwińska[12], Madeleine L. Vickers[8], Valentin Zuchuat[13,14], Lars Eivind Augland[9], Jan Inge Faleide[9], Jennifer M. Galloway[15], William Helland-Hansen[7], Maria A. Jensen[2], Erik P. Johannessen[16,] Maayke Koevoets[17], Denise Kulhanek[18], Gareth S. Lord[19], Tereza Mosociova[2,9], Snorre Olaussen[2], Sverre Planke[9,20], Gregory D. Price[21], Lars Stemmerik[12], Kim Senger[2]

[1] Department of Geosciences, Norwegian University of Science and Technology (NTNU), Trondheim 7031, Norway.
[2] Department of Arctic Geology, The University Centre in Svalbard (UNIS), Longyearbyen 9171, Norway.
[3] Norwegian Geotechnical Institute (NGI), Oslo NO-0806, Norway.
[4] Natural History Museum, University of Oslo, Oslo 0562, Norway.
[5] Institute for Geology, University of Hamburg, Hamburg 20146, Germany.
[6] Department of Geosciences, The Arctic University of Norway, Tromsø 9019, Norway.
[7] Department of Earth Science, University of Bergen, Bergen N-5020, Norway.
[8] Department of Ecology and Environmental Science (EMG), Umeå University, Sweden.
[9] Department of Geosciences, University of Oslo, Oslo 0315, Norway.
[10] Centre for Planetary Habitability (PHAB), University of Oslo, Oslo 0315, Norway.
[11] Research School of Earth Sciences, Australian National University, Acton, Canberra, Australia.
[12] Department of Geo-energy and Storage, The Geological Survey of Denmark and Greenland (GEUS), Copenhagen 1350, Denmark.
[13] Mineral Resources, CSIRO, Australia.
[14] Geological Institute, RWTH-Aachen University, Aachen 52062, Germany.
[15] Geological Survey of Canada (GSC)/Commission géologique du Canada, Calgary, AB, Canada.
[16] EP Skolithos, Sisikveien 36, 4022 Stavanger, Norway.
[17] Geological Survey of the Netherlands (TNO), Utrecht 3584 CB, the Netherlands.
[18] Institute of Geosciences, Kiel University, Kiel 24118, Germany.
[19] Equinor, Norway
[20] Volcanic Basin Petroleum Research AS (VBPR), Oslo
[21] School of Geography, Earth and Environmental Sciences, University of Plymouth, Devon PL4 8AA United Kingdom

*Correspondence to*: Aleksandra Smyrak-Sikora (aleksandra.a.smyrak-sikora@ntnu.no)

**Abstract.** Sedimentary rocks can provide information about the Earth paleoenvironment and are studied extensively to understand the causes and consequences of global climate changes in deep time. They facilitate long-time perspectives that



constrain climate models and provide analogues for how Earth systems may respond to, and recover from, intervals of profound environmental change, including projected anthropogenic change. The Norwegian Svalbard archipelago offers an extensive Phanerozoic stratigraphic record that reflects the geological evolution of the northern flanks of continental assemblages that include Laurentia, Eurasia, and Pangea. Svalbard's Phanerozoic sedimentary and paleoclimatic archive is controlled largely by Svalbard's overall northward plate-tectonic motion from equatorial to high latitudes, but also by regional to local formation of topography and basins in response to long-term plate reorganization, as well as the near- and far-field influence of large igneous province activity on the tectono-stratigraphic and paleoclimatic development. Various sedimentary and geochemical proxies, such as bentonite beds and carbon isotope excursions associated with the far-reaching environmental effects of the Siberian Traps, the High Arctic Large Igneous Province, and the North Atlantic Igneous Province are present in Svalbard's near complete geological record. As such, Svalbard is unique in that these and numerous other global environmental perturbations are recorded within a relatively restricted study area, with most of the key events preserved and recorded in easily accessible drill cores and well-exposed outcrop sections. Here we review deep-time paleoenvironmental and paleoclimate research in Svalbard by summarizing 148 peer-reviewed scientific articles. The review builds on the well-established tectono-stratigraphic and lithostratigraphic framework, as well as state-of-the art environmental reconstructions to provide insights into the Earth system during the Phanerozoic northward drift of Svalbard and the many major biotic crises in the geological past. We focus on globally significant events including i) the expansion of Devonian vegetation, ii) the Carboniferous-Permian response to icehouse conditions during the Late Paleozoic Ice Age (LPIA), iii) the End-Permian Mass Extinction (EPME) and the subsequent Triassic recovery, the iv) Carnian Pluvial Episode, v) Jurassic-Early Cretaceous climate perturbations including the Volgian Isotopic Carbon Excursion (VOICE) and the Aptian Ocean Anoxic Event 1a (OAE1a), and vi) the Paleocene-Eocene Thermal Maximum (PETM). We present and synthesize existing core and outcrop data that preserve biological and geochemical proxies and climate sensitive sedimentary facies that reflect environmental change in terrestrial and marine settings. Finally, we discuss the Phanerozoic climate recorded in Svalbard and its role in providing high latitude calibration points for several global paleoclimate events to provide a higher latitude perspective to complement the dominance of mid- and low-latitude locations and datasets in the literature.

# 1 Introduction

The recent Intergovernmental Panel on Climate Change report (IPCC; Pörtner et al., 2022) and the Intergovernmental Science-Policy Platform on Biodiversity and Ecosystem Services (Watson et al., 2019) highlight the challenges humanity is facing due to ongoing and projected climate change. Human-environment interactions have accelerated dramatically through the industrial revolution, and the human species is now considered to be a dominant geological force on the planet (Stewart, 2016). The rate of change the planet is experiencing is unprecedented for at least the last 66 Ma (Zeebe et al., 2016), which could lead to a biodiversity crisis of similar amplitude to those experienced during previous hyperthermals throughout the Phanerozoic. However, it is still uncertain when ecological tipping points occur, at which an ecosystem can no longer cope with



environmental change, will be reached. It is thus critical to understand and define the trajectories and pace of ecological change that is the result of a major climate perturbation.

Studying episodes of past climate change, as recorded in the geological record, can provide insights into the response of Earth system processes to climate perturbations. Deep-time paleoclimatology, here considered to be pre-Quaternary (i.e., older than 2.58 Ma; all absolute ages refer to the International Stratigraphic Chart 2023/09), in its broadest sense refers to deciphering how and why the climate changed in the past, and the consequences of those changes to life on Earth. Throughout the pre-Quaternary Phanerozoic (ca. 538.8 Ma to 2.58 Ma), mass extinctions or smaller scale biodiversity crises occurred repeatedly,

often in response to rapid climate change (Fig.1.; e.g., Bond and Grasby 2017; Kemp et al., 2015). Understanding past climate trends and episodes of major environmental perturbations will better constrain our understanding of the causes and consequences of future change (e.g., Soreghan 2004; Jansen et al., 2007). Many proxies exist to constrain past paleoenvironmental and climatic settings, grouped into biological, chemical and geophysical including climate sensitive sedimentary facies. Different proxies are suitable for quantifying various paleoclimatic signals, including volcanic activity,

atmospheric gas concentration, land/sea temperature, seasonality, precipitation, and ocean oxygenation.

Significant perturbations to global climate are often related to relatively short-lived catastrophic events (from minutes to 1-2 Myr), including meteorite impacts, volcanic and kimberlite eruptions, the emplacement of large igneous provinces (LIPs; Bryan and Ernst, 2008; Green et al., 2022) and the biogeochemical cascades that these events cause. These events disturb Earth's biosphere by causing extreme changes in temperature, precipitation, wildfire frequency, sea level, oxygen level, and

saturation states of biologically important elements (e.g., aragonite saturation state of the oceans; Hönisch et al., 2012), release of deleterious substances such as mercury (e.g., Grasby et al., 2011; Sanei et al., 2012;   Galloway and Lindström, 2023), and trophic knock-on effects, affecting both the atmosphere and marine realms (Jenkyns, 2010). LIP volcanism in particular directly perturbs the climate system via release of gasses directly to the atmosphere, including $SO_2$, $CH_4$ and $CO_2$, as well as HCl, halocarbons, and Hg. LIP emplacements are in turn often associated with global environmental changes, including mass

extinctions and smaller scale biotic crises such as oceanic anoxic events (OAEs; Grasby et al. 2011; Bond et al, 2014; Ernst and Youbi, 2017; Jones et al., 2016; Svensen et al., 2019). LIPs, by definition (Bryan and Ernst, 2008), involve significant igneous volumes if igneous material (>0.1 million km$^3$) emplaced or erupted over large areas (>0.1 million km$^2$) in an intraplate setting and within a short duration (1-5 Myr pulse for >75% of the volume, 50 Myr maximum lifespan), although some igneous activity broadly accepted as LIPs have protracted histories (e.g., the High Arctic Large Igneous Province; HALIP; Dockman

et al., 2018; Heyn et al., 2024).



**Figure 1: The stratigraphic record of Svalbard in a global deep-time climate context. (a) global data coverage including mean tropical sea surface temperatures per Myr marked as a green curve, with shaded area 95% confidence intervals, based on oxygen isotopes from phosphatic and carbonate fossils from Scotese et al., (2021), after Song et al., (2019); The scale of $\delta^{18}O_{Phos}$, black axis on the left represents phosphatic fossils (phosphatic brachiopod, conodont, and fish). The scale of $\delta^{18}O_{Carb}$ black axis on the right, represents carbonate fossils, (belemnite, bivalve, brachiopod, planktonic foraminifera). The Ice extent from pole after Macdonald (2020); major LIPs recorded in Svalbard: ST-Siberian Traps, HALIP- High Arctic Large Igneous Province, NAIP- North Atlantic Igneous Province, (b) Phanerozoic time scale, (c) Sea level curve (after Dallmann (ed) (2015), based on work of the International Commission on Stratigraphy,) (d) overview of the geological record of Svalbard (modified from Dallmann (ed) (2015)).**





**Figure 2: Geological map of Svalbard and a regional cross-section illustrating the major structural elements of central Spitsbergen**
**and spatial coverage of industrial and research boreholes of relevance to deep time paleoclimatic studies. Upper left: International Bathymetric Chart of the Arctic Ocean (IBCAO; Jakobsson et al. (2012)). Geological Map of Svalbard and cross-section (A-A'; bottom) from Dallmann (ed) (2015), Norwegian Polar Institute. Location of boreholes from Senger et al., (2019). NH- Nordfjorden High, SHH- Sørkap Hornsund High, ALB- Andrée Land Basin, BFZ- Billefjorden Fault Zone, WSFTB- West Spitsbergen Fold and Thrust Belt.**

Svalbard is a Norwegian archipelago comprising all islands between 74°–81°N and 15°–35°E, including the largest island of

Spitsbergen (Fig. 2). Extensive amounts of geological data have been acquired from outcrops and drill cores across Svalbard.

This data collection was largely triggered by the importance of Svalbard as an equivalent to the sedimentary successions in the

offshore areas of the Norwegian Barents Sea (Steel and Worsley 1984; Olaussen et al., 2025 and references therein). The

stratigraphic succession in Svalbard is almost complete (Fig. 1) with distinctive shifts in deposition over time, indicating a

genetic link between deposition and paleolatitudinal position (Fig. 3), as initially identified by Steel and Worsley (1984).

Svalbard's Phanerozoic paleoenvironmental evolution is largely controlled by two main factors: 1) the northward tectonic





motion of Svalbard from equatorial to polar latitudes (Fig. 3), and 2) the influence of proximal and distant LIPs (Fig. 1). Svalbard's lower Paleozoic geological record expresses its affinity to Laurentia. Broadly speaking, Svalbard was part of the Laurasian or Eurasian plates during its post-Devonian history. The overall northward motion from an equatorial position in
the Devonian and early Carboniferous to its present polar location during the Late Cretaceous (Fig. 3) was a response to absolute plate movement and relates to the break-up of the supercontinent Pangea near the end of the Jurassic. The LIPs influencing the depositional record in Svalbard (Fig.3) include the Siberian Traps, implicated as the causal factor of the end-Permian mass extinction (EPME; ca. 252 Ma; Reichow et al., 2009; Burgess et al., 2017), the High Arctic Large Igneous Province (HALIP) that influenced paleoenvironments of Svalbard and much of the circum-Arctic area during the Early
Cretaceous (Midtkandal et al., 2016; Vickers et al., 2019; Galloway et al., 2022; 2023; Galloway and Lindström, 2023), and the North Atlantic Igneous Province (NAIP), associated with the opening of the North Atlantic and the warming of the Paleocene-Eocene Thermal Maximum (PETM; started ca. 56 Ma; Charles et al., 2011). Many of these regional to global-scale events can be directly studied on the exceptional sedimentary rock exposures of Svalbard, notably the vertically tilted Festningen section in western Spitsbergen (e.g., Grasby et al., 2015; Vickers et al., 2019; 2023; Senger et al., 2022). These and
other events have also been recorded in drill cores collected for coal exploration, research purposes, and $CO_2$ storage in central Spitsbergen (e.g., Dypvik et al., 2011; Midtkandal et al., 2016; Grundvåg et al., 2017; Olaussen et al., 2019; Senger et al., 2019; Zuchuat et al., 2020; Jelby et al., 2025).

The Arctic has warmed twice to nearly four times faster than the rest of the globe in recent decades (Rantanen et al., 2022). This phenomenon, known as polar amplification, is largely due to oceanographic and atmospheric feedback processes (e.g.,
Screen and Simmonds, 2010). Polar amplification is apparent from the geological past as extreme climates that are inexplicable in current model gradients (Evans et al., 2018; Price et al., 2020). Due to the current position of Svalbard and its paleogeographic history, the nearly continuous Phanerozoic record on Svalbard provides an ideal site to study polar amplification; and prior to the Cretaceous, to elucidate the effects of the break-up of supercontinents and subsequent northward plate tectonic movement (Fig.3).

Currently, there are no compilations of paleoclimate and paleoenvironmental research from Svalbard addressing the pre-Quaternary Phanerozoic depositional record that is available. As a first, this study synthesizes the stratigraphic record covered by drill core material and high-quality outcrops with reliable geochronological constraints. This contribution also provides an overview of the range of proxies used to reconstruct Svalbard's paleoenvironmental evolution. Owing to its own unique tectonostratigraphic evolution, we do not include the successions exposed on Bjørnøya (the southernmost island of the
Svalbard archipelago) in this review (see papers by Worsley et al., 2001; Grundvåg et al., 2023; Janocha et al., 2024, for details on this succession). To provide a framework for the review of Svalbard's deep-time climate history, the five-step paleoclimate classification of Zhang et al., (2016) was used which recognizes five major climates. These are: A- Tropical, B-Dry, C-Temperate, D-Continental and E-Polar. This division enables the deep-time climate classification estimated primarily from climatically sensitive deposits and paleontological evidence supplemented by geochemical proxies including isotope data. We
systematically compile published literature (n=148) with relevance for deep-time paleoclimate and paleoenvironments in



Svalbard. The synthesized data proxies (e.g., TOC, $\delta^{18}$O, and $\delta^{13}$C) reflect environmental changes in terrestrial and marine ecosystems that are presented in the broader context of pan-hemispherical and global climate events. The overall evolution of the paleoclimate preserved in the geological record of Svalbard is discussed and compared with the paleo-position of Svalbard and global average temperature trends.


**Figure 3: Global paleogeography (PaleoDEMs) from Scotese and Wright (2018), redrawn via export from GPlates (v2.5 Müller et al. 2018) for selected timesteps. Terrane boundaries in orange and political boundaries and present-day coastlines in black. Yellow dashed ring in the global Mollweide projections identifies the approximate location of Svalbard at the selected time periods with zoom in shown to the side of global maps. As this is a global model there may be discrepancies from regional Svalbard**
**paleogeography whereby the reader is directed to Dallmann (ed) (2015) for regional resolution.**





## 2. Tectonic and stratigraphic development

The sedimentary successions preserved in Svalbard record a changing climate controlled to a large degree by the paleo-latitude of Svalbard along with global climatic transitions (e.g., Steel and Worsley, 1984). Since the start of the Paleozoic, Svalbard
gradually drifted from near-equatorial latitudes to its present position at 74–81°N (Fig. 3; Scotese et al., 1979; Torsvik et al., 2002; Torsvik and Cocks, 2019).

**Figure 4: Stratigraphic column and climate summary of Svalbard highlighting the nearly continuous sedimentary record from the**
**Devonian to the Neogene. The stratigraphic coverage of research boreholes is indicated. Post-Devonian lithostratigraphic column after Olaussen et al. (2025). The Climate zones from Zhang et al., (2016). Illustration of data coverage is based on Gruszczyński et al., (1989); Mii et al., (1997); Wignall et al., (1998; 2016); Galfetti et al., (2007); Cui et al., (2011); Buggisch et al., (2012); Mueller**



et al, (2014); Bond et al., (2015); Grasby et al. (2015a); Koevoets et al., (2016); Midtkandal et al., (2016); Vickers et al., (2016; 2019a; 2023), Hammer et al., (2019); Doerner et al., (2020); Jelby et al., (2020); Zuchuat et al. (2020); Wesenlund et al., (2021); Blattmann
et al., (2024); Leu et al., (2024).

## 2.1. Paleozoic (Cambrian to Permian ca. 538-252 Ma)

The early Paleozoic succession of the Oslobreen Group (Fig. 4) preserved in northeastern Svalbard (Fig. 2) was deposited on the northern margin of Laurentia in a post-rift to a passive margin setting (Fig. 3; Smelror et al., 2024). Later, the Caledonian
Orogeny started with the closure of the Iapetus Ocean and subsequent collision of Laurentia and Baltica in the Early Ordovician-Early Devonian (ca. 485–410 Ma; Barentsian Caledonides in Gee et al., 2006; 2008; Gee and Tebenkov, 2004; Harland et al., 1974). Caledonian deformation, metamorphism, and late crustal magmatism impacted mainly western and central-northern Svalbard, leaving the north-eastern parts of the archipelago practically undeformed (Johansson et al.2004; 2005; Smelror et al., 2024).

The syn- to post-Caledonian, Upper Silurian (Pridoli?) to Upper Devonian (Fransian) Old Red Sandstone (ORS) succession (ca. 423–372 Ma) represented by the Siktefjellet, Red Bay and Andrée Land groups (Fig. 4; Friend et al., 1997; Blomeier et al., 2003a; 2003b) is preserved in post-orogen collapse basins located in the central part of northern Spitsbergen (Fig. 2; Piepjohn et al., 2000; Blomeier et al., 2003; Braathen et al., 2018; Smelror et al., 2024). Significant deformation of the ORS succession subsequently took place during the Late Devonian compressional Svalbardian Event that is correlated with
Ellesmerian Orogeny in Arctic Canada (McCann, 2000; Piepjohn, 2000; Bergh et al., 2011; Piepjohn and Dallmann 2014; Piepjohn and von Gosen, 2018; Beranek et al., 2020).

The Tournaisian (?)- Viséan (ca. 359-330 Ma) continental coal-bearing deposits of the Billefjorden Group are widespread across Spitsbergen (Fig. 2) and unconformably overlie the deformed Devonian and pre-Caledonian succession (Piepjohn et al., 2000). The thickness of the Billefjorden Group reaches up to 250 m in central Spitsbergen (Cutbill and Challinor, 1965;
Gjelberg and Steel, 1981) and 40-100 m in northeastern Spitsbergen (Lauritzen and Worsley, 1975; Scheibner et al., 2012). The thickness along the west coast of Spitsbergen is uncertain due to younger deformation and repetition of the succession. The Billefjorden Group is unconformably overlain by the Serpukhovian (upper Mississipian) to Artinskian (Cisuralian) mixed siliciclastic-carbonate-evaporite deposits of the Gipsdalen Group (ca. 331–284 Ma). The lower Gipsdalen Group consists of syn-tectonic units filling up rift basins, the Billefjorden, Lomfjorden, St Jonsfjorden, and Inner Hornsund troughs, developed
along north-south striking long-lived lineaments formed in response to regional-scale extension (Fig. 4; Holliday and Cutbill, 1972; Gjelberg and Steel, 1981; Johannessen and Steel, 1992; Faleide et al., 2008; Braathen et al., 2012). The thickest (> 1.5 km) and best-preserved basin fill occurs in the Billefjorden Trough while corresponding succession is missing on the structural highs (Fig. 2; Cutbill and Challinor 1965; Johannessen and Steel, 1992; Braathen et al., 2012; Smyrak-Sikora et al., 2018; 2021). The syn-rift units of the lower Gipsdalen Group were subsequently overlain by an up to 500 m thick warm-water
carbonate-platform deposits of the upper Gipsdalen Group (Hüneke et al., 2001; Blomeier et al., 2011; Ahlborn and Stemmerik



2015; Sorento et al., 2020). The Gipsdalen Group is overlain by upper Artinskian (Cisuralian) to Changhsingian (Lopingian) cool-water carbonate and spiculitic platform sediments of the Tempelfjorden Group (ca. 284 (?)–252 Ma). The thickness of Tempelfjorden Group varies across Spitsbergen from 6 m to 460 m (Blomeier et al., 2013; Uchman et al., 2016; Matysik et al., 2018), including complete absence of Permian deposits on the Sørkapp–Hornsund High, where Carboniferous fluvial

deposits are unconformably overlain by Lower Triassic continental conglomerates (Zuchuat, 2014). These thickness variations indicate ongoing uplift of the Nordfjorden High and the Sørkapp–Hornsund High that can be linked with the late Permian rift event along the western Barents shelf margin (Faleide et al., 2008; Olaussen et al., 2025).

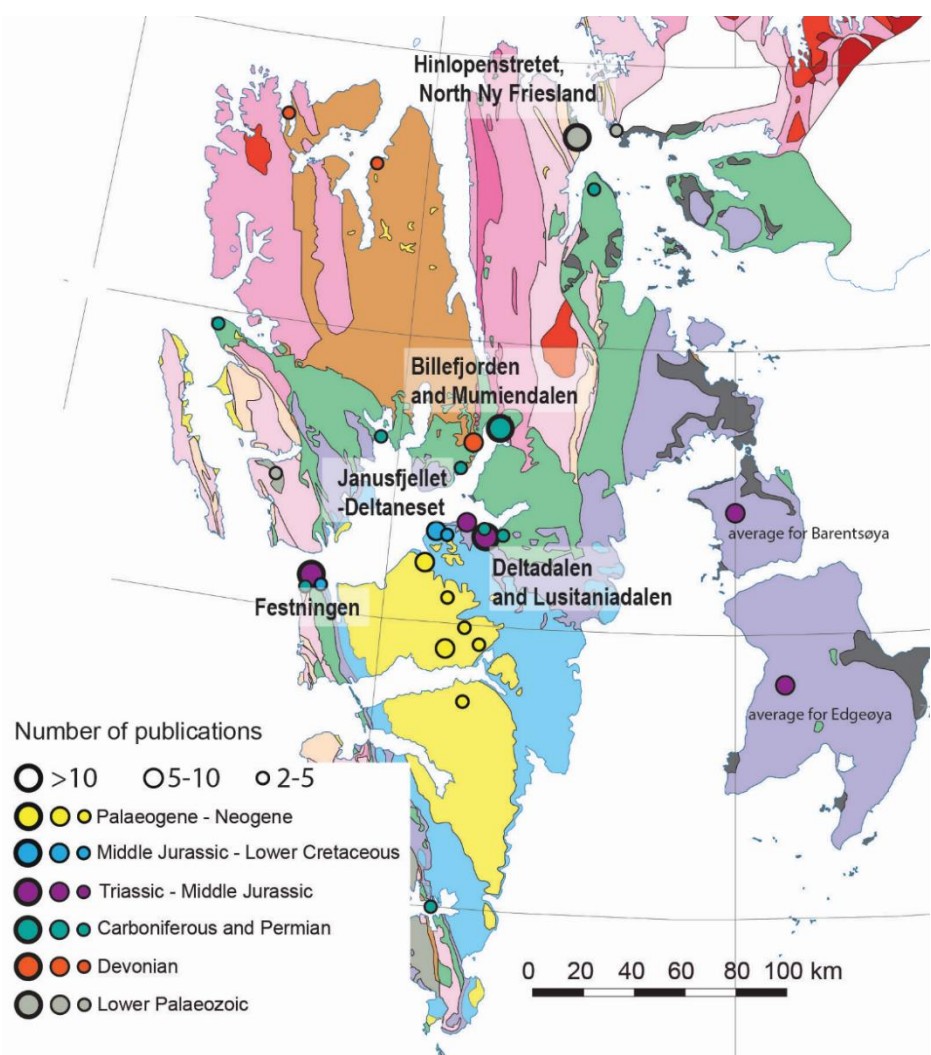


**Figure 5: Location of data presented in the reviewed articles summed up in Table A1 (Supplementary table) highlighting the most studied sections in Svalbard. The size of a circle corresponds to the number of publications addressing deep-time paleoclimate proxies. For legend to the geological map see Fig. 2.**



## 2.2. Mesozoic

In the Early to Middle Triassic (ca. 252–237 Ma), Svalbard was part of a shallow shelf that experienced significant subsidence and which was filled with up to 700 m of sediments sourced from west and east (the Sassendalen Group; Fig. 4; Mørk et al., 1982; 1999; Wesenlund et al., 2022; Bjerger et al., 2023). By the end of the Middle Triassic (ca. 237 Ma), deltaic systems sourced in the Uralides and the Fennoscandian Shield reached and probably traversed Svalbard (Riis et al., 2008; Glørstad-Clark et al., 2010; Høy and Lundschien, 2011; Anell et al., 2013; Klausen et al., 2017; 2019; Gilmulina et al., 2022). Towards the latest Triassic and Early Jurassic, subsidence rates gradually decreased and sometimes even became negative, as expressed by condensed units with a 20 m-thick, shallow-marine and continental sandstone-shale unit of Rhaethian (latest Triassic; ca. 208–201 Ma) to Bathonian (?) age (Middle Jurassic; ca. 168–165 Ma), truncated by several subaerial unconformities (Drachev, 2016; Faleide et al., 2018: Olaussen et al., 2018; Rismyhr et al., 2018; Müller et al., 2019). This Upper Triassic deltaic and Upper Triassic to Middle Jurassic condensed section belongs to the Kapp Toscana Group. Subsidence rates increased again during the Late Jurassic (ca. 161–145 Ma), which led to the deposition of organic-rich marine strata of the lower Adventdalen Group (e.g., Koevoets et al., 2016; 2019).

The upper Middle and Upper Jurassic/lowermost Creatceous succession preserved in Svalbard consist of the Agardhfjellet Formation, and the Lower Cretaceous succession is represented by the Rurikfjellet, Helvetiafjellet, and Carolinefjellet formations, all assigned to the Adventdalen Group (Fig. 4). These marine to continental units reflect increased subsidence with uplift in the north and northwest that formed a continental, siliciclastic platform. This uplift is related to emplacement of the HALIP across the Arctic, including Svalbard, Franz Josef Land, the New Siberian Islands, the Barents Shelf, Sverdrup Basin, northern Greenland, and the Alpha-Mendeleev Ridge, via both subaerial eruptive and intrusive magmatism (Maher, 2001; Estrada and Henjes-Kunst, 2013; Senger et al., 2014; Evenchick et al, 2015; Polteau et al., 2016; Davis et al., 2017; Dockman et al., 2018; Naber et al., 2021; Bédard et al., 2021; Galloway et al., 2022). HALIP magmatism is thought to be derived from the arrival of a thermally elevated mantle plume that caused large volumes of mafic rocks including sills, dykes, lavas, and pyroclastic material (Maher, 2001; Senger et al., 2014; Buchan and Ernst, 2018; Bédard et al., 2021; Naber et al., 2021; Heyn et al., 2024). Robust U-Pb dating points to a short magmatic pulse affecting Svalbard at ca. 124.5 Ma (Corfu et al., 2013). However, in the adjacent Sverdrup Basin (Arctic Canada), multiple magma emplacement episodes have been identified, with pulses peaking at $122 \pm 2$ Ma, at $95 \pm 4$ Ma and at $81 \pm 4$ Ma (Jens et al., 2015; Kingsbury et al., 2017; Davis et al., 2017; Dockman et al., 2018; Bédard et al., 2021; Dummann et al., 2024). In Spitsbergen, the HALIP-activity triggered southward tilting of the platform, which resulted in progradation of a sand-rich fluviodeltaic system towards the south (Steel and Worsley, 1984; Gjelberg and Steel, 1995; Worsley, 2008; Midtkandal and Nystuen, 2009; Grundvåg and Olaussen, 2017; Grundvåg et al., 2017). The extent of the uplifted and eroded area increased during the Late Cretaceous (ca. 100–66 Ma) to the whole of Svalbard, which resulted in the upper middle Albian deposits (uppermost Lower Cretaceous; ca. 113–100 Ma; Hurum et al., 2016) being unconformably overlain by Paleocene strata (Jochmann et al., 2019; Helland-Hansen and Grundvåg, 2021).



## 2.3. Cenozoic

The mid-Paleocene saw the recommencement of sediment deposition in Svalbard after a ~60 Myr hiatus in response to large-scale regional changes in plate tectonic configurations (Fig. 3). The base of the Paleocene strata in Svalbard has been dated to

61.8 Ma (Jones et al., 2017), close to the Danian–Selandian boundary (ca. 61.6 Ma). This age is contemporaneous with several changes around the Greenland microplate, including increased rifting between Greenland and Eurasia in the proto-Northeast Atlantic region (Abdelmalak et al., 2023), the first pulse of North Atlantic Igneous Province (NAIP) volcanism (Storey et al., 2007a), a change from carbonate- to siliciclastic-dominated sediments in the North Sea (Clemmensen and Thomas, 2005), widespread shear deformation along eastern Greenland (Guarnieri, 2015), and an increase in the rate of seafloor spreading in

the Labrador Sea (Roest and Srivastava, 1989; Oakey and Chalmers, 2012). The combination of seafloor spreading in the Labrador Sea and rifting along the mid-Norwegian margin instigated compression between Greenland and Svalbard, which evolved into a dextral transpressional regime as rifting transitioned to seafloor spreading in the NE Atlantic by 55 Ma (Storey et al., 2007b). This period was also coincident with the rifting and breakup of the Eurasia Basin to the north of Svalbard. The dextral transpression along the Greenland-Svalbard margin caused localized crustal shortening and the formation of the West

Spitsbergen Fold and Thrust Belt, WSFTB (Fig. 3; Harland, 1965), linked to the Eurekan deformation (ca. 63–35 Ma) and plate reorganization in the North Atlantic ( Dallmann et al., 1993; Braathen et al., 1995; 1999; Maher and Braathen, 1995; Bergh et al., 1997; Gee and Tebenkov 2004; Faleide et al., 2008; Leever et al., 2011; Blinova et al., 2013; Piepjohn et al., 2015; 2016; Gion et al., 2017). A north-south trending foreland basin, known as the Central Spitsbergen Basin (CSB) or the Central Tertiary Basin (CTB) in older literature, formed east of the WSFTB and was filled with over 1.9 km thick Paleocene

to Eocene (Oligocene?) deposits of the Van Mijenfjorden Group (Figs 3; Steel et al., 1981; 1985; Helland-Hansen, 1990; Müller and Spielhagen, 1990; Bruhn and Steel, 2003; Jochmann et al., 2020; Helland-Hansen and Grundvåg, 2021). During the Paleocene-Eocene transition (ca. 56 Ma), a passive margin started to develop to the north of Svalbard as a result of the opening of the Eurasia Basin. Finally, an Oligocene (ca. 34–23 Ma) transtensional rift phase eventually gave way to the formation of a passive margin west of Spitsbergen (Faleide et al., 2008; Lasabuda et al., 2018; Haaland et al., 2024).

The transpressional deformation related to the WSFTB was followed by NW–SE transtensional rifting that formed a series of grabens along the western Svalbard margin (Steel et al., 1985; Blinova et al., 2009; Kleinspehn and Teyssier, 2016; Kristoffersen et al., 2020; Haaland et al. 2024). The Forlandsundet Graben, one of the grabens cropping out between the islands of Prins Karls Forland and Spitsbergen (Fig. 2) contains between 1000-3000 m of Eocene to potentially Oligocene strata (Gabrielsen et al., 1992; Schaaf et al., 2021). The final separation between Greenland and the Barents Shelf margin eventually

led to the opening of the Fram Strait. A shallow and narrow gateway was initially established around 20 Ma (Jokat et al., 2016; Fyhn and Hopper, 2025), and the transition from a restricted to fully ventilated Arctic Ocean took place around 17.5 Ma (Jakobsson et al., 2007). The establishment of a deep-water connection through the Fram Strait is currently debated, with suggested ages of 13.7 Ma (Jakobsson et al., 2007), 10 Ma (e.g., Kristoffersen and Husebye, 1985; Kristoffersen, 1990), and

5 Ma (Lawver et al., 1990). During these times, Svalbard and the rest of the Barents Shelf margin experienced several changes
in motion relative to the adjacent Greenland plate.

The present archipelago configuration of Svalbard with respect to the otherwise submerged setting of the Barents Shelf is
thought to be a consequence of a combination of uplift during the Late Cretaceous, Paleocene-Eocene Eurekan deformation,
and ongoing Holocene (last 11.7 kyr) isostatic rebound (Dimakis et al., 1998; Worsley, 2008; Henriksen et al., 2011; Dörr et
al., 2013; Lasabuda et al., 2021). This uplift and exposed nature of Svalbard has resulted in the present-day exhumation of the
metamorphic succession along the northern and western coasts of Svalbard and the younger sedimentary cover as described
above (Fig. 3).

## 3. Data

Paleoclimate research in Svalbard has traditionally relied on the exceptionally exposed and vegetation-free outcrops, typical
of the present Arctic landscape. For the last two decades, research drilling across Svalbard has increasingly been utilized and
provides unique drill core material suitable for high-resolution paleoenvironmental and paleoclimate research.

### 3.1 Key stratigraphic sections

Figure 5 shows the geographic distribution of the primary study sites referenced in the 148 key publications summed up in
Table 1 and in Table 2 (listed with more details in Supplementary table). Many of these articles are centered on important sites
with good chronological and lithological constraints. Most of the outcrops are also covered by high-resolution digital outcrop
models freely available through the Svalbox database (Betlem et al, 2023), facilitating data integration. Four of the localities
are represented by 10 or more publications and are described below.

### 3.1.1 Festningen

The Festningen section in western Spitsbergen offers a nearly complete stratigraphic section spanning from the Mississippian
(ca. 359 Ma) to the Paleocene (Fig. 6). The ~7 km long section is easily accessible along the shoreline, with nearly-vertical
sedimentary layers due to Eurekan deformation. Festningen is an important regional stratigraphic profile and routinely targeted
by geologists (Hoel and Orvin 1937; Steel et al., 1978; Mørk et al.,1982; Nagy and Berge 2008; Midtkandal and Nystuen,
2009; Grundvåg et al., 2019), including those interested in deep-time paleoclimate (e.g., Grasby et al., 2015; Vickers et al.,
2023). Mørk and Grundvåg (2020) offers a geological guidebook for the section, whereas Senger et al., (2022) provided an
open-access digital outcrop model (DOM) of the 5 km long part of the protected section. The high-resolution (7 mm pixel
resolution) DOM is suitable for planning additional sampling, quantitative structural and sedimentological analyses, and
integrating existing paleoclimatic research (Fig. 6).





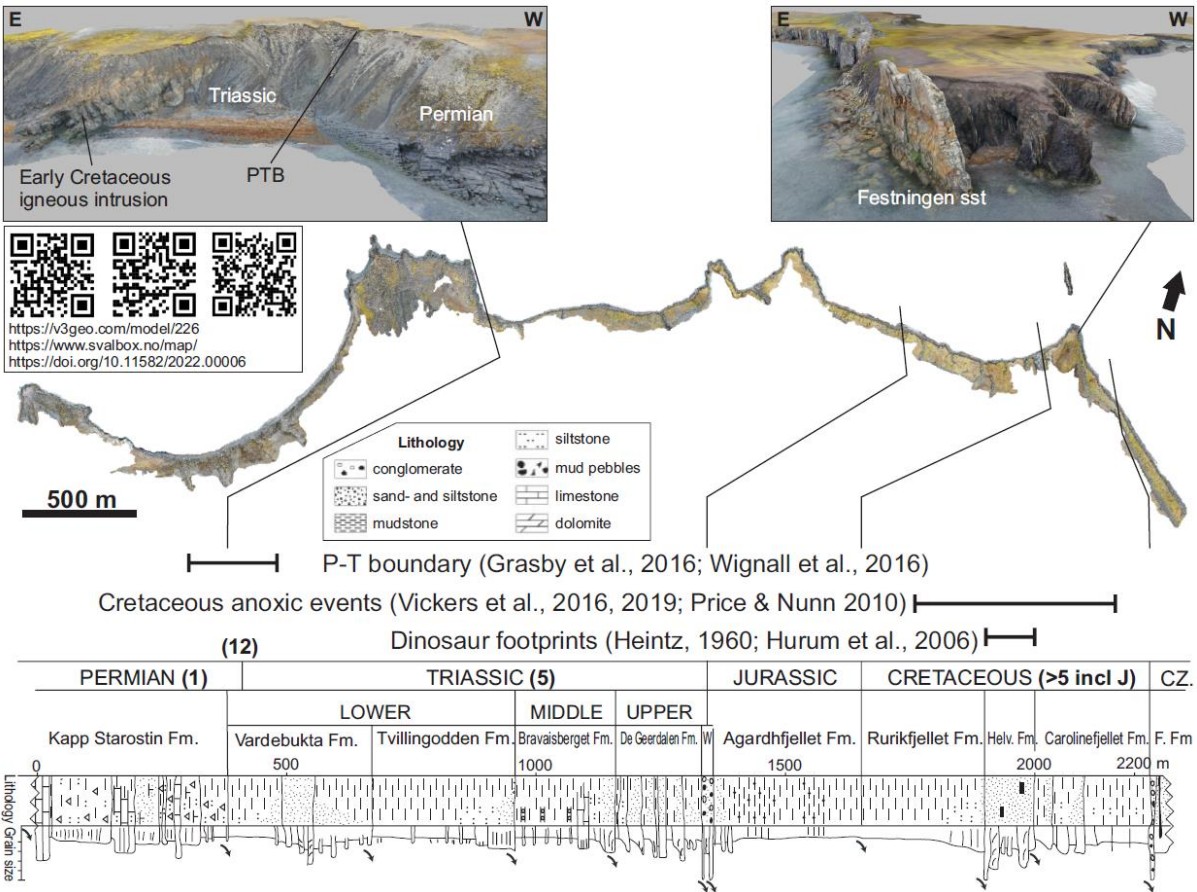

**Figure 6: Stratigraphic column of the best-preserved part of the Festningen section, from Mørk and Grundvåg (2020), tied to the**
**digital model of the entire Festningen section as presented in Senger et al., (2022). The inset images of the Permian-Triassic boundary**
**and the Festningen sandstone illustrate screenshots of the digital outcrop model that is accessible online and freely available for**
**download by following the QR codes and URLs. Paleoclimate-related research conducted on the section is highlighted for key events.**
**These include amongst others the end-Permian mass extinction and the subsequent recovery phase (e.g. Wignall et al., 1998; Grasby**
**et al., 2016), as well as several Cretaceous cooling events, anoxic events, and their associated deposits (Price & Nunn, 2010; Vickers**
**et al., 2016, 2019; Grundvåg et al., 2019). Abbreviations: CZ = Cenozoic, W = Wilhelmøya Subgroup, Helv. Fm = Helvetiafjellet Fm,**
**F. Fm = Firkanten Fm. The QR codes provide direct access to the digital model. Figure modified after Senger et al. (2022).**

### 3.1.2 Janusfjellet-Deltaneset

The Janusfjellet section in central Spitsbergen exposes an Upper Triassic to Paleocene siliciclastic-dominated succession that
has been extensively studied as part of the Longyearbyen $CO_2$ lab project (Olaussen et al., 2019). The succession includes both
the Upper Triassic-Middle Jurassic sandstone reservoirs of the Kapp Toscana Group as well as the overlying Upper Jurassic-
Lower Cretaceous Adventdalen Group. The Agardhfjellet Formation, the lowermost part of the Adventdalen Group, has also
been extensively studied as one of the richest marine reptile sites in the world, yielding sixty specimens so far (Hurum et al.,
2012; Delsett et al., 2016; 2019), along with an abundant seep fauna (Hryniewicz et al., 2015). The outcropping section dips





gently at about 3° to the south-west and exposes the same stratigraphy as in the fully cored boreholes in Adventdalen (Olaussen et al., 2019). As an excellent analog to the Longyearbyen $CO_2$ lab reservoir-caprock system, the outcrops have been systematically studied with focus on sedimentology (Rismyhr et al., 2018; Jelby et al., 2020a), fault and fracture characterization (Ogata et al., 2014a; 2014b; Mulrooney et al., 2018; Betlem et al., 2024; Rizzo et al., 2024,), sandstone injectites (Ogata et al., 2023) and paleoclimatic signals (Koevoets et al., 2018; Jelby et al., 2020b).

### 345   3.1.3 Sassendalen

Sassendalen is a key region for understanding the Permian-Triassic transition and evolution of Svalbard, and within Sassendalen there are many key sections for defining different aspects of the lithostratigraphic framework of central Spitsbergen (e.g., Mørk et al., 1999). It is also a unique area to study global ecosystem recovery after the end-Permian mass extinction (see Hurum et al. 2018; Kear et al. 2023). Deltadalen and Lusitaniadalen are two valleys on the western side of

Sassendalen that excellently expose the Permian Kapp Starostin to Botneheia formations. In addition is the more eastern Fulmardalen (Hammer et al., 2019; Hansen et al., 2024). The Deltadalen outcrop is directly next to the Deltadalen research boreholes, where deposits of the EPME and Permian-Triassic boundary were cored (Zuchuat et al., 2020). As such, it provides excellent borehole-outcrop correlation with the benefit of facilitating detailed sedimentological studies and high-resolution sampling away from the boreholes. In addition, a well-exposed section of Permian-Triassic transition crops out along

Lusitaniadalen located around 5 km northwest from Deltadalen and has been the focus of multiple studies directly focussed on the Permian-Triassic boundary (e.g., Mørk et al., 1999a; 1999b; Foster et al., 2017; Rauzi et al., 2024).

### 3.1.4 Hinlopenstretet, North Ny Friesland

The Ny Friesland section in northeastern Spitsbergen is exposed along the south coast of the Hinlopen Strait (Fig. 2) and consists of a ~1 km thick Terreneuvian-Middle Ordovician carbonaceous succession of the Oslobreen Group (ca. 539–458

Ma; Hansen and Holmer, 2010; Stouge et al., 2012; Lehnert et al., 2013; Abay et al., 2022; Smelror et al., 2024). This succession, consisting of siliciclastic shoreline facies at the base, passing up to a shallow marine warm water carbonate platform deposits, belongs to the North Atlantic/Arctic warm water carbonate platform formed on eastern Laurentia (McKerrow et al., 1991; Stouge et al., 2012).


**Table 1: Overview of selected drill core material and key sections (including digital sections)**




## Key, selected boreholes with significant core or cuttings material

| Borehole(s) | Penetrated strata (age and formation) | | Depth (meters) | Drill core availability | Complementary data sets | Key reference |
|---|---|---|---|---|---|---|
| | **Youngest** | **Oldest** | | | | |
| Gipsdalen (DD1-DD8) | Pennsylvanian Ebbadalen Fm | Precambrian Basement | 91,90 - 210.50 | Berlin-Spandau and Uni Oslo | Technical reports | Senger et al. (2019). |
| Deltadalen (DD1 and DD2) | Lower Triassic Vikinghøgda Fm | Lopingian Kapp Starostin Fm | 92,9 -99,3 | Oslo | Reports | Zuchuat et al. (2020); Schobben et al., (2020); Rodríguez-Tovar et al. (2021) |
| Tromsøbreen II | Lower Cretaceous Carolinefjellet Fm | Pennsylvanian (?) Minkinfjellet Fm (?) | 2337 | 5 cores, in Lund | Reports | Senger et al. (2019). |
| Reindalspasset | Lower Cretaceous Carolinefjellet Fm | Mississippian, Billefjorden Gr | 2315 | Cuttings, in Endalen Svalbard | Reports | Senger et al. (2019). |
| UNIS CO$_2$ lab (DH1-DH8) | Lower Cretaceous Carolinefjellet Fm | Upper triassic De Geerdalen Fm | down to 969,7 | Longyearbyen | Reports, wireline logs, publications | Corfu et al., (2013): Midtkandal et al., (2016); Polteau et al., (2016); Olaussen et al. (2019): Senger et al. (2024). |
| SNSK boreholes, including core BH9/05 | Eocene Frysjaodden Fm | Metamorphic basement | numerous | Endalen | Reports | Cui et al. (2010); Dypvik et al. (2011); Charles et al. (2011); Nagy et al. (2013); Jones et al. (2019). |
| Sysselmannbreen (BH2008-10) | Eocene (?) Oligocene (?) Aspelintoppen Fm | Late Paleocene Basilika Fm | 1085 | Endalen, Equinor lab (Bergen) | Reports | Johannessen et al. (2011); Doerner et al. (2020). |

## Key outcrop sections (> 10 publications)

| Outcrop sections | Penetrated strata (age and formation) Youngest | Penetrated strata (age and formation) Oldest | Extent | Key reference(s) |
|---|---|---|---|---|
| Hinlopenstretet, North Ny Friesland | Ordovician | Cambrian | around 10 km | Hansen and Holmer (2010); Stouge et al. (2012); Lehnert et al. (2013); Abay et al. (2022). |
| Billefjorden | Lower Permian | Upper Devonian | over 20 km of composite sections | Cutbill and Challinor (1965); Gjelberg and Steel (1981) Ahlborn and Stemmerik (2015) Berry & Marshall (2015); Davies et al. (2021); Smyrak-Sikora et al., (2021). |



| Festningen | Paleogene Firkanten Fm | Carboniferous Billefjorden Gr | 5-6 km | Hoel & Orvin (1937); Steel et al. (1978); Mørk et al. (1982); Nagy & Berge (2008); Midtkandal & Nystuen (2009); Grasby et al. (2015); Grundvåg et al. (2019); Mørk & Grundvåg (2020); Senger et al. (2022); Vickers et al. (2023). |
|---|---|---|---|---|
| Deltadalen and Lusitaniadalen | Lower Triassic Vikinghøgda Fm | Lopingian Kapp Starostin Fm | around 7 km | Dagis & Korcinskaja (1987); Mørk et al. (1999); Foster et al., (2017); Zuchuat et al., (2020); |
| Deltaneset-Janusfjellet | Lower Cretaceous Carolinefjellet Formation | | around 7 km | Hurum et al., (2018); Rismyhr et al., (2018); Olaussen et al. (2019); Koevoets et al. (2018); Jelby et al. (2020a; 2020b). |

### 3.1.5 Billefjorden and Mumiendalen

The inner part of Billefjorden exposes the Billefjorden Fault Zone, where the several km-thick Devonian Old Red Sandstone (ORS) deposits on the west are faulted against the several metamorphic and sedimentary successions in the east. These include
pre-Devonian metamorphic basement, up to 250 m of Mississippian units and over 2 km of Pennsylvanian to Permian deposits (Braathen et al., 2012; Smyrak-Sikora et al., 2018; 2021). In Mumiendalen, the lowermost Upper Devonian deposits of the Andrée Land Group expose plant fossils that represent terrestrialization and evolution of one of the oldest forests in the world, that likely thrived in a monsoonal climate (Berry and Marshall, 2015; Davies et al., 2021). Above the fossil Devonian forests units, Mississippian siliciclastics with coal seams belonging to the Billefjorden Group were deposited in a humid and tropical
climate (Cutbill and Challinor 1965; Gjelberg and Steel, 1981). The transition to the overlying mixed siliciclastic-carbonate-evaporite units of the Gipsdalen Group illustrates that the climate changed from subtropical semi-arid to an arid climate prevailing in through the Pennsylvanian to Cisuralian (Holliday and Cutbill, 1972; Gjelberg and Steel, 1981; Johannessen and Steel, 1992; Blomeier et al., 2011; Ahlborn and Stemmerik, 2015; Sorento et al., 2020; Smyrak-Sikora et al., 2021). Up section, the Bashkirian to Sakmarian part of the Gipsdalen Group consisting of interbedded evaporites and carbonates formed in
response to glacioeustatic sea level fluctuations (Stemmerik, 2000; 2008; Ahlborn and Stemmerik., 2015; Sorento et al., 2020; Smyrak-Sikora et al., 2021) related to the late Paleozoic Ice Age (LPIA; Gastaldo et al., 1996; Montañez et al., 2007; Isbell et al., 2008).

### 3.2 Drill cores

Table 1 summarizes the drill cores presently available from Svalbard. They are mostly from coal exploration by Store Norske
Spitsbergen Kulkompani (SNSK) and $CO_2$ storage research drilling in Adventdalen by The University Centre in Svalbard (UNIS); Olaussen et al., 2019; Senger et al., 2024). In addition, stratigraphic research boreholes, drilled purely out of scientific interests also exist, including the Sysselmannbreen (Johannessen et al., 2011) and Deltadalen (Zuchuat et al., 2020) boreholes. Limited core material from past petroleum exploration efforts are known to exist (eighteen boreholes were drilled from 1960



to 1994; see review by Senger et al., 2019), but these have, hitherto, not been used in any paleoclimate investigations.
Nonetheless, the associated wireline data from these wells are important calibration points for regional correlation of outcrops and onshore seismic reflection data.

### 3.2.1 Coal drilling

Exploration coal drilling focused on Paleocene stratigraphy of the Van Mijenfjorden Group in the Central Spitsbergen Basin
(CSB) and Mississippian coal-rich strata of the Billefjorden Group near Pyramiden. Lower Cretaceous and Eocene coal bearing strata were targeted in minor campaigns. Cores from the Russian/Soviet coal mining company Trust Arktikugol including sites near Barentsburg, Colesdalen and Pyramiden, are not available and likely lost as evidenced by defunct core sheds scattered around Svalbard. However, reports from these drill cores (Verba, 2013) have been used locally to constrain surface geological mapping, such as in Mimerdalen (Piepjohn and Dallmann, 2014) and in the Billefjorden Trough (Smyrak-Sikora et al., 2021).
The Norwegian coal mining company SNSK stores most of its cores in Endalen near Longyearbyen with drill dates ranging from the late 1960s to 2014, the last year of Norwegian coal exploration. These cores were investigated in several paleoclimate-related studies, particularly across the PETM and North Atlantic Igneous Province (NAIP)-related ash deposits (Table 1; Dypvik et al 2011; Jones et al, 2019).

### 3.2.2 Longyearbyen CO2 lab, DH1 to DH8

Eight boreholes were fully cored near Longyearbyen from 2007 to 2013 to characterize a potential $CO_2$ storage site (Braathen et al., 2012; Olaussen et al., 2019; Senger et al., 2024). Four of the boreholes reach the planned Upper Triassic units of the Kapp Toscana Group target storage unit at 670-700 m, while the other boreholes focus on the cap rock and overburden succession of the Adventdalen Group. The full coring across the shale-dominated cap rocks provides important constraints on the stratigraphy of the Jurassic-Cretaceous strata (Koevoets et al., 2016; Midtkandal et al., 2016; Jelby et al., 2020a; 2020b;
Śliwińska et al., 2020) and also contributed to refining the global geological time scale (Zhang et al., 2021). Senger et al. (2024) provide a full overview of the data sets generated by the project including a live database of the resulting publications, including those focusing on deep-time paleoclimate.

### 3.3.3 Sysselmannbreen, BH10-2008

The BH10-2008 (also known as Sysselmannbreen) research borehole was drilled and fully cored in 2008 to recover a full
section of the world-famous Eocene-Oligocene (?) clinoform succession of the Van Mijenfjorden Group in the CSB (Johannessen et al., 2011). The 1085 m-long core was split with one half stored in a container in Endalen, outside of Longyearbyen, and the other half stored in Equinor's laboratory in Bergen (Doerner et al., 2020).





### 3.3.5 Deltadalen, DD-1 and DD-2

The most recent research drilling in Svalbard was conducted in 2014 at Deltadalen specifically to target the uppermost Permian
part of the Tempelfjorden Group and Lower Triassic succession of the Sassendalen Group with a specific interest in the EPME
and its aftermath (Zuchuat et al., 2020). The two ca. 100 m deep boreholes were drilled and fully cored over a 1-week period,
with all equipment transported onsite by helicopter. The drill cores are stored at the University of Oslo.

**Table 2. Summary of proxies based on Table A1 (Supplementary material), which reviews 148 selected papers. The data set is plotted**
**in Figure 13.**

| | | | | | | | | |
|---|---|---|---|---|---|---|---|---|
| **N of articles** | **N of papers** | 10 | 11 | 27 | 22 | 30 | 30 | 18 |
| | **-boreholes** | 0 | 0 | 0 | 3 | 2 | 10 | 9 |
| | **-outcrops** | 10 | 11 | 27 | 21 | 30 | 27 | 11 |
| **Proxies and data** | **Biological indicators** | 8 | 8 | 15 | 14 | 22 | 24 | 10 |
| | **Climate sensitive facies** | 3 | 7 | 18 | 2 | 2 | 3 | 9 |
| | **Carbon isotopes δ¹³C** | 2 | 2 | 8 | 8 | 6 | 13 | 6 |
| | **Oxygen isotopes δ¹⁸O** | 1 | 1 | 4 | 0 | 0 | 4 | 1 |
| | **TOC/Rock-Eval** | 2 | 2 | 1 | 3 | 6 | 7 | 6 |
| | **Mercury, Hg** | 0 | 0 | 0 | 2 | 1 | 1 | 1 |
| **Stratigraphic intervals** | | **Cambro-Silurian** | **Devonian** | **Carboniferous-Permian** | **Permian-Triassic boundary** | **Triassic** | **Jurassic-Cretaceous** | **Paleogene** |

### Deep-time paleoclimate in Svalbard

### 4.1 Early Paleozoic (Cambrian, Ordovician, Silurian, 538.8-419.2 Ma)

### 4.1.1. Cambrian to Middle Ordovician- the Great Ordovician Biodiversification Event

The early Paleozoic registered two of the greatest evolutionary events in the history of life: the Cambrian Explosion (ca. 540–
510 Ma), and the Great Ordovician Biodiversification Event (GOBE; ca. 497–445 Ma.; Webby et al., 2004; Servais and Harper,



2018). This upper Furongian to Upper Ordovician diversification event is linked to cooling of previously very warm tropical oceans (Webby et al., 2004; Servais and Harper, 2018).

During the Cambrian and Ordovician, Svalbard was in a near-equatorial position (Torsvik et al., 2012; Fig. 2). The

Terreneuvian to Middle Ordovician Oslobreen Group (Fig. 4) strata preserved in Svalbard formed on a large carbonate platform along the northern margin of Laurentia, also exposed in the Northeast Greenland Basin and eastern North Greenland Basin (Stouge et al., 2012; Fig. 2). The mildly deformed, 1–1.2 km thick sandstones, fossiliferous limestone and dolomite are preserved in northeastern Spitsbergen (Fig. 7; Harland and Wilson, 1956; Oslobreen Series in Gobbett and Wilson, 1960; Fortey and Bruton, 1973; Stouge et al., 2011; 2012; Dallmann et al., 2015). The succession is potentially interrupted by a ~15

Myr-hiatus spanning over the Series 2, Miaolingian, Furongian, and possibly the earliest Ordovician (Fortey and Bruton, 1973; Smelror et al., 2024), although the lack of dateable fossils might affect this interpretation (Smelror et al., 2024). The Oslobreen Group shows surprisingly low maximum burial temperatures (Bergström, 1980; Abay et al., 2022) and eastward increasing tectonothermal influence linked to the Caledonian Orogeny (Johansson et al., 2004). The trilobites and fauna generally show a Pacific and Laurentian affinity (Fortey and Bruton, 1973; Hansen and Holmer, 2010; Stouge et al., 2012).

In Ny Friesland, the Terreneuvian microbial laminated limestone/dolomite contain cm-scale erratic chert nodules. The Lower to Middle Ordovician (ca. 485–458 Ma) carbonates were deposited in a paleotropical marine shelf setting experiencing episodes of water column redox-stratification (Lee et al., 2019). Stouge (2012) also interprets the Tremadocian (Lower Ordovician; ca. 485–478 Ma) environment in Svalbard and Greenland as a typical tropical shelf. Occurrence of oolite beds interbedded with domed stromatolites throughout the Tremadocian on Ny Friesland and adjacent islands (Kröger et al., 2017)

is consistent with a peritidal tropical carbonate factory. Uchman and Hanken, (2024) recognise that carbonates of the uppermost part of Terreneuvian and the Cambrian Series 2 contain pseudomorphs after evaporites.  Hansen and Holmer (2010) recognize strong ties of Lower and Middle Ordovician brachiopods to faunas in North America and Greenland at the generic level. Hansen and Holmer (2010) also discuss the transition from low diversity brachiopod fauna in the Tremadocian and early Floian (Early Ordovician; ca. 478–470 Ma), followed by an abrupt diversification event in the late Floian and into the Middle

Ordovician. The hypersaline conditions, however, mask the expected record of the Great Ordovician Biodiversification Event (Uchman and Hanken, 2024). The Middle Ordovician part of the succession represents an overall deepening, transgressive sequence (Kröger et al., 2017; Lee et al., 2019). Kröger et al., (2017) suggest that the gradual transition to deeper deposits with more shale and local siltstone and glauconitic horizons accompanied with increased burrowing and fossiliferous, cherty mud-wackestone and skeletal grainstone are evidence of general climate cooling in the transition to the Middle Ordovician (i.e., the

incipient phase of Ordovician cooling). Although changing to colder sea floor conditions, tropical carbonate production continued in an inner carbonate ramp while a cold-water carbonate factory prevailed in the outer ramp (Smelror et al., 2024).



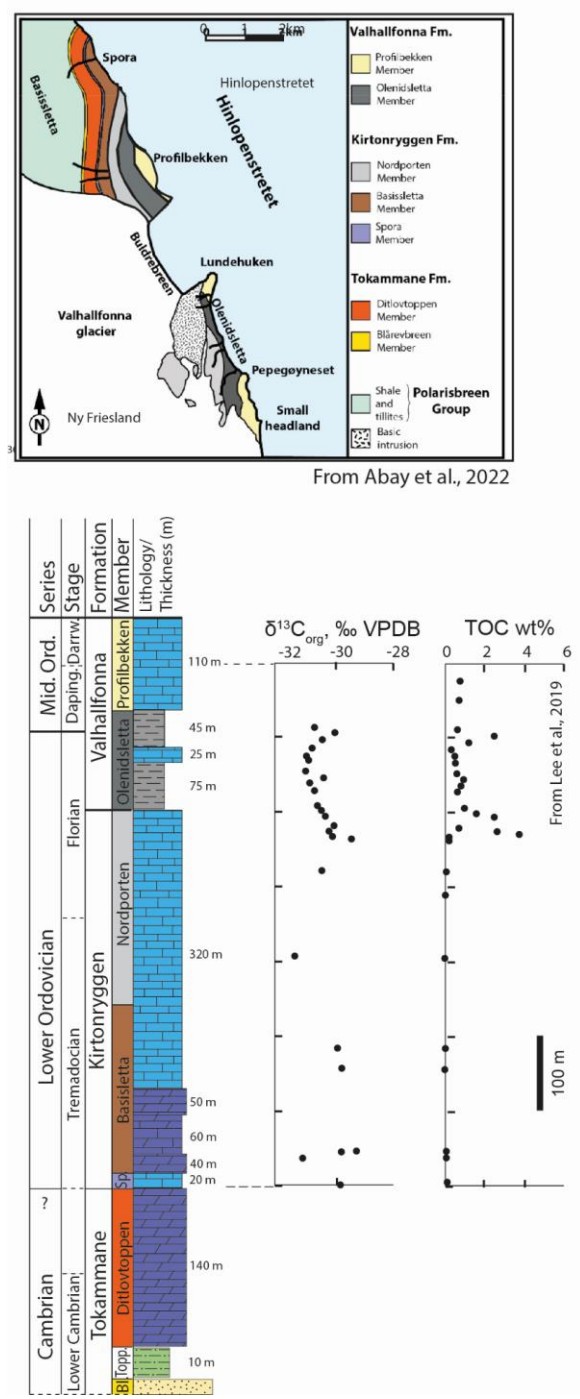

**Figure 7 Stratigraphic section of the Cambrian and Ordovician Oslobreen Group succession in Ny Friesland, northern Spitsbergen**
**(see Fig. 5 for location) After Fortey and Bruton (1973); Harland (1997); Stouge et al. (2012); Lehnert et al. (2013); Lee et al., (2019). Bulk organic carbon isotopes ($\delta^{13}C_{org}$, in ‰ VPDB); Total Organic Carbon (TOC, in weight percent); Bl.- Blårevbreen Member; Topp.- Topiggane Member; Sp.- Spora Member.**





## 4.2 Late Paleozoic (Devonian, Carboniferous, Permian, 419.2-251.9 Ma)

The Devonian, Carboniferous, and Permian periods record the only complete greenhouse to icehouse to greenhouse cycle
(LPIA) on a vegetated Earth (cf. Isbell et al., 2008). In Svalbard, the relatively complete Devonian to Permian sedimentary
succession, which encompasses the Old Red Sandstone, Billefjorden, Gipsdalen, and Tempelfjorden groups (Fig. 4), provides
a unique opportunity to study responses of the tropical and near-tropical depositional systems with the terrestrial and shallow-
marine settings to the LPIA glaciations. Svalbard occupied a near equatorial position for most of the Devonian and
Carboniferous and from Permian started northwards drift (Fig. 2; Torsvik and Cocks, 2019).

### 4.2.1. Devonian: Old Red Sandstone, terrestrialization, and first forest

The advent of terrestrial vascular plants in the latest Silurian-earliest Devonian influenced weathering processes, soil
formation, and strongly impacted the $CO_2$ cycle and global climate (Gensel et al., 2008 and references therein; Kenrick et al.,
2012). The impact of vascular plants can be observed in Svalbard in the Uppermost Silurian to Upper Devonian Old Red
Sandstone succession (Friend et al., 1997; Blomeier et al., 2003a; 2003b). This >8 km thick succession is restricted to
extensional collapse basins formed in pure extensional (Piepjohn and Dallmann, 2014), or more likely, transtensional settings
(Braathen et al., 2018). The Andrée Land Basin exposed in central-north Spitsbergen was filled mainly by a terrestrial
succession, with marginal-marine conditions recorded in the northernmost part (Blomeier et al., 2003a). It notably includes
red and gray-green fluvial, alluvial, lacustrine, and coastal sedimentary strata arranged into fining-upward units, with abundant
plant material (Friend, 1965; Moody-Stuart, 1966; Blomeier et al., 2003a; 2003b; Piepjohn and Dallmann 2014). The
succession recorded indications of long-term climatic variability, such as shifts in paleosols from calcretes and vertisols to coal
and preservation of in situ tropical forests (Berry and Marshall, 2015). The biological evidence of environmental conditions
recorded in the Old Red Sandstone come from plant fossils (Berry 2005; Berry and Marshall 2015; Davies et al., 2021),
palynomorphs (Vigran, 1964; Allen, 1965; 1967; Friend et al., 1997), vegetation-induced sedimentary structures (Davies et
al., 2021), and scarce marine-influenced fauna including ostracods and bivalves (e.g., Friend, 1961; Worsley, 1972). Bulk
geochemistry along with extraction and biomarker analysis of Middle Devonian coal indicate a terrestrial plant origin with
high liptinite content (Vogt, 1941; Blumenberg et al., 2018). The flora evolved over time from diminutive plants in the Middle
Devonian to the first in situ forests of lycopsids and archaeopterids in the early Frasnian (Late Devonian; ca. 383–372 Ma;
Berry and Marshall 2015; Davies et al., 2021). Based on sedimentological and biological evidence for highly variable seasonal
discharge and scarcity of thick calcretes, Davies et al., (2021) suggested that the precipitation regime in the Devonian was
tropical and monsoonal, and that the stratigraphic partitioning into red bed and gray-green strata attest to long-term fluctuations
in drainage and oxidizing conditions.



**4.2.2. Carboniferous to Cisuralian: Late Paleozoic Ice Age (LPIA)**

The Late Paleozoic Ice Age (LPIA) is one of the most important climatic events of the Phanerozoic that significantly influenced climate and depositional systems on Earth (Gastaldo et al., 1996; Montañez et al., 2007; Isbell et al., 2008). The LPIA is the
closest analog to present climate conditions, characterized by discrete periods of glaciations separated by warm interglacials (Montañez and Poulsen, 2013). The LPIA glaciations started ca. 347 Ma at the onset of the Viséan (Mississippian), and lasted at least until ca. 285 Ma during the middle of the Artinskian (Cisuralian), potentially extending until ca. 260 Ma around the Guadalupian-Lopingian transition in the more Alpine settings in eastern Gondwana (Montañez and Poulsen, 2013). During this time, Svalbard drifted from a tropical position into the northern subtropical warm arid zone (Fig.2; e.g. Torsvik and Cocks,
510 2019).

The early stage of the LPIA coincided with deposition of the Mississippian Billefjorden Group. This succession unconformably overlies the folded Paleo-Neoproterozoic and Devonian successions. The terrestrial deposition occurred on broad floodplains and included abundant coal seams deposited under the humid tropical climate (Fig. 3; Gjelberg and Steel, 1981; Fairchild et al., 1982; Steel and Worsley, 1984; Lopes et al., 2019). The coal-bearing succession reaches up to 55 m thickness in eastern
Spitsbergen (Scheibner et al., 2012), and 350 m in central Spitsbergen (Gjelberg and Steel, 1981; see borehole SLE 116 in Smyrak-Sikora et al., 2021). Over 1200 m of cumulative thickness is reported along the west and south parts of Spitsbergen (Gjelberg and Steel, 1981), where this succession is repeated several times due to the Paleocene and Eocene WSFTB (Maher et al., 1995; Braathen and Bergh 1995; Fig.10 in Horota et al., 2022). The fluvio-lacustrine coal deposits were commercially mined in Pyramiden from 1910 to 1998. The coal is characterized by relatively heavy $\delta^{13}$C values, a low gammacerane index
and high Pr/Ph ratios, distinctive from the Pennsylvanian coals associated with evaporites (Nicholaisen et al., 2019). Based on spores and plant fossils, Scheibner et al., (2012) suggested that the Billefjorden Group strata were deposited in a humid climate, in accordance with a paleogeographic position 10-15 °N (Fig. 2).

The shift from humid tropical to warm, arid to semi-arid depositional environments occurred during the late Serpukhovian (Mississippian) at the boundary between the Billefjorden and Gipsdalen groups (Fig. 3; Holliday and Cutbill, 1972; Gjelberg
and Steel, 1981; Johannessen and Steel, 1992) and coincides with the initiation of regional-scale rifting in Svalbard and the Barents Shelf (Nøttvedt et al., 1993; Faleide et al., 2008; Braathen et al., 2012; Smyrak-Sikora et al., 2018; 2021). The following mixed siliciclastic-evaporite-carbonates succession of the lower Gipsdalen Group was deposited during the Pennsylvanian in an array of north-south striking rift basins (Gjelberg and Steel 1981; Smyrak-Sikora et al., 2021). The shift from the Billefjorden Group to the Gipsdalen Group is abrupt across most of Svalbard, and the boundary likely represents a
period of non-deposition or erosion, especially on the structural highs. Contrastingly, in the inner part of one of the Billefjorden Trough, the transition is more gradual and occurs in a fluvial succession where the only changes recorded in the meandering river system is the shift from humid to arid-climate soil profiles (Olaussen et al., 2023). This change in climate setting does not correspond to recognized northward drift of Svalbard (Torsvik and Cocks, 2019) and the reasons for this change are poorly




understood. The Billefjorden Trough began as a continental rift basin followed by the opening of a connection to the ocean in

the Bashkirian, which made the preposition sensitive to glacio-eustatic sea level variations (Smyrak-Sikora et al., 2019; 2021).

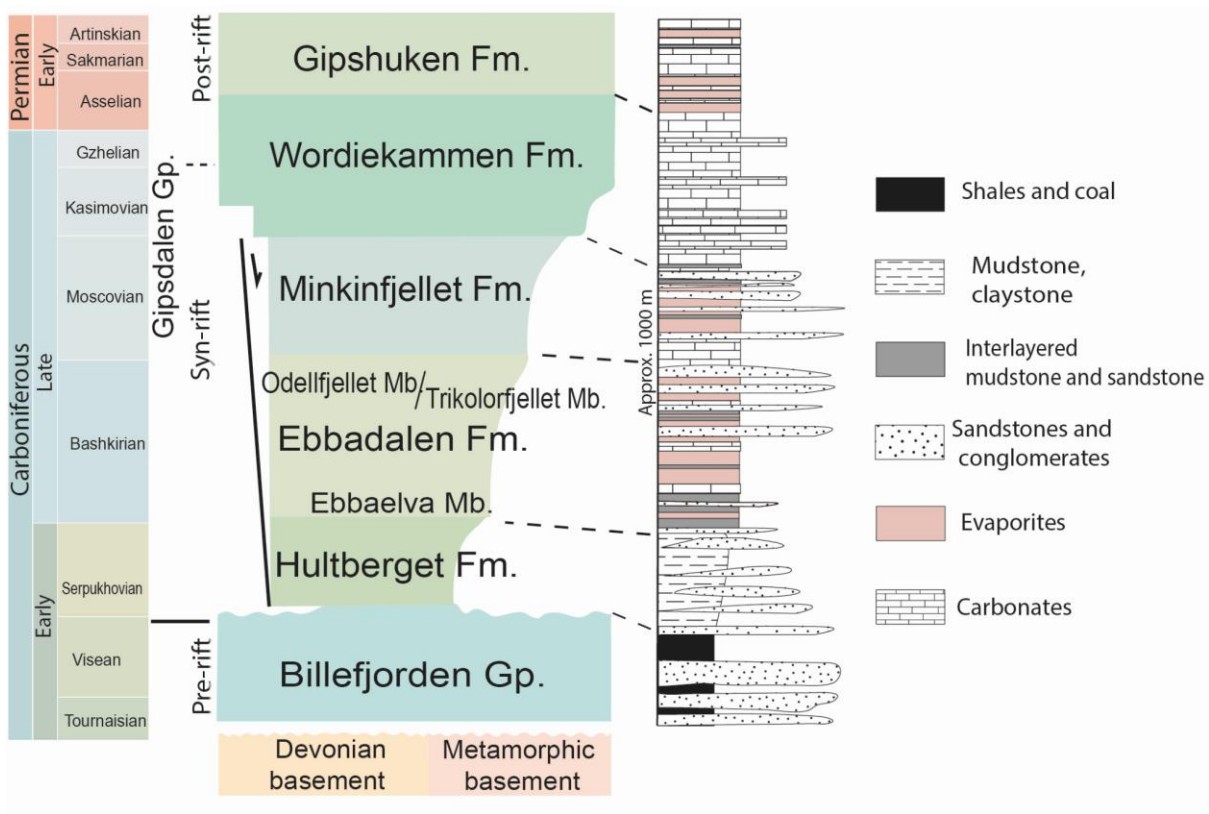

**Figure 8. Carboniferous stratigraphy and simplified lithological profile demonstrating a shift from humid-tropical climate during deposition of the coal-bearing Billefjorden Group to the semi-arid to arid climate of the Gipsdalen Group seen as a change of climate**
**sensitive facies from coal-bearing units to red siliciclastics, evaporites and warm water carbonates. Modified from Braathen et al. (2011) and Smyrak-Sikora et al. (2021).**

The impact of the LPIA glaciations and deglaciations is most readily recognized in the Bashkirian (Mississippian; onset ca.

323 Ma) to Sakmarian (Cisuralian; ca. 293–290 Ma) part of Gipsdalen Group, namely the paralic to marine syn- to post-rift

succession comprising the upper Ebbadalen to Gipshuken formations (Fig. 8). Glacio-eustatic sea-level variations related to

LPIA significantly impacted sedimentation in shallow shelf and coastal environments. Episodes of sea-level lowstands are

represented by terrestrial siliciclastics, gypsum strata that precipitated in salinas and sabkhas, karst and exposure surfaces

which are interbedded with restricted to open-marine carbonate deposits formed during the sea level highstands (Stemmerik,

2000; 2008; Ahlborn and Stemmerik., 2015; Sorento et al., 2020; Smyrak-Sikora et al., 2021). The number of cycles in the

Bashkirian to Sakmarian part of the Gipsdalen Group exceeds 130 cycles (Ahlborn and Stemmerik, 2015; Sorento et al., 2020;





Smyrak-Sikora et al., 2021), however, the lack of good stratigraphic control limits cyclostratigraphic constraints. The LPIA in Svalbard is manifested also by Asselian (peak icehouse) atmospheric dust load estimated to be higher than the Moscovian (moderate icehouse; Oordt et al., 2020). This is consistent with the record from the Russian Platform that shows a $\delta^{18}$O maximum during the glacial maximum in the Asselian (Grossman et al., 2008). Asselian to Artinskian ca. 2.5‰ decrease in

$\delta^{18}$O in the Southern Urals is attributed to ca. 4-7 °C increase in temperature and used as evidence for glacial retreat (Korte et al., 2005). This glacial retreat is questioned by Grossman et al., (2008) who show increasing $\delta^{18}$O values related to aridification. A major deglaciation event is, however, seen in the Sakmarian as regional deepening of the carbonate platforms and temporary stop of glacio-eustatic cyclic deposition (Ahlborn and Stemmerik, 2015; Sorento et al., 2020).

**4.2.3. Late Permian: cold-water carbonate platform**

A transition from warm-water carbonate-evaporite deposition to the temperate to cool-water mixed siliceous–carbonate ramp occurred in the upper Artinskian (Cisuralian; ca. 285 Ma), and corresponds to the transition from the Gipsdalen Group to the Tempelfjorden Group (Ezaki et al., 1994; Stemmerik, 2000; 2008; Hüneke et al., 2001; Stemmerik and Worsley, 2005; Blomeier et al., 2009; 2013; Buggisch et al., 2012; Dustira et al., 2013; Sorento et al., 2020; Olaussen et al., 2025). The

transition is attributed to the continued northern drift of Svalbard, closure of the Uralian seaway to the warmer Tethys to the southeast (Stemmerik 2008) and was likely also the result of a deepening of the entire shelf (Blomeier et al., 2013). For the remainder of the Permian, cool- to cold-water conditions prevailed along the northwestern margin of Pangea, leading to the deposition of a ca. 460 m thick succession dominated by spiculitic chert and cool-water carbonates (Cutbill and Challinor 1965; Blomeier et al., 2013; Uchman et al., 2016; Matysik et al., 2018).

Extensive oxygen isotopic data has been derived from brachiopods from the Artinskian-Changhingian (middle Cisuralian to Lopingian; ca. 285-252 Ma) Kapp Starostin Formation (e.g., Gruszczynski et al., 1989; Mii et al., 1997; Korte et al., 2005; Nielsen et al., 2013). However, the high variability in the isotopic data from these Permian brachiopods (e.g., Gruszczynski et al., 1989; Korte et al., 2005) is inconsistent with the marine habitat of these taxa (Grossman et al., 2008). Diagenetic alteration accounts for the low $\delta^{18}$O values of most samples (Mii et al., 1997). To exclude diagenetically altered brachiopods, many

researchers have used geochemistry and petrography (e.g., Mii et al., 1997) and targeted the best-preserved parts of shells. Excluding these potentially altered values, brachiopod $\delta^{18}$O values are generally -2 to -7 ‰ for the Kungurian-Wuchiapingian (upper Cisuralian-Lopingian; ca. 283–254 Ma) interval of the Kapp Starostin Formation (Mii et al., 1997). The Guadalupian-Lopingian $\delta^{13}$C maximum of 7.5‰ represents the highest spiriferid brachiopod $\delta^{13}$C values in the Phanerozoic (Gruszczynski et al., 1989; Mii et al., 1997) and may reflect changes in global storage of organic carbon (Mii et al., 1997). Matysik et al.

(2017) investigated the multistage diagenesis of the Kapp Starostin Formation, at medium burial depths with deep-burial overprinting. The cement in Kapp Starostin Formation shows $\delta^{18}$O values between +2‰ and 30‰ and demonstrates a peak at temperatures of 23 to 43°C corresponding to burial depths of 0.6 to 1.3 km (Matysik et al., 2018).




### 4.2.4. Capitanian Crisis

Within the Kapp Starostin Formation, it has also been proposed that a less severe middle Permian (Capitanian) mass extinction
is recorded (Bond et al., 2015). This Capitanian crisis is thought to be indicated by a negative $\delta^{13}C_{org}$ isotope excursion, a
lithofacies change (i.e., the loss of carbonate beds), and a drop in species richness (Bond et al., 2015). In addition, these changes
have been correlated to similar changes in the Sverdrup Basin, Arctic Canada (Bond et al., 2020). These changes are also
associated with redox proxies (pyrite framboids, Th/U and V/Al) suggesting the development of anoxic conditions, and the
loss of carbonates from the Kapp Starostin Formation was interpreted to be the consequence of a sustained interval of ocean
acidification (Beauchamp and Grasby, 2009; Grasby et al., 2015), making this transition consistent with other hyperthermal
events. The timing of these changes is also consistent with the changes observed at tropical paleolatitudes (Sun et al., 2010;
Wignall et al., 2012), suggesting that the Capitanian crisis was a global event. However, the interpretation that the Capitanian
crisis is recorded in the Kapp Starostin Formation is disputed, owing to the lack of biostratigraphical data confirming the rock
are of Capitanian age (Nakazawa 1999; Shen et al., 2005; Lee et al., 2022). A reanalysis of the same sections using brachiopod
data suggested that this event is not the Capitanian crisis, but instead a faunal turnover that occurred during the Kungurian
(Lee et al., 2022). Moreover, the development of ocean acidification is interpreted to have persisted for millions of years
(Grasby and Bauchamp, 2009), although the large carbonate buffering capacity of ocean water suggests that ocean acidification
is unlikely to persist for such long intervals of time (Hönisch et al., 2012). It, therefore, remains equivocal whether the Kapp
Starostin Formation records the Capitanian Crisis.



# 600 **4.3. Mesozoic (Triassic, Jurassic, Cretaceous, 252.2–66 Ma)**





**Figure 9. Top: Geological map of central Spitsbergen highlighting the drill site (yellow star) and the adjacent Deltadalen river section (green star; geological map adapted from Major et al., 1992). Overview of the sedimentary section illustrating the lithostratigraphic formations in central Spitsbergen (after Dallmann et al. (1999), Mørk et al. (1999b), Midtkandal et al. (2008), Nagy and Berge (2008), Dypvik et al. (2011), Blomeier et al. (2013), Lord et al. (2014), Koevoets et al. (2016), and Smelror and Larssen (2016). Bottom: The Permian/Triassic boundary as it appears in the Deltadalen DD-1 drill core. Modified after Zuchuat et al. (2020). EMPE – End-Permian Mass Extinction Event; Kp. St. Fm. - Kapp Starostin Formation; Mbr. – member; Fm – formation; H.p - Hindeodus parvus; * - member boundary after Mørk et al. (1999b); \*\*Induan-Olenekian boundary age after Burgess et al., (2014); \*\*\*palynological data from Vigran et al. (2014).**

### 4.3.1 The Permian-Triassic transition

The End-Permian Mass Extinction Event (EPME) at ca. 252 Ma (Burgess et al., 2014) was the most catastrophic extinction event of the Phanerozoic, which decimated 75% of terrestrial species (Hochuli et al., 2010) and 81% of marine species (Stanley, 2016). This extinction is associated with a marked and continuous global negative carbon isotope excursion (CIE; Korte and Kozur, 2010). The cause(s) of the EPME are debated; both a bolide impact, and the emplacement and eruptions of the Siberian Traps LIP are implicated as causal mechanisms (Svensen et al., 2009; 2018; Grasby et al. 2011; Sanei et al.2012; Ogden and Sleep, 2012; Ivanov et al., 2013; Burgess and Bowring, 2014; Wu et al., 2021). Even though the intrusive and extrusive character of the Siberian Traps LIP is generally accepted as the extinction trigger that led to the cascading environmental changes, it is not fully understood which environmental changes led to the collapse of terrestrial and marine ecosystems (e.g., Hochuli et al., 2010; Korte and Kozur, 2010; Black et al., 2014; Grasby et al., 2015; Joachimski et al., 2020; Scotese et al., 2021; Wu et al., 2021; Galloway and Lindström, 2023a).

In Svalbard, the mass extinction event is expressed differently compared to equatorial Tethyan carbonate successions. In west and central Spitsbergen, the pre-extinction interval belongs to the uppermost part of the Kapp Starostin Formation, which is usually devoid of any skeletal fossil material (Fig. 9; Bond et al., 2015; Grasby et al., 2015; Lee et al., 2022). The only exception is the poorly preserved lingulid brachiopod species documented (e.g., Gobbet 1964) and rare impressions of large brachiopods and bivalves (WJ. Foster, pers. obvs.) so far, making it virtually impossible to robustly reconstruct diversity dynamics during this important interval (Uchman et al., 2016; Foster et al., 2022). The Permian Kapp Starostin Formation is, therefore, poorly age-constrained, and no index fossils of Changhsingian (pre-extinction) age have yet been identified. Based on sedimentological evidence as well as on the nature of the sharp negative $\delta^{13}C_{org}$ excursion, the upper part of the Kapp Starostin and the overlying Triassic Sassendalen Group (separated in the Vardebukta Formation in the west and Vikinghøgda Formation in the east) have been interpreted to represent continuous deposition across the Permian-Triassic boundary (Wignall et al., 1998; Schobben et al., 2020; Zuchuat et al., 2020). This transition is also associated with the abrupt disappearance of cemented, highly bioturbated, spiculite- and chert-bearing mudstones and sandstones, conformably overlain by easily weathered, usually laminated, scarcely bioturbated, silica-poor mudstones (e.g., Mørk et al., 1993; 1999a; 1999b; Uchman et al., 2016; Rodriguez-Tovar et al., 2021). The EPME has, therefore, been interpreted as a single horizon at the base of the Vardebukta Formation in



west Spitsbergen (Wignall et al., 1998) and ~1.6 m into the Vikinghøgda Formation in central Spitsbergen (Mørk et al., 1999; Nabbefeld et al., 2010).

The post-extinction sediments, which in Svalbard are assigned to the Lower Triassic Vardebukta and Vikinghøgda formations, yield a scarce and low-diversity ichno-assemblage (Wignall et al., 1998; Uchman et al., 2016; Rodriguez-Tovar et al., 2021), as well as abundant macrofossils, including key Triassic index fossils (ammonoids and conodonts), which have been useful in inferring the timing of the extinction and the recovery. From the base of both formations, a diverse assemblage of ammonoids have been recorded including *Otoceras boreale*, *Glyptophiceras nielseni*, *Ophiceras spathi*, *O.* cf. *compressum*, *O.* cf. *kochi*, *O.* cf. *poulseni*, *Paravishnuites paradigma*, and *P. oxynotus* (Mørk et al., 1999), which occur ca. 6 m above the base of the Vikinghøgda Formation (Nakrem et al., 2008) and 2.5 m into the Vardebukta Formation (WJ. Foster, pers. obvs.). Mørk et al. (1999), Nakrem et al. (2008), and Zuchuat et al. (2020) also described conodonts from the base of the Vikinghøgda Formation, which suggest that the interpreted extinction horizon in Svalbard is time equivalent with the onset of the mass extinction at the Global Stratotype Section and Point (GSSP) in Meishan, China, and the Permian-Triassic boundary occurs 4.1 m into the Vikinghøgda Formation. This is similar to other high-latitude clastic Permian-Triassic successions in Jameson Land, Greenland (Twitchett et al., 2001), the Sverdrup Basin in Canada (Henderson and Baud, 1997), and South Verkhoyansk region, Russia (Biakov et al., 2016). Furthermore, in South Verkhoyansk, the Changhsingian sandstones record bivalve communities with large Intomodesma species that suddenly go extinct at the base of the Otoceras concavum ammonoid zone (Biakov et al., 2016), coincident with the change in bioturbation record in Svalbard (Uchman et al., 2016; Rodriguez-Tovar et al., 2021).

Numerous geochemical and sedimentological studies have investigated the environmental changes recorded in Svalbard associated with the EPME. The negative $\delta^{13}C_{org}$ and $\delta^{13}C_{carb}$ isotope excursions, which occur just prior to the Permian-Triassic boundary, reflects a rapid influx of isotopically light carbon into the atmosphere, while the influx of heavy metals and the presence of abundant tephra layers, including one just above the first appearance datum (FAD) of *Hindeodus parvus* and dated at $252.13 \pm 0.62$ Ma (Zuchuat et al., 2020), which is in good agreement with the tephra beds from the Induan GSSP section in Meishan (Burgess et al., 2014) that have been inferred to link the Siberian Traps LIP and the mass extinction in Svalbard (Gruszczynski et al., 1989; Grasby et al., 2015; Zuchuat et al., 2020). The reduced Iron (Fe)/Potassium (K) elemental ratio that accompanied the extinction horizon in the Vikinghøgda Formation seems to suggest that the tropical atmospheric circulation (Hadley Cell) could have expanded towards the poles, associated with an increased aridity in the hinterland of the basin (Zuchuat et al., 2020). Redox proxies, including lipid biomarkers (Summons et al., 2022), Fe and P speciation (Schobben et al., 2020), trace-metal data (Grasby et al., 2015; Uchman et al., 2016; Wignall et al., 2016; Zuchuat et al., 2020), and pyrite framboid sizes (Dustira et al., 2013; Wignall et al., 2016) also suggest that the mass extinction is associated with the expansion of oxygen minimum zones in the ocean, bringing anoxic and euxinic conditions into shallow-marine settings, as well as subsequent pulses of redox changes throughout the Early Triassic (Rodriguez-Tovar et al., 2021). Isotopic signatures of lipid biomarkers suggest frequent phytoplankton blooms, and phosphorus speciation data indicate an increase in nutrient supply and the remobilization of biologically available P as a consequence of the mass-extinction event initiating feedback that further developed anoxic conditions (Nabbfeld et al., 2010; Schobben et al., 2020).





Thermal stress and ocean acidification are also widely considered as key factors in the EPME, with global average temperature increases reaching 7°C (Kidder and Worsley, 2004; Svensen et al., 2009; Stordal et al., 2017; Burger et al., 2019), potentially as much as 9–12°C (Joachimski et al., 2012; 2020; Schobben et al., 2014; Chen et al., 2016). In Svalbard, there are currently no published geochemical investigations of the environmental changes associated with this hyperthermal event, but the presence of warm-water taxa such as red algae (Wignall et al., 1998), conodont genus *Clarkina* (Nakrem et al., 2008) and both

ostracod and radiolarian species that were equatorial during the Changhsingian (Foster et al., 2023) suggest that higher paleolatitudinal settings were unusually warm following the mass extinction. The cessation of carbonate rocks at the top of the Kapp Starostin and across the Boreal Realm have also provided an alternative hypothesis that ocean acidification developed and persisted for most of the Late Permian (Beauchamp and Grasby, 2009; Grasby et al., 2015). This hypothesis, however, requires the persistence of undersaturated conditions for millions of years, which is inconsistent with some Earth system

models that suggest that ocean acidification events cannot persist for this length of time (e.g., Hönisch et al., 2012). In addition, the lack of dissolution and repair marks on well-preserved mollusks from the extinction aftermath have also been interpreted to suggest that ocean acidification was not severe enough to have impacted skeletal calcification in the Boreal realm, at least at the onset of the Triassic (Foster et al., 2022). More research is, therefore, required to understand the role of thermal stress and ocean acidification in high-latitude marine extinctions.

The impact of EMPE on terrestrial ecosystems can also be investigated from Svalbard's marine successions. The Permian to Triassic palynological record of Svalbard and the Barents Shelf has been intensely investigated (e.g., Mangerud and Konieczny, 1993), in-part due to their utility for petroleum exploration (e.g., Vigran et al., 2014). A spore spike demise of gymnosperms, malformed spores and pollen, a drop in abundance of acritarchs, and compound specific isotopes of algal and land-plant-derived biomarkers all coincide with the mass extinction event, suggestingnear-synchroneity between effects in

marine and terrestrial realms (Stemmerik et al., 2001; Nabbfeld et al., 2010; Uchman et al., 2016). The presence of Permian plant taxa, including major Paleozoic plant groups in the Lower Triassic successions of Svalbard and the Barents Shelf, however, have led to some authors interpreting that the EMPE only had a minor impact on plant communities (Hochuli et al., 2010; Vigran et al., 2014). Aberrant pollen and spores reported from the Barents Sea and elsewhere have been suggested to be a consequence of severe atmospheric pollution and increased UVA-B radiation due to emissions from emplacement of the

Siberian Traps (Black et al., 2014; Hochuli et al., 2017; Galloway and Lindström, 2023 and references therein).

**4.3.2. Early Triassic ecosystem recovery**

The post-EPME survival and recovery of marine organisms recorded from Svalbard and the Boreal realm was unique. Within the early Griesbachian H. parvus conodont zone in central Spitsbergen (Lusitaniadalen and Deltadalen), a diverse assemblage of macro and microfossils have been recorded, including the only documented fully silicified marine assemblage of the Early

Triassic (Foster et al., 2017), the oldest record of post-extinction silica organisms globally (radiolarians and siliceous sponges; Foster et al., 2023), and the presence of an ecological complex assemblage of trace fossils (Nabbefeld et al., 2010; Rodríguez-Tovar et al., 2021). In addition, across Svalbard, the Lower Triassic succession preserved many groups, including bryozoans



(Nakrem and Mørk, 1991), algae (Wignall et al., 1998), conodonts (Nakrem et al., 2008), bivalves and gastropods (Buchan, 1965; Tozer and Parker 1968; Foster, 2015; Foster et al., 2017), ammonoids (see Nakrem et al., 2008), ostracods (Olempska and Błaszyk 1996), echinoderms (Salomon et al., 2015), and trace fossils (Wignall et al., 1998). Whilst sedimentation-rate calculations suggest marine ecosystems only required ca. 150 kyr to recover from the mass extinction (Rodríguez-Tovar et al., 2021), based on index conodonts, there is a distinctive pulse of environmental and ecological recovery in the Dienerian (Hatleberg and Clark, 1984; Wignall et al., 1998; Mørk and Worsley, 2006; Salamon et al., 2015).

The Lower Triassic succession of Svalbard is also fundamental for understanding the evolution and radiation of marine vertebrates following the Permian-Triassic transition. The Triassic succession of Svalbard has long been well-known for four described vertebrate fossil horizons (in stratigraphic order): The Fish Niveau, Grippia Niveau, Lower Saurian Niveau, and Upper Saurian Niveau (Wiman, 1910; Wiman, 1928). These bonebeds correspond to the Lusitaniadalen (the Fish Niveau) and Vendomdalen (the Grippa and Lower Saurian Niveau) members of the Vikinghøgda Formation that span the Smithian-Spathian transition (Lower Triassic; ca. 249.2 Ma), and the Ladinian-age (Middle Triassic; ca. 242–237 Ma) Blanknuten Member of the Botneheia Formation (the Upper Saurian Niveau; Maxwell and Kear, 2013; Hurum et al., 2018). Recent work on Early and Middle Triassic ecosystems in Svalbard reveals an exceptionally rapid diversification among marine vertebrates andichthyosaurs likely evolved prior to the EPME (Kear et al., 2023). The bonebeds reveal a much more complex foodweb than previously thought (Hurum et al., 2014; 2018; Bratvold, 2016; Delsett et al., 2017; Ekeheien, 2018; Engelschiøn et al., 2018; Økland et al., 2018; Roberts et al., 2022; Kear et al., 2023), suggesting that the Boreal Sea served as a climatic refuge after the EPME (Kear et al., 2023; Foster et al., 2023).

It has been hypothesized that the recovery from the EPME was delayed due to subsequent crises throughout the Early Triassic (Payne et al., 2004; Ware et al., 2011; Song et al., 2012; Foster et al., 2017b; Zuchuat et al., 2020; Wu et al., 2021), despite the degree of ichnofacies diversity and intensity reached pre-extinction level in ca. 150 Kyr (Rodríguez-Tovar et al., 2021). The first crisis occurring after the EPME is the "late Dienerian biotic crisis" (late Early Triassic; ca. 251 Ma), which is recognized by a negative CIE in the Vikinghøgda Formation in Deltadalen, central Spitsbergen, where it is sandwiched between two thin tephra layers. This benthic crisis was associated with dysoxic conditions in the water column and the seafloor (Zuchuat et al., 2020) and was first documented from the Tethyan with the Werfen Formation, northern Italy (Hofmann et al., 2015; Foster et al., 2017), as well as from subtropical latitudes along the Panthalassic margin (Ware et al., 2011; Hofmann et al., 2011; 2014). The palynological record from Greenland suggests that this CIE coincides with a more significant turnover in plant communities than at the end-Permian mass extinction (Hochuli et al., 2016). This indicates that the late Dienerian Crisis might have been of global significance. The presence of the two tephra horizons directly above and below the negative CIE of the late Dienerian biotic crisis suggest a volcanic trigger for the event, but the exact nature of this potential biotic crisis in Svalbard and globally requires additional research.

The most prevalent of the biotic crises in the Triassic is the Smithian-Spathian transition, which is associated with a negative CIE greater than at the EPME (Payne et al., 2004; Grasby et al., 2013; 2016b). In Svalbard, the Smithian-Spathian transition is recorded as a regression and subsequent transgression, with numerous fossiliferous horizons (e.g., Hoel and Orvin, 1937;





Buchan, 1965; Tozer, 1967; Weitschat and Lehmann, 1978; Mørk et al., 1999; Foster, 2015; Hammer et al., 2019; Hansen et al., 2024; Leu et al., 2024). While the Smithian-Spathian transition has received considerably less attention than the Permian-Triassic transition, recent research has highlighted the multi-factor nature of this event on the dynamic of the C and P cycles
in the Arctic (Hammer et al., 2019; Blattmann et al., 2024). Nevertheless, many outstanding questions remain unanswered both globally and on Svalbard: What was the timing of the event? What caused the CIE? What environmental changes are associated with the CIE? How were terrestrial and marine ecosystems impacted by the event?

At the Festningen section, Hg concentrations and Hg/TOC both show a noticeable spike in the Tvillingodden Formation of the Sassendalen Group, which have been associated with Hg loading associated with increased activity from the Siberian Traps
(Grasby et al., 2016). This peak has also been correlated with Hg loading recorded at the Smithian-Spathian boundary at Smith Creek, Arctic Canada, even though the Smithian-Spathian boundary is associated with a carbon isotope peak at most sections globally. In the equivalent Vikinghøgda Formation at the Wallenbergfjellet section, however, the Smithian-Spathian boundary (the top of the Wasatchites tardus ammonoid zone) is associated with a positive CIE (Galfetti et al., 2007; Hammer et al., 2019) and the Hg loading occurred prior to the Smithian-Spathian boundary in the middle to late Smithian (Hammer et al.,
2019). This is consistent with the Festningen section, which suggests the Hg loading occurred synchronously during the Smithian (Grasby et al., 2016a). At the Kongressfjellet and Vikinghøgda sections, just above the base of the Vendomdalen Member of the Vikinghøgda Formation, a significant shift in the palynological record is recorded (Mørk et al., 1999; Galfetti et al., 2007). This Smithian-Spathian turnover is also recorded in the shallow cores drilled at the Svalis Dome, central Barents Sea (Hochuli and Vigran, 2010). This turnover in the palynological record has been associated with the re-establishment of
diverse woody gymnosperm ecosystems, marking a recovery signal from both the Permian-Triassic climate crisis and a "late Smithian Thermal Maximum". This is supported by the synchronous recovery in the latitudinal diversity gradient of ammonoids (Brayard et al., 2009) and the recovery of equatorial benthic marine communities (Twitchett and Wignall, 1996; Chen et al., 2011; Pietsch and Bottjer, 2014; Hofmann et al., 2014; 2015; Foster et al., 2015; 2017; 2018; 2023a). The Smithian-Spathian transition, therefore, appears to be marked by a late Smithian Thermal Maximum and a subsequent Smithian-Spathian
boundary cooling and associated biotic recovery in Svalbard, and pan-Arctic.

### 4.3.3 Middle to Late Triassic: organic-rich mudstones rich in phosphate

The Middle Triassic succession is dominated by organic-rich mudstone to siltstones of the Anisian-Ladinian (Middle Triassic; ca. 247–237 Ma) Botneheia Formation of the upper Sassendalen Group (Mørk and Bjorøy, 1984; Krajewski, 2008, 2013;
Wesenlund et al., 2021). The upper Blanknuten Member of the Botneheia Formation (Ladinian) reaches 12 wt.% TOC, and certain stratigraphic intervals, particularly in the underlying Muen Member (Anisian), contain abundant phosphorite nodules (e.g., Krajewski, 2013; Wesenlund et al., 2021; 2022; Engelschiøn et al., 2023a). In the Anisian, nutrient saturated runoff from continental areas, particularly associated with the approaching delta system from the southeast (see next section), coupled with upwelling of nutrient-rich waters from the Panthalassic Ocean, resulted in extensive algal blooms and the formation of oxygen



minimum zones, which promoted dysoxia and anoxia and preservation of organic matter and precipitation of phosphate (Krajewski, 2013; Vigran et al., 2014; Wesenlund et al., 2022; Engelschiøn et al., 2023a). Repeated transgression-regression events, influenced by the emerging delta system, likely contributed to fossil-preservation potential of the Middle Triassic strata as the relatively shallow offshore environment was temporarily punctuated by anoxic events (Mørk et al., 1989; Krajewski, 2013; Engelschiøn et al., 2023a). Fossil preservation has also occurred by complete barium sulfate (barite) pseudomorphing,

possibly by sulfate remobilization from the organic-rich shales (Engelschiøn et al., 2023b). Moreover, the high TOC values in the Ladinian Blanknuten Member suggest deposition under euxinic conditions, possibly governed by restricted water circulation due to shallowing of the basin, as well as water mass stratification caused by the increasing influx of riverine waters into the marine basin (Wesenlund et al., 2022).

**4.3.4. Late Triassic: the Carnian Pluvial Episode and the world's largest delta plain**

The Carnian Pluvial Episode (CPE, ca. 233 Ma) marks a period of climate and biotic changes spanning the Julian 2 and Tuvalian 1 substages of the Carnian (Simms and Ruffell, 1989; Breda et al., 2009; Dal Corso et al., 2018). The CPE was first recognized as concomitant to global carbonate platform environments perturbations associated with an increased terrigenous input into sedimentary basins (Dal Corso et al., 2018) and the rapid diversification of dinosaurs (Benton et al., 2018). The CPE

is also associated with several isotope perturbations in terrestrial and marine C and Hg records, which potentially reflects multiple pulses of volcanic activity of the Wrangellia LIP (Dal Corso et al., 2018; Jin et al., 2023). Analyses of palynological assemblages, chemical weathering indices, $\delta^{18}O$ apatite records, and redox-sensitive elements suggest that the CPE was characterized by an extremely humid climate (e.g., Roghi, 2004; Baranyi et al., 2019), warm temperatures (Rigo and Joachimski, 2010; Rigo et al., 2012; Sun et al., 2016) and widespread marine dys- and anoxia (Soua, 2014; Sun et al., 2016;

Tomimatsu et al., 2021). Shallow-water carbonate production switched as a result of climatic variations and eustatic sea-level fall (Jin et al., 2020), which impacted both the marine and continental biosphere, with high extinction rates in ammonoids and conodonts (Rigo et al., 2007; Dal Corso et al., 2022), rapid extinction of terrestrial tetrapods and a subsequent diversification of dinosaurs (Bernardi et al., 2018), mammals (Benton et al., 2018), scleractinian corals (Stanley, 2003), calcareous dinoflagellates and plants (e.g., Dal Corso et al., 2022).

While the CPE is well-documented in the Tethyan Realm (Dal Corso et al., 2018), evidence from the Boreal Realm remains limited. In Svalbard, preliminary palynological evidence from the Kapp Toscana Group in central Spitsbergen integrated with organic carbon isotope and paleomagnetic constraints indicate warming during the late Julian-1 (Mueller et al., 2016; Paterson et al., 2016) and suggest wetter conditions starting from the Julian-2, which occurs in the lower part of the De Geerdalen Formation (Mueller et al., 2016). In addition, the detailed study of paleosols in the De Geerdalen Formation above CPE seems

to indicate the transition from humid (coal) to warm arid (caliche) climate settings (Lord et al., 2022). Nevertheless, the precise location of the CPE in Svalbard's stratigraphy remains uncertain, and increased research can help understand the exact triggering mechanism of these climate perturbations.





### 4.4.2. Jurassic-Cretaceous: A greenhouse with cold snaps

The Jurassic and Cretaceous saw some of the warmest background global temperatures of the Phanerozoic (Jenkyns et al.,
2012). Permanent polar ice caps were not present, although evidence for episodic cooling and the growth and decay of small,
ephemeral polar ice caps has been presented (e.g., Price, 1999; Miller et al., 2005; Grasby et al., 2017; Alley et al., 2020).
Oxygen isotope records of Jurassic and Early Cretaceous from Svalbard have been derived largely from belemnites (e.g.,
Ditchfield, 1997; Price and Nunn, 2010; Hammer et al., 2011) and an extensive dataset has been derived from Kong Karls
Land (Ditchfield, 1997). The usefulness of these data, however, is hampered by modest biostratigraphic age constraints and
pervasive diagenesis. Nonetheless, some well-preserved belemnites were identified, and oxygen isotope data from Hammer et
al. (2011) from the upper Agardhfjellet Formation (Volgian–Ryzanian, which approximately correlates with late Tithonian to
early Berriasian, Late Jurassic-Early Cretaceous; ca. 149-140 Ma) and Price and Nunn (2010) from the Hauterivian (Early
Cretaceous; ca. 133-126 Ma) part of the Rurikfjellet Formation of Festningen (see Jelby et al., 2020b for a discussion of the
age constraints) show $\delta^{18}O$ values ranging from -3.0 to 0.8 ‰ VPDB. The belemnite data from Kong Karls Land (spanning
the Aalenian to Valanginian; Middle Jurassic to Early Cretaceous; ca. 175–132 Ma) give $\delta^{18}O$ values of -0.7 to 1.2 ‰ VPDB.
These data argue for cooling and warming episodes at these high paleolatitudes and may indicate high seasonality and/or high
frequency climatic variability, although the absolute temperatures they represent are hard to interpret. Common practice would
be to assume that the seawater $\delta^{18}O$ was -1 ‰ SMOW (i.e., that of an ice-free world), and use either the equation for molluscan
calcite (Anderson and Arthur, 1983) or experimentally derived synthetic calcite (Kim and O'Neil, 1997) to calculate the
precipitation temperature of the calcite. However, Price and Nunn (2010) used the presence of glendonites in certain horizons
to independently assess paleotemperature of those intervals, and backcalculated the $\delta^{18}O_{sw}$. This suggested that the seawater
$\delta^{18}O$ was much lower than the global average (i.e., heavily meteorically influenced), which is supported by the depositional
setting (ranging from marine to terrestrial delta plain). Furthermore, recent work using clumped isotope thermometry on
belemnites suggests that the equations of Kim and O'Neil (1997) or Anderson and Arthur (1983) are not suitable for belemnite
calcite, and rather, the temperature equations of Kele et al. (2015) or Daëron et al. (2019) should be used (Price and Passey,
2013; Wierzbowski et al., 2018; Bajnai et al., 2020; Price et al., 2020; Vickers et al., 2019b; 2020; 2021). Using the latterresults
in warmer estimated temperatures than previously thought, even if one assumes a meteorically influenced $\delta^{18}O_{sw}$ of -2.5 ‰
SMOW. For example, the belemnite data from Kong Karls Land suggests temperatures closer to 12-20 °C rather than the 8 to
13°C originally postulated by Ditchfield (1997). The belemnites from Festningen may indicate a temperature range of around
13–30 °C rather than the 9–25 °C originally suggested by Price and Nunn (2010) and Hammer et al. (2011).

### 4.4.2.1. Jurassic–Cretaceous boundary and the Volgian Isotopic Carbon Excursion (VOICE)

The global carbon-isotope ($\delta^{13}C$) signal of the Upper Jurassic and Jurassic-Cretaceous (J-K, ca. 145 Ma) boundary as
recognized in Tethyan, Atlantic, and Pacific sections is generally characterized by a steady decline. This decline has been
attributed to progressive deceleration of the carbon cycle due to the development of more oligotrophic oceanic conditions and



reduced marine primary production (Weissert and Channell, 1989; Weissert et al., 1998; Weissert and Erba, 2004; Tremolada et al., 2006; Price et al., 2016). In Svalbard, however, the upper Kimmeridgian-middle Volgian succession displays a prominent negative CIE termed the "Volgian Isotopic Carbon Excursion" or "VOICE" by Hammer et al. (2012). The VOICE is of much greater magnitude than the entirety of the long-lived decline of the lower-latitude records (Fig. 11; Morgans-Bell et al., 2001;

Price and Rogov, 2009; Žák et al., 2011; Hammer et al., 2012; Zakharov et al., 2014; Koevoets et al., 2016; 2018; Galloway et al., 2020; Jelby et al., 2020b). This recently recognized carbon-isotope marker of Boreal sections (observed across Svalbard, northern Siberia, Arctic Canada, the Russian Platform and possibly southern UK; Galloway et al., 2020; Jelby et al., 2020b and references therein) and newly discovered occurrences in the Neuquén Basin of Argentina (Capelli et al., 2022; Weger et al., 2022) and possibly in the eastern Tethys (Fallatah et al., 2024), is characterized by a relatively abrupt negative excursion

($\leq 6.4$‰ in Svalbard) in $\delta^{13}C_{org}$ values. This excursion is followed by a positive trend in $\delta^{13}C_{org}$ values through the upper Volgian-Ryazanian and across the J-K boundary (Price and Rogov, 2009; Hammer et al., 2012; Dzyuba et al., 2013; Zakharov et al., 2014; Koevoets et al., 2016; Galloway et al., 2020; Jelby et al., 2020b; Vickers et al., 2023).

In Svalbard, the VOICE is documented in both drill core (DH-2, central Spitsbergen) and outcrop sections (e.g., Myklegardfjellet, eastern coast of Spitsbergen; Festningen, western Spitsbergen), and falls within the paper-shale-dominated

Oppdalssåta and Slottsmøya members of the Agardhfjellet Formation (Koevoets et al., 2016; 2019; Jelby et al., 2020b; Vickers et al., 2023). The VOICE and positive recovery across the J-K boundary were originally considered unique to the Boreal realm, as a result of decoupling from the global carbon reservoir during the Late Jurassic. Galloway et al. (2020) and Jelby et al. (2020b) attributed this Boreally-limited CIE to restricted oceanographic connectivity between the shallow epeiric seas of the high northern latitudes and open oceans, associated with water-mass stratification and increased continental runoff (Park et al.,

2024) due to an eustatic sea-level lowstand. As a result, basinal depletion of $^{13}C$ resulted from oxidation of terrestrial organic matter, or input of isotopically light $CO_2$ by respiration of marine organisms, and/or riverine dissolved inorganic carbon (DIC) (Patterson and Walter, 1994; Holmden et al., 1998). However, the recent discovery of the VOICE in the southern hemisphere (Rodriguez Blanco et al., 2022; Capelli et al., 2022; Weger et al., 2022), albeit with diachroneity seen also in the Arctic (e.g., Rogov, 2021; Rogov et al., 2023), that may reflect poor age control or a latitudinal climate gradient, indicates that the excursion

is not limited to high northern latitudes. Weger et al. (2022) suggested that the VOICE was driven by changes in the input of terrestrially-derived organic matter, controlled by relative sea-level change and climate, although this hypothesis is refuted by Fallatah et al., (2024) and Galloway et al. (2024) because the VOICE may be present in a restricted setting in the Tethys (Fallatah et al., 2024) and because there is a lack of consistent changes in organic matter type (as indicted by OI and HI), indices of weather and grain size across the VOICE in the Canadian and Festningen material (Galloway et al., 2024).


Collectively, the VOICE is ascribed variously to restricted circulation (Galloway et al., 2020; Jelby et al., 2020; Śliwińska, et al., 2020, Fallatah et al., 2024), precipitation of authigenic carbonate in reducing conditions (Rodriguez Blanco et al., 2022), or variations in the input of terrestrially derived organic matter that is in turn a manifestation of relative sea level change and climate (Capelli et al., 2021; Weger et al., 2022). However, none of these models alone can explain the contemporaneous





enrichment of Ag in Arctic basins (Galloway et al., 2024) and alternatives must be considered. An interval of silver (Ag) enrichment occurs across the VOICE in the Sverdrup Basin (Canadian Arctic, 3-6x higher than average shale) and at Festningen (6x higher than average shale). Silver is similarly enriched in black shales of Jurassic and Cretaceous age in the Barents Sea, Norwegian Shelf, and West Siberia Basin (Lipinski et al., 2003; Zanin et al., 2016). The relationship of Ag to organic matter, S, Fe, and redox-sensitive trace elements in the strata from Canada and Festningen suggest that an extra-basinal

source of Ag to seawater during the Volgian existed and that the source was enhanced hydrothermal flux in the proto-Amerasia basin during rift climax, with sufficient circulation to transport high Ag-seawater to surrounding shelves (Svalbard) and within and to extensional basins (Sverdrup Basin). Europium (Eu) values show an anomaly (Eu/Eu*>1) during the VOICE both in the Canadian strata and strata at Festningen, suggesting the presence of hydrothermal fluids. It is possible that the negative carbon isotopic signature of the VOICE is, in part, associated with the putative hydrothermal systems hypothesized to have

caused Ag enrichment. Nearly all negative carbon isotope excursions in the geological record, but so far not the VOICE, are interpreted to reflect episodes of massive carbon release. Its global manifestation could therefore be the result of widespread rifting and outgassing of large quantities of $CO_2$ (see Brune et al., 2017) associated with the breakup of Pangea (Galloway et al., 2024).

**4.4.2.2. Valanginian Weissert Event**

The Weissert Event is a prominent global carbon-cycle perturbation which occurred in the Early Cretaceous (ca. 133 Ma). It is expressed in carbon isotope records (Lini et al., 1992; Price et al., 2016) and is manifested in Arctic Canada (Galloway et al., 2020) and Svalbard (Jelby et al., 2020b; Vickers et al., 2023) (Fig. 11). The isotope event consists of a globally recognized positive CIE of a significant magnitude (2–5 ‰), which is widely documented in marine carbonates, fossil shell material,

terrestrial plants, and organic matter (e.g., Lini et al., 1992; Gröcke et al., 2005; Aguirre-Urreta et al., 2008; Price et al., 2016; Galloway et al., 2020; Jelby et al., 2020b; Vickers et al., 2023; Fallatah et al., 2024). Regional to global climate cooling in the Valanginian (Early Cretaceous; ca. 140–133 Ma) is well documented (e.g. Pucéat et al., 2003; Weissert and Erba, 2004; McArthur et al., 2007; Bodin et al., 2015; Meissner et al., 2015; Grasby et al., 2017b), although the timing, magnitude and extent of this cooling is debated (e.g., van de Schootbrugge et al., 2000; McArthur et al., 2007), as is the mechanism for the

CIE. Paleoclimatic reconstructions using biomarkers indicate a ~3°C global surface cooling across the event (Cavalheiro et al., 2021), and glacial deposits from the Eromanga Basin (Australia) suggest the transient development of a small southern polar ice-cap (Alley et al., 2020), whereas stable oxygen isotope records show mixed signals, with rising $\delta^{18}O_{belemnite}$ values suggesting cooling in the Boreal Realm from the Late Valanginian to Early Hauterivian (Podlaha et al., 1998; McArthur et al., 2004; Price and Mutterlose, 2004; Bodin et al., 2015; Meissner et al., 2015), with little change in Tethyan records (e.g., van

de Schootbrugge et al., 2000; McArthur et al., 2007).



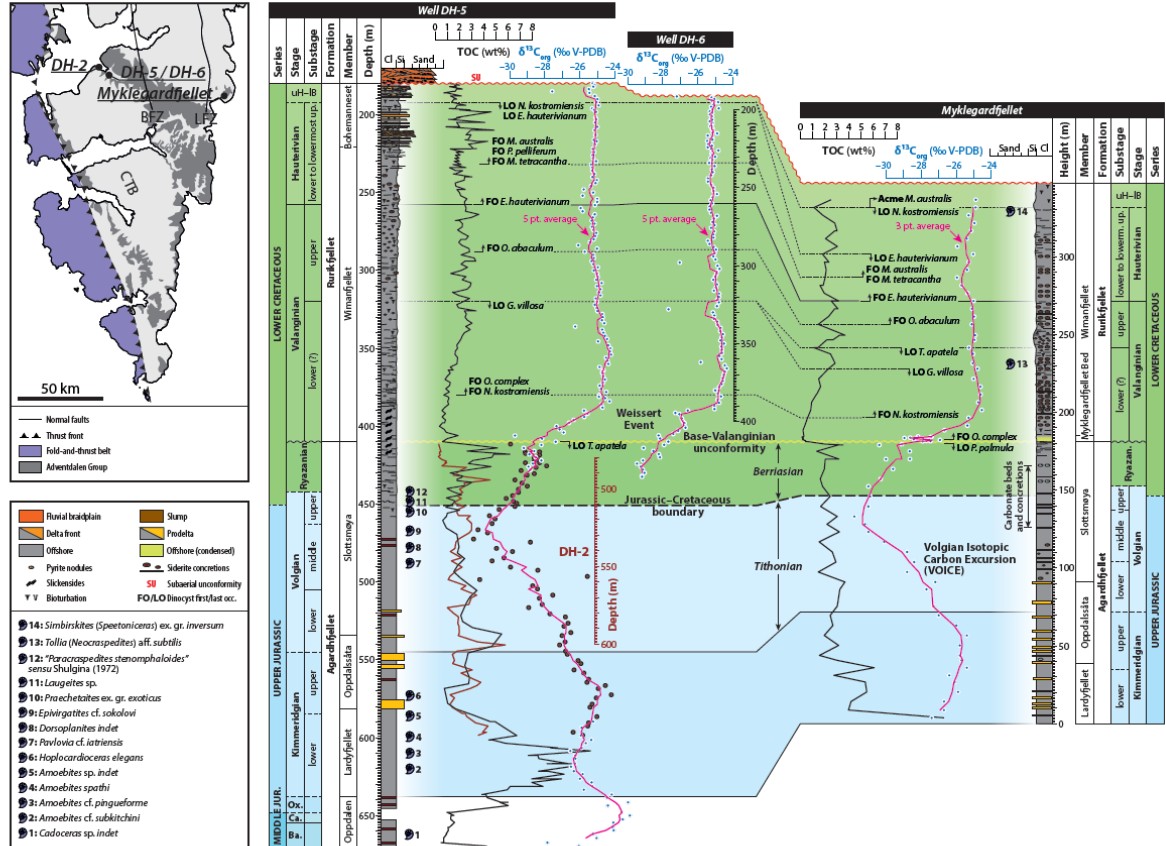

**Figure 10.** Correlation of organic stable carbon-isotope (δ¹³C_org) stratigraphy between the cored wells DH-2, DH-5 and DH-6 and the Myklegardfjellet outcrop section, calibrated by total organic carbon (TOC) trends, and dinocyst and ammonite biostratigraphy (compiled and modified from Jelby et al. 2020b; 2024, and Koevoets et al. 2016, 2018). The boundary between the Agardhfjellet and Rurikfjellet formations (conforming to a base-Valanginian unconformity and demarcated by a marked drop in TOC values) is used as a correlation datum. Note the clear expression of the Weissert Event and 'Volgian Isotopic Carbon Excursion' (VOICE) in the different sections. uH–lB, upper Hauterivian–lower Barremian. For details on the dinocyst bio-events, the reader is referred to Jelby et al. (2020b; 2024) and Śliwińska et al. (2020).

In the Arctic (including Svalbard), the widespread occurrence of glendonites around the Weissert Event interval (from the Berriasian to Hauterivian; Grasby et al., 2017b; Vickers et al., 2019a; Galloway et al., 2020; Jelby et al., 2025) has led to speculation of the cooling being decoupled from the CIE (e.g., Rogov et al., 2017), although this may partially be an artifact of age uncertainties for the Arctic successions (e.g. Vickers et al., 2019a; Jelby et al., 2020b; 2025). It may also suggest that episodic cooling punctuated the background warmth throughout the Early Cretaceous. The emplacement of the Paraná-Etendeka LIP is broadly coincident with the Weissert Event CIE (Fig.11), although uncertainties surrounding the exact relative timings have led to much debate around the causal mechanism for the event (e.g., Weissert and Erba, 2004; Duchamp-Alphonse et al., 2007; Dodd et al., 2015; Rocha et al., 2020; Gomes and Vasconcelos, 2021). Recently, it has been shown that the peak



of Paraná-Etendeka activity coincided with the onset of the Weissert Event (Martinez et al., 2023), supporting a volcanic trigger for this CIE.

In Svalbard, the $\delta^{13}C_{org}$ record reveals an abrupt and pronounced positive excursion of up to 5.5‰ in the lower Valanginian, coincident with the base of the Rurikfjellet Formation (Fig. 11; Jelby et al., 2020b; 2025). The excursion is clearly observed in both core (DH-5 and DH-6 in Adventdalen) and outcrop (Myklegardfjellet) sections, where $\delta^{13}C_{org}$ reaches the most positive values recorded since the Callovian–Oxfordian (Koevoets et al., 2016). Glendonites are found in numerous horizons in the Festningen locality, although a patchy $\delta^{13}C_{org}$ curve and poor stratigraphic dating of this section due to local small-scale tectonism has led to uncertainty as to the relative age of the glendonites with respect to the Weissert Event (Vickers et al., 2019a; 2023; Jelby et al., 2020b; 2025). Glendonites in the Canadian successions occur in Valanginian of the Deer Bay Formation (Grasby et al., 2017b; Galloway et al., 2020).

Examination of the Hg record across this interval at both Festningen and correlative sites in the Sverdrup Basin similarly yielded uncertain results, due to the poor age constraints preventing convincing identification of the Weissert Event CIE (Vickers et al., 2023). Nonetheless, Arctic Hg/TOC ratios are observed to increase across the proposed Weissert Event intervals in both Svalbard and the Sverdrup Basin. This supports recent work indicating that Paraná-Etendeka volcanism was synchronous with the onset of the Weissert Event CIE (Gomes and Vasconcelos, 2021). In other localities on Svalbard (DH-5 borehole in Adventdalen and Myklegardfjellet in eastern Spitsbergen, Fig. 10), the Weissert Event is well dated with age-diagnostic ammonites and palynomorphs (Jelby et al., 2020b; 2025), and with the observed $\delta^{13}C_{org}$ trend being consistent with the globally recognized Weissert Event (Fig. 11; Lini et al., 1992; Weissert et al., 1998; Weissert and Erba, 2004). The onset of the Weissert Event in Svalbard appears to have occurred earlier than in other Boreal sites, but this diachroneity may reflect a depositional hiatus and/or stratal condensation in the glauconitic plastic clay of the Myklegardfjellet Bed at the base of the Rurikfjellet Formation, corresponding to the Tethyan T. pertransiens – N. neocomiensiformis ammonite zones (Jelby et al., 2020b; 2025).



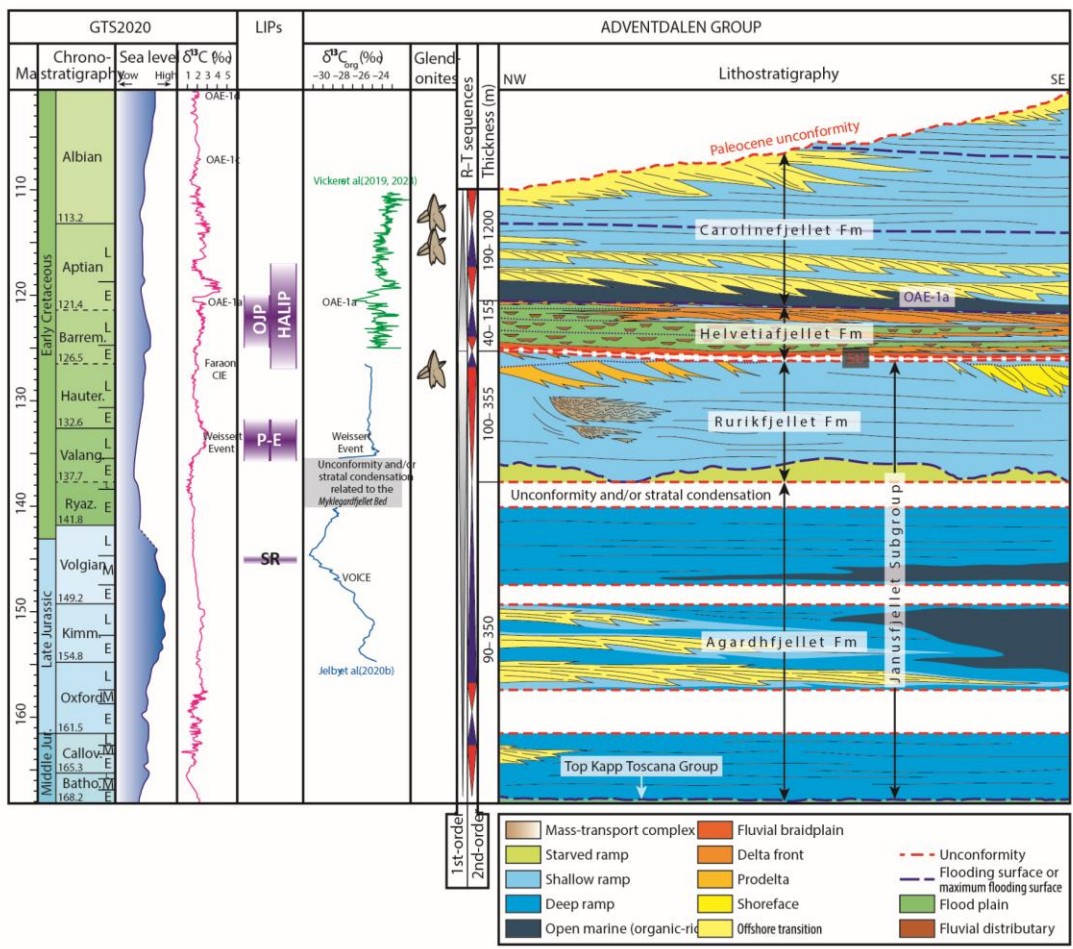

**Figure 11. Overview of the Upper Jurassic and Lower Cretaceous stratigraphy of Svalbard plotted against time, from Jelby et al. (2024), based on Grundvåg et al. (2017; 2019). To the left of the stratigraphic summary chart, thicknesses of the formations, first- and second-order regressive-transgressive sequences, occurrence of glendonites (Vickers et al., 2019), LIP volcanic episodes (see references in Vickers et al., 2023), and organic stable carbon isotopes (see Jelby et al., 2024 and references therein) are shown. These are plotted against global δ$^{13}$C trends, sea level, and ages from the Geological Time Scale 2020 (Gale et al., 2020; Hesselbo et al., 2020).**

In summary, the Weissert Event represents an important recoupling of the carbon cycle between Boreal (including Svalbard) and lower-latitude basins (including Tethyan, Atlantic, and Pacific), probably because of re-established ocean connections in response to a global eustatic sea-level rise and changing oceanic gateway connections (Haq, 2014; Galloway et al., 2020; Jelby et al., 2020b). However, Jelby et al. (2020b) recognized that the decay of the event in Svalbard spanned the late Valanginian to early Barremian (Early Cretaceous; ca. 125.77–121.4 Ma) but is negligible compared to most other Boreal and lower-latitude records, and that the signal remains relatively stable at near-peak values following the positive excursion. This indicates that



the oceanographic reconnection between higher and lower latitudes, and thus ocean ventilation, must have been sufficiently limited to keep some Boreal basins (including Svalbard) relatively deviated from prevailing global carbon-cycle dynamics

following the isotopic event due to water exchange only through narrow, shallow straits (Price and Mutterlose, 2004).

### 4.4.2.3. The Cretaceous HALIP and OAE1a

The largest carbon cycle perturbation of the Early Cretaceous occurred in the early Aptian (a. 121–113 Ma). It is associated with Ocean Anoxic Event 1a (OAE1a, ~120 Ma), when black shale deposition occurred across multiple marine sites, indicative of widespread ocean anoxia (e.g., Jenkyns, 1980). OAE1a is characterized by a globally-recognized sharp negative CIE

followed by a twin-peaked positive "recovery" CIE (Fig. 11; Jenkyns, 1995; Menegatti et al., 1998; Ando et al., 2002; Price, 2003; Weissert and Erba, 2004; Herrle et al., 2015; Dummann et al., 2021), which has also been recognized in multiple localities across Svalbard and in the Canadian Arctic (Herrle et al., 2015; Midtkandal et al., 2016; Vickers et al., 2016; 2019a; 2023; Grundvåg et al., 2019; Dummann et al., 2021). Significant perturbations in global climate are believed to have occurred along with the CIE, with global warming followed by cooling being evident from multiple proxies in sites from across the

globe (e.g., Bottini et al., 2015; Bodin et al., 2015; Harper et al., 2021; Galloway et al., 2022).

In Svalbard and other Arctic localities, the occurrence of numerous glendonite horizons immediately after the CIE (inAptian - Albian strata) supports the global extent of the post-OAE1a cooling (Fig. 11; Schröder-Adams et al., 2014; Herrle et al., 2015; Grasby et al., 2017b; Rogov et al., 2017; Vickers et al., 2019a). The OAE1a is believed to be linked to LIP volcanism, as both the Greater Ontong-Java Plateau (OJP) in the Pacific Ocean and the HALIP were being emplaced approximately synchronously

with the onset of the event (Fig.11; Midtkandal et al., 2016; Percival et al., 2021; Galloway et al., 2022). However, uncertainties surrounding the exact relative timings of the volcanism and the OAE1a complicate the interpretation of the cause of the event (Tarduno et al., 1991; Mahoney et al., 1993; Parkinson et al., 2002; Tejada et al., 2002; 2009; Erba et al., 2004; 2015; Chambers et al., 2004; Thordarson, 2004; Dockmann et al., 2018; Kasbohm et al., 2021; Galloway et al., 2022).

Attempts to use Hg as a proxy for volcanism to resolve the question of relative timing of HALIP and OJP with regards to the

OAE1a CIE and accompanying climatic/environmental perturbations have yielded ambiguous results (Percival et al., 2021; Vickers et al., 2023), and recent work using palynology suggests HALIP-related landscape disturbances began to occur in the latest Barremian, coincident with the first pulse of the HALIP, but prior to the early Aptian onset of OAE1a (Galloway et al., 2022). The onset of the negative $\delta^{13}$C excursion across Svalbard occurs within a sapropel-rich interval which is highly impoverished in marine palynomorphs (dinocysts), but which yield a number of reworked taxa (e.g., in the Ullaberget outcrop

section, the DH2, and the DH1 cores near Longyearbyen (Midtkandal et al., 2016; Grundvåg et al., 2019; Śliwińska et al., 2020), narrowing the age to the Barremian-Aptian transitional interval. The regional flooding event that is considered to correlate globally with the OAE1a yields well preserved dinocysts and has been assigned an earliest Aptian age (Midtkandal et al.,2016; Grundvåg et al., 2017; 2019). Younger Cretaceous strata that elsewhere record other OAEs (e.g., OAE1b, OAE 2, OAE3) are absent from Svalbard (Fig. 4).



## 4.5. Cenozoic

The Paleogene succession in Svalbard includes Paleocene to possibly mid-Oligocene age strata (61.8 to ~30 Ma), and thus was deposited during the global climate transition from greenhouse to coolhouse conditions (Zachos et al., 2001; Westerhold et al., 2020). Since the Late Cretaceous, Svalbard was already located north of the Arctic Circle (e.g., Harland, 1997) at paleolatitudes comparable to the present (Fig. 3). Therefore, the Cenozoic geological record in Spitsbergen provides a natural reference interval for future polar amplification to global warming in a warm Arctic. The deposits of the CSB belong to the Van Mijenfjorden Group with thicknesses up to 2200 m. The basin fill begins with the Paleocene coal-bearing successions that demonstrates a transgressive trend from delta plain to prodelta/outer shelf facies, followed by marine mudstones and intensely bioturbated sandstones that mainly were sourced from north and northeast of Svalbard (Petersen et al., 2016; Lüthje et al., 2020; Jochmann et al., 2020). The first evidence of WSFTB-derived sediments occurs in the latest Paleocene with a westerly-derived clastic wedge known as the Hollendardalen Formation. The upper part of the CSB continued to fill into the Eocene and possibly the Oligocene and consists of >800 m thick deposits of shelf and shelf-edge deltas, slope clinotherms, and basin floor fans sourced from the WSFTB to the west (Steel et al., 1981; Johannessen and Steel, 2005; Helland-Hansen and Grundvåg, 2021). Maximum subsidence occurred during the deposition of the Frysjaodden Formation, which contains a significantly expanded (>30 m) Paleocene-Eocene Thermal Maximum (PETM) sequence (Cui, 2010; Charles et al., 2011; Dypvik et al., 2011). The shallowing-upward uppermost CSB fill is seen as a transition from deepwater marine via shallow marine/delta front to coastal plain and continental strata. The Norwegian-Greenland Seaway became severely restricted at this time, isolating the Arctic from the Atlantic Ocean during the PETM and early Eocene (Blakey, 2021; Hovikoski et al., 2021; Jones et al., 2023). Overall, the Paleogene coal-bearing successions developed in several stratigraphic levels of Svalbard are an excellent archive for reconstruction of the past vegetation and climate. These intervals provide an excellent insight into the fauna/vegetation of Svalbard at these times (see section c below and 6.3).

### 4.5.1. Paleocene hothouse from an Arctic Circle perspective

The Paleocene was characterized by hothouse climate (Zachos et al., 2001; Westerholt et al., 2020). In Svalbard, the CSB Paleocene succession consists of the terrestrial to nearshore Firkanten Formation (yielding fossil fauna) and the offshore marine Basilika, Grumantbyen and lowermost Frysjaodden formations. The Firkanten Formation of early Paleocene age (Selandian) contains fossil plant-bearing units also known as the Barentsburg flora (see Golovneva et al. 2023). These deposits have been extensively investigated due to economic interest and exploitation of coal resources over the last century, yet there are relatively few publications describing the succession in detail (cf. Steel et al., 1981; Nøttvedt, 1985; Nagy, 2005; Jochmann et al., 2020; Lüthje et al., 2020). Coal exploration over the past few decades has provided more than 500 drill cores through the early Paleogene of Svalbard and focuses in particular on the coal-bearing Todalen Member of the Firkanten Formation. Fission-track ages from the Firkanten Formation dated the unit to 63 ± 2 Ma and 64 ± 2 Ma (Blythe and Kleinspehn, 1998). A more





precise age is derived from an ash layer that cross-cuts the lowermost coal seam in the Firkanten Formation. It has been dated using U-Pb methods to 61.596 ± 0.028 Ma Ma (Jones et al., 2017) at the Danian–Selandian boundary, constraining the main coal deposition to the early Selandian. The Paleocene coals have recently been shown to comprise much higher resolution

stratigraphic records than previously anticipated (Large and Marshall, 2015; Large et al., 2021), especially when coals are formed from peat growing in colder climates. The age model developed by Large and Marshall (2015) and an improved understanding of long-term storage of peatland carbon (Large et al., 2021) can be implemented to assess accumulation rates. For example, the 1.5 m-thick Longyear coal seam has a modeled accumulation rate of 59 kyr and 99 kyr for temperate and boreal climates, respectively, which suggests a high-resolution record that can be used to infer variation in atmospheric dust

input and the effects of forest fires (Marshall, 2013). Material has been collected from the now closed coal mines in ongoing research projects to expand on these existing datasets. The Selandian coal seams provide details of the vegetation on Svalbard at that time. The paleoflora from Svalbard reveal a temperate (mean annual temperature 10.1 ±2°C), maritime, humid climate, with warm summers and cool mild winters (e.g., Ulf et al. 2007; Golovneva et al. 2023). Temperate Arctic paleotemperatures are corroborated by Pantodont tracks discovered at the upper boundary of the coal layer of the Firkanten Formation (Lüthje et

al. 2010). The climatic conditions were similar in the larger Arctic region (e.g., O'Regan et al., 2011).

### 4.5.2. The Paleocene-Eocene Thermal Maximum (PETM)

The PETM was a transient period (~150-200 kyr) of rapid global warming that began around 56 Ma, superimposed on already greenhouse conditions of the early Paleogene (Zachos et al., 2001). The event is recognized as a global negative CIE in sections

worldwide, attributed to a massive release of $^{13}$C-depleted carbon to the ocean-atmosphere system (McInerney and Wing, 2011). Potential sources include surface reservoirs such as dissociation of methane hydrates from marine sediments (e.g., Dickens et al., 1995) and/or volcanic and thermogenic degassing from the emplacement of the NAIP (e.g., Svensen et al., 2004; Gutjahr et al., 2017; Berndt et al., 2023; Jones et al., 2023). Existing High Arctic records for the PETM and following hyperthermal events suggest subtropical temperatures in both marine and terrestrial realms (e.g., Suc et al., 2020; Sluijs et al.,

2000). In Svalbard, the PETM and its various environmental effects has been thoroughly documented in several chemostratigraphic to biostratigraphic multi-proxy studies (Charles et al., 2011; Dypvik et al., 2011; Harding et al., 2011; Nagy et al., 2013; Wieczorek et al., 2013; Jones et al., 2019; Cui et al., 2021; Pogge von Strandmann et al., 2021). The bulk of these studies are focused on a SNSK drill core (BH09/05) from a coal-exploration well located on the eastern flank of the CSB (Fig. 2).

The timing and duration of the PETM is well constrained from Svalbard strata with a high-precision U-Pb radiometric age of an ash layer within the CIE outcropping near Longyearbyen, couple with evidence of orbital cycles within the strata of the BH09/05 core (Charles et al., 2011). Other studies explore the myriad consequences of extreme warming in Svalbard strata. The abundance of kaolinite in the PETM interval (Dypvik et al., 2011) and a large negative lithium isotope excursion coincident with the CIE (Pogge von Strandmann et al., 2021) suggests increased weathering rates in response to warmer and wetter



conditions. Increased runoff rates resulted in a stratified water column with a freshwater surface layer and oxygen depleted bottom waters, which severely reduced the diversity of various fauna elements in the basin (Harding et al., 2011; Dypvik et al., 2011; Nagy et al., 2013). Osmium isotopes (Wieczorek et al., 2013) and mercury anomalies (Jones et al., 2019) show key variations in the activity of the NAIP at this time, supporting the hypothesis that NAIP volcanism and magmatism was at least partially responsible for the extreme carbon emissions at the PETM onset. While studies on the BH09/05 core are now plentiful,

there are numerous other cores and outcrops that contain the PETM strata that have received little to no attention.

### 4.5.3. The decline of the Eocene greenhouse climate

The hothouse conditions of the early Eocene were followed by a gradual global cooling that culminated in the transition into the icehouse climate at around 34 Ma (e.g., Westerhold et al., 2020; Hutchinson et al., 2021). Paleobotanical records from the

Arctic suggest that during the middle Eocene, the mean annual precipitation was >120 cm/yr (Greenwood et al., 2010). In Svalbard, the continental, flora-bearing units are found within the Aspelintoppen Formation (Steel et al. 1978; 1985), probably spanning from the Middle Eocene and not younger than Late Eocene or Early Oligocene (Matthiessen, 1986; Cepek and Krutzsch, 2001), and the Upper Eocene (Golovneva and Zolina, 2023) or Early Oligocene (Head, 1984) Renardodden Formation. Abundant paleoflora from the Aspelintoppen Formation and the Renardodden Formation (Manum,1962, Kvacek

and Manum, 1993; Kvacek et al., 1994 and Cepek and Krutzsch, 2001; Uhl et al., 2007) indicates mean annual precipitation rates of 1423 and 1716 mm/yr, respectively (Golovneva, 2000). Angiosperm morphotypes indicate a strong seasonal precipitation pattern (from 356 to 656 mm in the three wettest months and 112 -247 mm for the three driest months; Clifton, 2012). The high rates of precipitation and increased weathering rates, in combination with active tectonism, promoted the transfer of sand into deeper settings via flood-generated hyperpycnal flows at this time (Grundvåg et al., 2023b). In the late

Eocene, Svalbard was only a few degrees south of the present 78°N (Fig. 3), but fossil plant material indicates that the temperature at that time was much warmer than today, and the estimated mean annual air temperature was around +9 °C (Golovneva, 2000; Goloneva et al. 2023; Uhl et al., 2007). Several studies have evaluated the abundance and diversity of fossil plants, the occurrence of coal seams, as well as fossil insects in the terrestrial parts of the Eocene Aspelintoppen Formation (e.g. Dallmann et al., 1999; Uhl et al., 2007; Marshal et al., 2015). In other areas of the High Arctic, such as Ellesmere Island,

fossil remnants of a varanid lizard, the tortoise Geochelone, and the alligator Allognathosuchus confirm warm temperatures that remained above freezing (e.g., Estes and Hutchinson, 1980; Eberle and Greenwood, 2012).

The sporadic occurrence of glendonites and outsized clasts, the latter possibly indicating rafting by temporal sea ice, in the marine parts of the succession suggests strong seasonal or temporal temperature variations in Svalbard (Kellogg, 1975; Dalland, 1977; Spielhagen and Tripati, 2009). This is in accordance with some of the paleo-floristic/insect studies that infer

freezing temperatures during winter months, and an overall cooling trend for the entire interval (Golovneva, 2000; Uhl et al., 2007; Wappler and Denk, 2011). Some of the signals preserved in the sedimentary rock record may be caused by other allogenic forcing factors than climate fluctuations, such as tectonics and relative sea-level changes, but similar results from a



range of proxies including plant morphotypes support the validity of paleoclimate reconstructions. As such, deconvolving climatic and tectonic signals in tectonically active basins is of major importance. In the restricted Arctic Ocean during the early

middle Eocene (ca. 49 Ma), increased runoff caused stratification of the water column (with a fresh-water lid) and led to the well-known Azolla freshwater algal blooms that eventually contributed to the withdrawing of the atmospheric $CO_2$ and cooling of the global climate from the middle Eocene onwards (Brinkhuis et al., 2006; Speelman et al., 2009).

### 4.5.4. The Eocene-Oligocene Transition and connection with the Arctic Ocean

A major step in the long-term Cenozoic climate evolution took place at the Eocene Oligocene Transition (EOT, ~34 Ma), when decreasing of atmospheric $CO_2$, and changes to ocean gateways led to a development of the first permanent ice-cap in Antarctica and initiated the icehouse type of climate which exists until today (Straume et al., 2020; Westerhold et al., 2020; Hutchinson et al., 2021). However, the global scale of the transition is not fully understood, since in contrast to the well-studied deep sea sites from southern and equatorial regions, the signature of the EOT for the northern high latitudes remains poorly

constrained. Climate models suggests that closing and opening the gateways to the Arctic Ocean (such as the Fram Strait) had equally large impact on the temperature development in the high northern latitudes as the $CO_2$ decrease (e.g., Hutchinson et al., 2019; 2021; Straume et al., 2022; Śliwińska et al., 2023). However, the number of proxy records from the northern polar regions to evaluate the history of the Fram Strait and validate the climate models is limited. The ACEX core (IODP Expedition 302) from the Arctic Ocean contains a hiatus that misses an estimated interval from 44.4 Ma to 18.2 Ma (Backman et al.,

2008). The ODP site 913 from the Greenlandic Sea suffers from a hiatus at the EOT (Eldrett et al., 2004). Furthermore, the existing sea surface temperature proxy data are of extremely low resolution (Liu et al., 2009) in comparison with time-equivalent records from the Labrador Sea and the North Sea (Śliwińska et al., 2019; 2023). Molecular fossil (alkenone) records suggest at least 5°C cooling in the northern high latitudes, associated with the transition towards the coolhouse climate (Liu et al., 2009; Śliwińska et al., 2023). The existing pollen record revealed a significant cooling of ca. 5°C in cold months mean

temperatures on East Greenland across the EOT (Eldrett et al., 2009).

During the EOT, Svalbard was already located at ~80°N and therefore provides unique insights into the climate evolution across the EOT in the northern high latitudes. In the CSB the youngest Paleogene unit is the Aspelintoppen Formation, which is assigned to the Late Eocene (plant fragments) or Oligocene (mollusks) (Manum and Throndsen, 1986). Unfortunately, the age model for this formation remains poorly constrained. With an improved age model, the Eocene–Oligocene succession

could provide a valuable contribution to the atmospheric temperature evolution across the EOT in the northern high latitudes and be further resolved with pollen records, comparable with the Norwegian-Greenland Sea (Eldrett et al., 2009). Marine to terrestrial deposits of possible late Eocene to Oligocene age have also been reported mainly from the exposed parts of the Forlandsundet and Bellsund grabens on the West Spitsbergen margin (Gabrielsen et al., 1992; Weber, 2019; Śliwińska and Head, 2020; Schaaf et al., 2021) . However, these studies are very limited. The foraminifera assemblages collected from the

Sarstangen conglomerate at the Balanuspynten profile on the eastern side of the Forlandsundet Graben reveal the presence of





marine Oligocene strata assigned to the Buchananisen Group (Feyling-Hanssen and Ulleberg, 1984). This age has later been substantiated by palynostratigraphic analyses, which suggests an early to middle Oligocene age, at least for the sediments exposed along the eastern basin margin (Schaaf et al., 2021). The two foraminiferal zones (TA and TB) that were originally assigned to the middle to upper Oligocene, can more accurately can be assigned to the lower Oligocene (lower Rupelian; the

TA zone) and the upper Oligocene (lowery Chattian; the TB zone). The early Oligocene age of the marine strata is confirmed by the presence of dinocyst *Svalbardella cooksoniae* (Manum, 1960). The appearance interval of *S. cooksoniae* in the earliest Oligocene seems to be associated with a cooling interval (Śliwińska and Heilmann-Clausen, 2011). A single sample from the Calypsostranda Group at the Renardodden section on the southern shore of Bellsund, a structural outlier interpreted to be an exposed part of the Bellsund Graben, has yielded dinocysts of late Eocene or early Oligocene age (Head, 1984; Śliwińska and

Head, 2020). Based on the association of pollen in the Skilvika Formation, an upper Paleocene to Eocene age can be suggested for the lower part of the section (Weber, 2019).

### 4.5.5 Neogene hiatus

Svalbard experienced two uplift phases in recent times. The first and major uplift phase started in the Eocene (>36 Ma) and persisted to ca. 10 Ma. This was followed by less prominent uplift from ca. 10 Ma onwards that generated the modern

topography of the archipelago (Dörr et al., 2013). These uplift events are matched by contemporaneous uplift phases in Greenland, the Barents Shelf, and Baltica (Dörr et al., 2013) and are attributed to crustal thinning and the onset of ocean spreading in the Arctic and North Atlantic driven by mantle processes related to anomalously hot mantle underlying this part of the Arctic (Green and Duddy, 2010). The presence of thick pre-glacial (Miocene and Pliocene? 23 to 2.58 Ma) and glacial (late Pliocene and Pleistocene) offshore clastic wedges along the western and northern margins of Spitsbergen (Hjelstuen et

al., 1996; Lasabuda et al., 2018; Alexandropoulou et al., 2021) suggests a net denudation of ca. 3 km (Riis and Fjeldskaar, 1992; Lasabuda et al., 2021). The overall amplitude of Neogene uplift decreases eastwards with the uplift along parts of WSFTB exceeding 2.5 km and over 1.5 km in the CSB (Dörr et al., 2013). Estimates based on organic geochemical proxies suggest total uplift of 2.5 to 3.5 km (Throndsen, 1982; Marshall et al., 2015; Olaussen et al., 2019). As a result, Miocene and Pliocene sedimentary strata are not preserved anywhere on Svalbard, while only intermittent remains of Pleistocene glacial

deposits are found (Ingólfsson and Landvik, 2013). Uplift of 9 mm/yr continues in central-western Spitsbergen today, with only 1 mm/yr of that attributed to isostatic rebound due to the recent Weichselian glaciation (Kierulf et al., 2022).

### 5. Absolute radiochronology of Svalbard stratigraphy

Despite the remarkable continuous stratigraphic successions spanning much of the Phanerozoic in Svalbard, robust absolute

radiometric stratigraphic age constraints are scarce. Except for data on Devonian and older magmatic and metamorphic rocks from the pre-Caledonian "basement" (e.g., Myhre et al., 2008; Petterson et al., 2009; Majka and Kosminska, 2017; McClelland





et al., 2019), no robust ages are published from the stratigraphy pre-dating the Permian-Triassic boundary. The first radiometric age comes from an ash layer within the Deltadalen section at the Permian-Triassic boundary (Fig. 10). A zircon U-Pb chemical abrasion isotope dilution thermal ionization mass spectrometry (CA-ID-TIMS) age of 252.13 ±0.62 Ma from a bentonite bed

(volcanic ash) ca. 15 cm above the first appearance datum (FAD) of the age diagnostic *H. parvus* (onset Triassic) in the Vikinghøgda Fm. ties the biostratigraphic record to an absolute age. This age records the onset of the Triassic in Svalbard and the Panthalassic Ocean within error (Zuchuat et al., 2020). Given the sedimentation rate constraints for this section (Zuchuat et al., 2020), and the uncertainty in the age of the bentonite, there is overlap in age between the FAD of *H. parvus* in Svalbard and the FAD of *H. parvus* of 251.902 ±0.024 Ma at the Induan GSSP (Burgess et al., 2014). This indicates synchronicity of

the end-Permian mass extinction in the Panthalassic and Tethyan domains at a 0.2 % (2σ) level of uncertainty.

The next absolute stratigraphic tie point occurs in the Barremian to lower Aptian Helvetiafjellet Formation, where a bentonite layer from two cores taken in Longyearbyen (DH-3 and DH-7; Fig. 3) was dated to 123.1 ± 0.3 Ma (zircon U-Pb CA-ID-TIMS; Corfu et al., 2013; Midtkandal et al., 2016). Bio- and chemo-stratigraphical evidence suggests that this bentonite layer is of mid-Barremian age (Midtkandal et al., 2016), and that it occurs ~40 m below the Barremian–Aptian boundary and the

onset of the Early Aptian Oceanic Anoxic event 1a (OAE1a). Subsequent magnetostratigraphy tied this bentonite age to the magnetic polarity record and hence Zhang et al. (2021) were able to calculate an age of 121.2 ±0.4 Ma for the Barremian – Aptian boundary, accepting the M0r magnetochron as a boundary marker. Based on available ages for the HALIP in Svalbard (124.7 ±0.3 Ma to 123.9 ±0.3 Ma; Corfu et al., 2013) and Franz Jozef Land (~122-123 Ma; Corfu et al., 2013), there is no overlap in age with a HALIP magmatic pulse of this event with the OAE1a. However, mafic lavas and intrusives from the

Sverdrup basin in northern Canada do show overlapping ages (Evenchick et al., 2015; Dockman et al., 2018) with the updated Barremian – Aptian boundary age not precluding a relationship between a pulse of the HALIP with the EAO1a.

To our knowledge, there are no further absolute age constraints published for the Mesozoic stratigraphy of Svalbard. However, the onset of Paleogene sedimentation in the CSB (Fig. 4) and the Paleocene to early Eocene stratigraphy is well constrained through high precision zircon CA-ID-TIMS U-Pb ages (Charles et al., 2011; Jones et al., 2017; Jochmann et al., 2020). A

bentonite bed towards the base of the Firkanten Formation (see diagram in Fig. 12), dated from three different parts of the basin, yielded an age of 61.596 ±0.028 Ma overlapping with the Danian–Selandian boundary, and a bentonite from the lower part of the overlying Basilika Formation has an age of 59.32 ±0.19 Ma overlapping with the Selandian–Thanetian boundary (Jones et al., 2017). Based on these marker horizons, deposition within the CSB was estimated to start around 61.76 ±0.09 Ma. A bentonite horizon within the Frysjaodden Formation was dated to 55.785 ±0.034 Ma by Charles et al. (2011). This ash layer

is a key marker bed for constraining the Paleocene–Eocene boundary as it is found within the PETM CIE in Svalbard strata. Charles et al. (2011) used this age and possible precession cycles within borehole BH09/05 to estimate an age of 55.866 ±0.098 Ma for the Paleocene–Eocene boundary (i.e., the PETM onset).

To date, there are no further robust and precise radiometric ages from Svalbard's Paleozoic strata. However, well-studied outcrop successions and abundant drill core material from much of the Paleozoic offers an excellent possibility to search for



target volcanic rocks in the stratigraphy that may be dated by high precision zircon U-Pb methods, thus improving the chronostratigraphy not only of Svalbard but potentially also globally (e.g., Zhang et al., 2021).

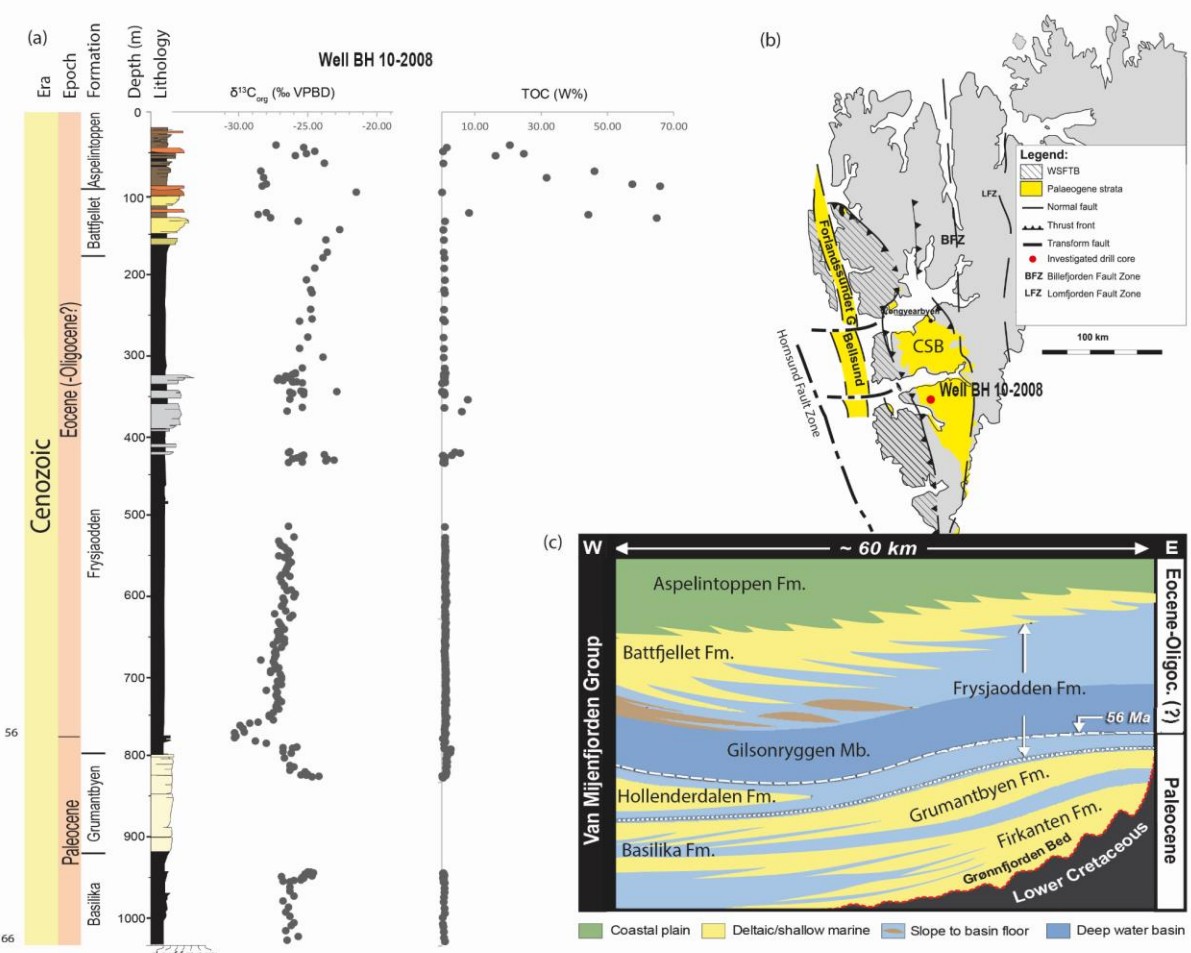

**Figure 12.** **(a) Geochemical data from well BH 10-2008 is from Doerner et al., (2020), and borehole stratigraphy after Grundvåg et**
**al. (2014). (b) geological map showing the outline of Paleocene deposits and position of BH 10-2008 from Helland-Hansen &**
**Grundvåg (2021). (c) Litostratigraphic summary of the Paleogene deposits of the Central Spitsbergen Basin. Modified from Helland-**
**Hansen & Grundvåg (2021). The age of the Paleocene-Eocene boundary from Charles et al. (2011) and Harding et al. (2011).**





## 6. Evolution of the Phanerozoic climate in Svalbard

**Figure 13. Correlation of the Phanerozoic climate plot based mainly on climate-sensitive facies (See Supplementary material), supplemented by biological and geochemical proxies with the paleogeographic position of Svalbard (Scotese, and Wright, 2018) and compilation of the published global average temperature curves.**

The parameters controlling the Phanerozoic climate in Svalbard can be simplified to the two core elements: global distribution of paleo-climatic zones (Köppen belts; e.g., Boucot et al., 2013) and the paleo-latitude of Svalbard (e.g., Steel and Worsley, 1984; Torsvik and Cocks, 2017, Olaussen et al., 2025). The deep-time models of Phanerozoic climate either for low latitude regions between 40°N and 40°S (Fig. 13; based on oxygen isotope record (Song et al., 2019; Verard and Veizer, 2019; Veizer and Prokoph, 2015; Grossman, 2012; Royer et al., 2004) or Global Average Temperature (GAT; Wing and Huber 2019, Valdes et al. 2018, Mills et al. 2019), are characterized by a "Double Hump" pattern (Fig. 13; Fischer, 1981; 1982; 1984; Frakes et al., 1992, Scotese et al., 1999; Summerhayes, 2015). The GAT trends show high temperatures during the early Paleozoic, cooler temperatures during the late Paleozoic, followed by warmer Mesozoic and early Cenozoic temperatures, finally returning to cooler temperatures in the late Cenozoic (Fig. 13). This pattern is formed in response to breakup and accretion of



supercontinents (Nance et al., 2014; van der Meer et al., 2014; 2017). The geochemical proxies, such as oxygen isotopes, have

limited application in Svalbard due to burial, diagenetic, and hydrothermal alterations (e.g., Buggies, 2013; Matysik et al., 2017). Therefore, the overall Phanerozoic climate trends illustrated in Figure 14 are based on climate sensitive lithofacies supplemented by biological and geochemical proxies where possible. The data sets used for all deep-time paleo-climatic and paleo-environmental studies in Svalbard are coming mainly for the outcrop's investigations (See Fig. 14). Only about 18% of the studies mainly of the Mesozoic and Cenozoic strata is addressing borehole data (Fig. 14).

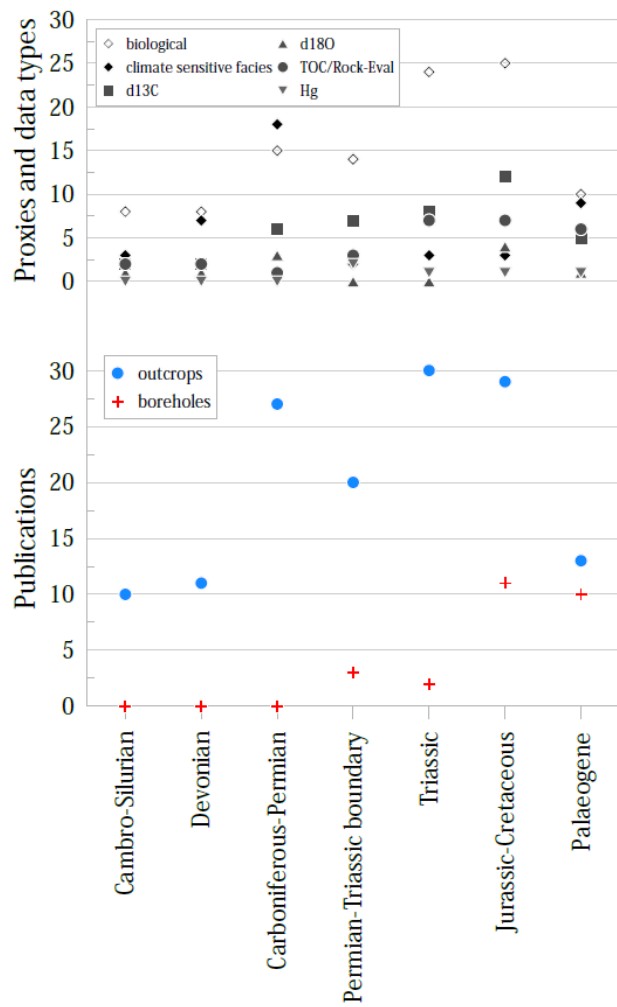


**Figure 14. Compilation plot of main proxies as a function of geological time, based on Table 2.**





## 6. 1. Paleozoic

The warm and tropical to subtropical climate recorded in the early Paleozoic sedimentary succession in Svalbard reflects its
near equatorial position (Figure 3). The Paleozoic climate slightly oscillates between the tropical and subtropical, dry climate.
While the Late Ordovician cooling (ca. 460–440 Ma) is not recorded in Svalbard, likely due to a stratigraphic gap, the highest
temperatures were reached in the Mississippian (Fig. 13). The northward drift of the continental plate on which Svalbard is
located accelerated in the Mississippian to the late Triassic (Fig. 13; Torsvik and Cocks, 2019), while its impact on the climate
can be recognized in the Carboniferous-Permian succession. The Mississippian to Pennsylvanian shift in climate seen as a
transition of climate sensitive facies form the Mississippian coal-bearing deposits indicating tropical humid climate, through
the Pennsylvanian semi-arid subtropical evaporites and siliciclastic red beds, and warm-water carbonate facies (chapter 4) is
potentially accommodated by the changes in paleolatitude (Steel and Worsley, 1984; Torsvik and Cocks, 2019), however, the
impact of global cooling related to the LPIA (e.g. Isbell et al., 2008) should be also considered. The following cooling trend
expressed by the shift from cool-water to cold-water carbonate platform deposits (chapter 4) takes place despite the overall
increased trend of global temperature in the Permian (Fig. 13).

## 6. 2. Mesozoic

Despite the general northward migration through different climate zones, Svalbard's sedimentary Mesozoic strata recorded a
more complex story reflecting both global and regional climatic and environmental trends and the control from the
paleolatitude position of Svalbard, which is evident in the Paleozoic, is not clearly seen. Indeed, the Mesozoic is characterized
at the global scale by overall warm conditions, punctuated by several established hyperthermal events and potential ones, such
as, the EPME, the late Dienerian biotic crisis, the Smithian-Spathian transition, and the Carnian Pluvial Episode. Most of these
hyperthermals are associated with the emplacement of LIPs or smaller-scale volcanic activities (see chapter 4 for details). The
emplacement of one LIP, however, might have triggered a global cooling rather than a global warming episode (the Weissert
Event; Martinez et al., 2023). In addition to these individual events, longer-lived climate perturbations and oscillations are
recorded in the Mesozoic strata of Svalbard, including periodic cooling-warming episodes during the Middle Jurassic to Early
Cretaceous, as testified by the presence of cool-climate-indicators such as glendonite crystals in certain stratigraphic intervals.
These climatic cycles generated (glacio-?) eustatic sea-level variations, leading to changes in global oceanic circulation as
shallow seaways were periodically exposed during sea-level lowstands. These climatic and (glacio-?) eustatic sea-level
variations impacted the amount and the redistribution of precipitation, runoff, temperature, salinity, water-mass stratification,
nutrients and productivity between basins, as well as notably impacting the global carbon- and phosphorus cycles. One of these
periodic perturbations recorded in Svalbard was the VOICE event.



## 6.3. Cenozoic

The Late Cretaceous climate maximum along with the Neogene cooling (Fig. 13) are not recorded in Svalbard due to
stratigraphic gaps. During the Cenozoic, Svalbard was positioned at the high northern latitudes (the Arctic Circle). Under the
greenhouse conditions of the Paleocene and Eocene, paleoflora of Svalbard suggest a humid and temperate climate at that
time, punctuated by hyperthermal conditions during the PETM (Chapter 4). Therefore, palynoflora from Svalbard provides a
unique insight into the high latitude end member for estimating the latitudinal gradient under the greenhouse conditions of the
early Cenozoic. Notably, the paleoflora suggest slightly warmer temperatures and higher mean annual precipitation during the
Paleocene, than in the Eocene-earliest Oligocene. This may be an effect of the decline of the greenhouse climate. During the
early Oligocene, Svalbard (located at approximately 80°N) experienced significant cooling: the presence of specific dinocysts
suggests a notable cooling interval during this period. The cooling trend is a direct response to the global cooling and the
transition to the modern icehouse climate state with a bipolar glaciation (Fig. 13).

## Conclusions

In this contribution we synthesized the review of the Pre-Quaternary Phanerozoic (ca. 541 Ma to 2.588 Ma) deep-time
paleoclimatic research conducted on Svalbard's sedimentary succession and conclude that:

- Svalbard represents an excellent location for studying multiple globally relevant paleoclimatic events within a
  spatially constrained area.
- Svalbard's geological record is influenced both by its northward drift and the recurring influence of large igneous
  provinces that affected its climate. These include, at least, the end-Permian Siberian Traps LIP, the Early Cretaceous
  High Arctic LIP and the Paleogene North Atlantic Igneous Province.
- Specific events that record environmental perturbations at the global scale imprinted on Svalbard's geological record
  include the LPIA, the End-Permian Mass Extinction, Early Cretaceous anoxic events and cold snaps, and the
  Paleocene-Eocene Thermal Maximum.
- These are recorded as changes in biological, lithological, and chemical proxies of past climates preserved in both
  outcrops and drill cores.
- Absolute radiometric ages constrain the continuous stratigraphic successions but are unevenly distributed throughout
  the stratigraphy.
- Of the 148 key publications, the most used proxies to quantify past environments and climate are sedimentological
  studies of biological indicators (104), climate sensitive facies (45), carbon isotopes $\delta^{13}C$ (42), oxygen isotopes $\delta^{18}O$
  (10), and mercury (5).

This contribution serves as an important foundation for future deep-time paleoclimate studies utilizing outcrops, opportunistic
drill cores or dedicated deep-time paleoclimate scientific drilling planned in the near future.





**Supplementary material**

The supplementary material presents a comprehensive table summarizing 148 scientific articles. The table outlines the diverse proxies utilized in deep-time climate studies conducted in Svalbard, highlighting their respective methodologies, findings, stratigraphic succession and implications for paleoenvironmental reconstructions. It can be accessed here: https://doi.org/10.5281/zenodo.14334261.

**Data availability**

All data sets are coming from published scientific contributions or are part of the manuscript, including The supplement presents a comprehensive table summarizing 148 scientific articles

**Authors contributions**

AASS- Data Curation, Conceptualization, writing – Original Draft and Editing, visualization, Project administration

PB - Writing-Review and Editing, Visualization

VSE - Writing-Original Draft, Writing-Review and Editing

WJF - Writing-Original Draft, Writing-Review and Editing

SAG - Writing-Original Draft, Writing-Review and Editing

MEJ - Writing-Original Draft, Writing-Review and Editing, Visualization

MTJ - Writing-Original Draft, Writing-Review and Editing

GES - Writing-Original Draft, Writing-Review and Editing, Visualization

KKS - Writing-Original Draft, Writing-Review and Editing

MLV - Writing-Original Draft, Writing-Review and Editing, Visualization

VZ - Writing-Original Draft, Writing-Review and Editing, Visualization

LEA - Writing-Original Draft

JIF - Conceptualization,

JMG - Writing-Original Draft

WHH - Editing

MAJ- Writing- Review and Editing

MMJ-

EJ -Writing- Review and Editing

MK, Visualization and editing,

DK - Writing-Review and Editing

GL -Writing-Original Draft

TM -Data Curation



AM -Writing-Original Draft

SO -Writing-Original Draft

SP - Conceptualization,

GDP - Writing-Original Draft

LS -Writing-Original Draft

KiS- Conceptualization, Writing - Original Draft, Writing-Review and Editing, visualization, Supervision, Funding acquisition

**Competing interests**

The authors declare that they have no conflict of interest.

**Acknowledgements**

Lilith Kuckero and Moritz Boehme kindly contributed with reference formatting. Atle Mørk is acknowledged for comments on the manuscript. Copilot was used to help organizing the references.

**Funding**

This study was jointly financed by the Research Council of Norway (through Svalbard Strategic Grants (283488, 295781, 352811), research projects (326238, 336293), Centres for Environment-friendly Energy Research (257579), and Centres of
Excellence (223272, 332523), industry funding (Locra, ARCEx), international funding agencies (MagellanPlus) and University of Arctic (UArctic) collaboration projects. Funding was also provided from the European Commission, Horizon 2020 (101024218, and grant no. 754513), the Danish Council for Independent Research/Natural Sciences (DFF/FNU) grant (no. 11-107497), the Geo-Mapping for Energy and Minerals Program and GEM GeoNorth (Geological Survey of Canada, Government of Canada) and a postdoctoral Internationalisation Fellowship from Carlsberg Foundation.

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

## C

Canadell, J. G., Monteiro, P. M., Costa, M. H., Cotrim da Cunha, L., Cox, P. M., Eliseev, A. V., ... & Zickfeld, K.:
Intergovernmental Panel on Climate Change (IPCC). Global carbon and other biogeochemical cycles and feedbacks. In: Climate change 2021: The physical science basis. Contribution of working group I to the sixth assessment report of the intergovernmental panel on climate change (pp. 673-816). Cambridge University Press, 2023.

Capelli, I.A., Scasso, R.A., Spangenberg, J.E., Kietzmann, D.A., Cravero, F., Duperron, M., and Adatte, T.:
Mineralogy and geochemistry of deeply-buried marine sediments of the Vaca Muerta-Quintuco system in the Neuquén Basin (Chacay Melehue section), Argentina: Paleoclimatic and paleoenvironmental implications for the global Tithonian-Valanginian reconstructions, *Journal of South American Earth Sciences*, 107, 103103, 2021.

Cavalheiro, L., Wagner, T., Steinig, S., Bottini, C., Dummann, W., Esegbue, O., Gambacorta, G., Giraldo-Gómez,
V., Farnsworth, A., and Flögel, S.: Impact of global cooling on Early Cretaceous high $pCO_2$ world during the Weissert Event, *Nature Communications*, 12 (1), 5411, 2021.

Cepek, P., and Krutzsch, W.: Conflicting interpretations of the Tertiary Biostratigraphy of Spitsbergen and New Palynological Results. In: Tessensohn F (Ed), Intra-continental fold belts, CASE 1: West Spitsbergen. *Geologisches Jahrbuch Reihe B*, Polar Issue 7. Federal Institute for Geosciences and Natural Resources,
Hannover, pp. 551–602, 2001.

Charles, A.J., Condon, D.J., Harding, I.C., Pälike, H., Marshall, J.E., Cui, Y., Kump, L., and Croudace, I.W.: Constraints on the numerical age of the Paleocene-Eocene boundary, *Geochemistry, Geophysics, Geosystems*, 12 (6), https://doi.org/10.1029/2010GC003426, 2011.

Chambers, L.M., Pringle, M.S., and Fitton, J.G.: Phreatomagmatic eruptions on the Ontong Java Plateau: an Aptian
[40]Ar/[39]Ar age for volcaniclastic rocks at ODP Site 1184, *Geology Society of London, Special Publications*, 229 (1), 325–331, 2004.

Chen, Z.Q., Tong, J., and Fraiser, M.L.: Trace fossil evidence for restoration of marine ecosystems following the end-Permian mass extinction in the Lower Yangtze region, South China, *Palaeogeography, Palaeoclimatology, Palaeoecology*, 299 (3-4), 449-474, 2011.

Chen, J., Shen, S., Li, X., Xu, Y., Joachimski, M.M., Bowring, S.A., Erwin, D.H., Yuan, D., Chen, B., Zhang, H., Wang, Y., Cao, C., Zheng, Q., and Mu, L.: High-resolution SIMS oxygen isotope analysis on condodont apatite from South China and implications for the end-Permian mass extinction, *Palaeogeography, Palaeoclimatology, Palaeoecology*, 448, 26-38, 2016.



Clemmensen, A., and Thomsen, E.: Palaeoenvironmental changes across the Danian−Selandian boundary in the North Sea Basin, *Palaeogeography, Palaeoclimatology, Palaeoecology*, 219 (3), 351-394, https://doi.org/10.1016/j.palaeo.2005.01.005, 2005.

Clifton, A.: The Eocene flora of Svalbard and its climatic significance, *Unpublished PhD Thesis, The University of Leeds*, Leeds, 401 pp., 2012.

Coldwell, B.C., and Pankhurst, M.: Evaluating the influence of meteorite impact events on global potassium feldspar availability to the atmosphere since 600 Ma, *Journal of the Geological Society*, 176 (2), 209-224, 2019.

Coldwell, B.C., and Pankhurst, M.J.: Evaluating the influence of meteorite impact events on global potassium feldspar availability to the atmosphere since 600 Ma, *Journal of the Geological Society*, 176 (2), 209-224, https://doi.org/10.1144/jgs2018-084, 2018.

Corfu, F., Polteau, S., Planke, S., Faleide, J.I., Svensen, H., Zayoncheck, A., and Stolbov, N.: U–Pb geochronology of Cretaceous magmatism on Svalbard and Franz Josef Land, Barents Sea large igneous province. *Geological Magazine*, 150 (6), 1127-1135, https://doi.org/10.1017/S0016756813000162, 2013.

Cramer, B.S., Wright, J.D., Kent, D.V., and Aubry, M.P., 2003. Orbital climate forcing of δ13C excursions in the late Paleocene–early Eocene (chrons C24n–C25n), *Paleoceanography*, 18 (4), https://doi.org/10.1029/2003PA000909, 2003.

Cui, Y.: Carbon addition during the Paleocene-Eocene Thermal Maximum: model inversion of a new, high-resolution carbon isotope record from Svalbard, (Master Thesis, The Pennsylvania State University) 2010.

Cui, Y., Kump, L. R., Ridgwell, A. J., Charles, A. J., Junium, C. K., Diefendorf, A. F., Freeman, K. H., Urban, N. M.,and Harding, I. C.: Slow release of fossil carbon during the Palaeocene–Eocene Thermal Maximum. *Nature Geoscience*, 4(7), 481-485, 2011.

Cui, Y., Diefendorf, A.F., Kump, L.R., Jiang, S., and Freeman, K.H.: Synchronous marine and terrestrial carbon cycle perturbation in the high arctic during the PETM, *Paleoceanography and Paleoclimatology*, 36 (4), e2020PA003942, , https://doi.org/10.1029/2020PA003942, 2021.

Cutbill, J., and Challinor, A.: Revision of the stratigraphical scheme for the Carboniferous and Permian rocks of Spitsbergen and Bjørnøya, *Geological Magazine*, 102 (5), 418-439, 1965.

**D**

Daëron, M., Drysdale, R.N., Peral, M., Huyghe, D., Blamart, D., Coplen, T.B., Lartaud, F., and Zanchetta, G.: Most Earth-surface calcites precipitate out of isotopic equilibrium. *Nat. Commun.*, 10 (1), 429, 2019.

Dal Corso, J., Ruffell, A., and Preto, N.: The Carnian Pluvial Episode (Late Triassic): New insights into this important time of global environmental and biological change. *Journal of the Geological Society*, *175* (6), 986-988, 2018.



Dal Corso, J., Mills, B.J., Chu, D., Newton, R.J., and Song, H.: Background Earth system state amplified Carnian (Late Triassic) environmental changes. *Earth and Planetary Science Letters*, 578, 117321, https://doi.org/10.1016/j.epsl.2021.117321, 2022.

Dalland, A.: Erratic clasts in the Lower Tertiary deposits of Svalbard--evidence of transport by winter ice, Norsk Polarinstitutt Arbok, 1976.

Dalland, A.: Mesozoic sedimentary succession at Andoy, northern Norway, and relation to structural development of the North Atlantic area, Geology of the North Atlantic Borderlands — Memoir 7, *CSPG Special Publications* 563-584, 1981.

Dallmann, W.K., Andresen, A., Bergh, S.G., Maher jr, H.D., and Ohta, Y.: Tertiary fold-and-thrust belt of Spitsbergen, Svalbard, Norsk Polarinstitutt, *Meddelelser* Nr 128, Oslo, 1993.

Dallmann, W.K., Gjelberg, J.G., Harland, W.B., Johannessen, E.P., Keilen, H.B., Lønøy, A., Nilson, I., Worsley, D.: Upper palaeozoic lithostratigraphy. In: Dallmann, W.K. (Ed.), Lithostratigraphic Lexicon of Svalbard. Norsk Polarinstitutt, Tromsø, pp. 25–126, 1999.

Dallmann, W., (Ed): Lithostratigraphic lexicon of Svalbard. Norwegian Polar Institute, 1999.

Dallmann, W.K.E. (Ed): Geoscience Atlas of Svalbard, *Norsk Polarinstitutt Rapportserie*, 148 ,http://hdl.handle.net/11250/2580810, 2015.

Dalseg, T.S., Nakrem, H.A., and Smelror, M.: Dinoflagellate cyst biostratigraphy, palynofacies, depositional environment and sequence stratigraphy of the Agardhfjellet Formation (Upper Jurassic-Lower Cretaceous) in central Spitsbergen (Arctic Norway), *Norwegian Journal of Geology*, 96 (2), 1-14, 2016.

David, G.G., Haakon, F., Niels, H., and Anthony, K.H.: From the Early Paleozoic Platforms of Baltica and Laurentia to the Caledonide Orogen of Scandinavia and Greenland, *International Union of Geological Sciences*, 31 (1), 44-51, https://doi.org/10.18814/epiiugs/2008/v31i1/007, 2008.

Davies, N.S., Berry, C.M., Marshall, J.E., Wellman, C.H., and Lindemann, F.-J.: The Devonian landscape factory: plant–sediment interactions in the Old Red Sandstone of Svalbard and the rise of vegetation as a biogeomorphic agent. *Journal of the Geological Society*, 178 (5), https://doi.org/10.1144/jgs2020-225, 2021.

Davis, W.J., Schroeder-Adams, C.J., Galloway, J.M., Herrle, J.O., and Pugh, A.T.: U-Pb geochronology of bentonites from the Upper cretaceous Kanguk Formation, Sverdrup Basin, Arctic Canada: constraints on sedimentation rates, biostratigraphic correlations and the late magmatic history of the High Arctic large Igneous Province. *Geological Magazine*, 154 (4), 757-776, https://doi.org/10.1017/S0016756816000376, 2017.

Delsett, L.L., Novis, L.K., Roberts, A.J., Koevoets, M.J., Hammer, Ø., Druckenmiller, P.S., and Hurum, J.H.: The Slottsmøya marine reptile Lagerstätte: depositional environments, taphonomy and diagenesis, *Geological Society, London, Special Publications*, 434 (1), 165-188, 2016.



Delsett, L.L., Roberts, A.J., Druckenmiller, P.S., and Hurum, J.H.: A new ophthalmosaurid (Ichthyosauria) from Svalbard, Norway, and evolution of the ichthyopterygian pelvic girdle. *PloS One*, 12. (1), e0169971, 2017.

Derby, J., Fritz, R., Longacre, S., Morgan, W., and Sternbach, C.: The Great American Carbonate Bank: The Geology and Economic Resources of the Cambrian-Ordovician Sauk Megasequence of Laurentia, AAPG Memoir, 98, 2013.

Dickens, G.R., O'Neil, J.R., Rea, D.K., and Owen, R.M.: Dissociation of oceanic methane hydrate as a cause of the carbon isotope excursion at the end of the Paleocene, *Paleoceanography*, 10 (6), 965-971, 1995.

Dickson, A.J., Cohen, A.S., and Davies, M.: The Osmium Isotope Signature of Phanerozoic Large Igneous Provinces, *Large Igneous Provinces: A Driver of Global Environmental and Biotic Changes*, 229-246, https://doi.org/10.1002/9781119507444.ch10, 2021.

Dietmar Müller, R., and Spielhagen, R.F.: Evolution of the Central Tertiary Basin of Spitsbergen: towards a synthesis of sediment and plate tectonic history, *Palaeogeography, Palaeoclimatology, Palaeoecology*, 80 (2), 153-172, https://doi.org/10.1016/0031-0182(90)90127-S, 1990.

Dimakis, P., Braathen, B.I., Faleide, J.I., Elverhøi, A., and Gudlaugsson, S.T.: Cenozoic erosion and the preglacial uplift of the Svalbard–Barents Sea region, *Tectonophysics*, 300 (1), 311-327, https://doi.org/10.1016/S0040-1951(98)00245-5, 1998a.

Dimakis, P., Braathen, B.I., Faleide, J.I., Elverhøi, A., and Gudlaugsson, S.T.: Cenozoic erosion and the preglacial uplift of the Svalbard–Barents Sea region, *Tectonophysics*, 300 (1-4), 311-327, 1998b.

Ditchfield, P.W.: High northern palaeolatitude Jurassic-Cretaceous palaeotemperature variation: new data from Kong Karls Land, Svalbard. *Palaeogeography, Palaeoclimatology, Palaeoecology*, 130 (1-4), 163-175, 1997.

Dockman, D., Pearson, D., Heaman, L., Gibson, S., and Sarkar, C.: Timing and origin of magmatism in the Sverdrup Basin, Northern Canada—Implications for lithospheric evolution in the High Arctic Large Igneous Province (HALIP), *Tectonophysics*, 742, 50-65, 2018.

Dodd, S.C., Mac Niocaill, C., and Muxworthy, A.R.: Long duration (> 4 Ma) and steadystate volcanic activity in the early cretaceous Paraná-Etendeka Large Igneous Province: new palaeomagnetic data from Namibia, *Earth Planet. Sci. Lett.*, 414, 16-29, 2015.

Doerner, M., Berner, U., Erdmann, M., and Barth, T.: Geochemical characterization of the depositional environment of Paleocene and Eocene sediments of the Tertiary Central Basin of Svalbard, *Chemical Geology*, 542, 119587, https://doi.org/10.1016/j.chemgeo.2020.119587, 2020.

Drachev, S.: Fold belts and sedimentary basins of the Eurasian Arctic, *Arktos*, 2, 21, 2016.

Duchamp-Alphonse, S., Gardin, S., Fiet, N., Bartolini, A., Blamart, D., and Pagel, M.: Fertilization of the northwestern Tethys (Vocontian basin, SE France) during the Valanginian carbon isotope perturbation:



evidence from calcareous nannofossils and trace element data, *Palaeogeography, Palaeoclimatology, Palaeoecology*, 243 (1-2), 132-151, 2007.

Dummann, W., Schröder-Adams, C., Hofmann, P., Rethemeyer, J., and Herrle, J.O.: Carbon isotope and sequence stratigraphy of the upper Isachsen Formation on Axel Heiberg Island (Nunavut, Canada): High Arctic expression of oceanic anoxic event 1a in a deltaic environment, *Geosphere*, 17 (2), 501-519, 2021.

Dummann, W., Wennrich, V., Schröder-Adams, C. J., Leicher, N., & Herrle, J. O.: Ash deposits link Oceanic Anoxic Event 2 to High Arctic volcanism. *Geology*, 52 (12): 927–932. doi: https://doi.org/10.1130/G52368.1, 2024.

Dustira, A.M., Wignall, P.B., Joachimski, M., Blomeier, D., Hartkopf-Fröder, C., and Bond, D.P.G.: Gradual onset of anoxia across the Permian–Triassic Boundary in Svalbard, Norway, *Palaeogeography, Palaeoclimatology, Palaeoecology*, 374, 303-313, https://doi.org/10.1016/j.palaeo.2013.02.004, 2013.

Dypvik, H., Håkansson, E., and Heinberg, C.: Jurassic and Cretaceous palaeogeography and stratigraphic comparisons in the North Greenland-Svalbard region, *Polar Research*, 21 (1), 91-108, 2002.

Dypvik, H., Riber, L., Burca, F., Rüther, D., Jargvoll, D., Nagy, J., and Jochmann, M.: The Paleocene–Eocene thermal maximum (PETM) in Svalbard—clay mineral and geochemical signals, *Palaeogeography, Palaeoclimatology, Palaeoecology*, 302 (3-4), 156-169, 2011.

Dzyuba, O.S., Izokh, O.P., and Shurygin, B.N.: Carbon isotope excursions in Boreal Jurassic–Cretaceous boundary sections and their correlation potential, *Palaeogeography, Palaeoclimatology, Palaeoecology*, 381-382, 33-46, https://doi.org/10.1016/j.palaeo.2013.04.013, 2013.

Dörr, N., Lisker, F., Clift, P., Carter, A., Gee, D.G., Tebenkov, A., and Spiegel, C.: Late Mesozoic–Cenozoic exhumation history of northern Svalbard and its regional significance: Constraints from apatite fission track
analysis, *Tectonophysics*, 514, 81-92, 2012.

Dörr, N., Clift, P., Lisker, F., and Spiegel, C.: Why is Svalbard an island? Evidence for two-stage uplift, magmatic underplating, and mantle thermal anomalies, *Tectonics*, 32 (3), 473-486, 2013.

**E**

Eberle, J., and Greenwood, D.: Life at the top of the greenhouse Eocene world--A review of the Eocene flora and vertebrate fauna from Canada's High Arctic, *Geological Society of America Bulletin*, 124, 3-23, https://doi.org/10.1130/B30571.1, 2012.

Ekeheien, C., Delsett, L.L., Roberts, A.J., and Hurum, J.H.: Preliminary report on ichthyopterygian elements from the Early Triassic (Spathian) of Spitsbergen, *Norwegian Journal of Geology*, 98 (2), 219-238, 2018.

Eldrett, J., Greenwood, D., Harding, I., and Huber, M.: Increased seasonality in the latest Eocene to earliest Oligocene in northern high latitudes, *Nature*, 459, 969-973, 2009.



Eldrett, J.S., Harding, I.C., Firth, J.V., and Roberts, A.P.: Magnetostratigraphic calibration of Eocene–Oligocene dinoflagellate cyst biostratigraphy from the Norwegian–Greenland Sea, *Marine Geology*, 204 (1-2), 91-127, 2004.

Engelschiøn, V.S., Bernhardsen, S., Wesenlund, F., Hammer, Ø., Hurum, J.H., and Mørk, A.: Bivalve beds reveal rapid changes in ocean oxygenation in the Boreal Middle Triassic–a case study from Svalbard, Norway, *Norwegian Journal of Geology,* 1-25, https://dx.doi.org/10.17850/njg103-2-1, 2023.

Engelschiøn, V.S., Delsett, L.L., Roberts, A.J., and Hurum, J.H.: Large-sized ichthyosaurs from the Lower Saurian niveau of the Vikinghøgda formation (Early Triassic), Marmierfjellet, Spitsbergen, *Norwegian Journal of Geology*, 98 (2), 239-266, 2018.

Engen, Ø., Faleide, J.I., and Dyreng, T.K.: Opening of the Fram Strait gateway: A review of plate tectonic constraints, *Tectonophysics*, 450 (1-4), 51-69, 2008.

Erba, E., Bartolini, A., and Larson, R.L.: Valanginian Weissert oceanic anoxic event, *Geology*, 32 (2), 149-152, 2004.

Erba, E., Duncan, R.A., Bottini, C., Tiraboschi, D., Weissert, H., Jenkyns, H.C., and Malinverno, A.: Environmental consequences of Ontong Java Plateau and Kerguelen plateau volcanism. The origin, evolution, and environmental impact of oceanic large igneous provinces, *Geological Society of America Special Paper*, 511, 271-303, https://doi.org/10.1130/2015.2511(15), 2015.

Erickson, T.M., Timms, N.E., Kirkland, C.L., Tohver, E., Cavosie, A.J., Pearce, M.A., and Reddy, S.M.: Shocked monazite chronometry: integrating microstructural and in situ isotopic age data for determining precise impact ages, *Contributions to Mineralogy and Petrology*, 172 (2), 11, https://doi.org/10.1007/s00410-017-1328-2, 2017.

Ernst, R.E., and Youbi, N.: How Large Igneous Provinces affect global climate, sometimes cause mass extinctions, and represent natural markers in the geological record, *Palaeogeography, palaeoclimatology, palaeoecology*, 478, 30-52, 2017

Estes, R., and Hutchinson, J.H.: Eocene Lower Vertebrates from Ellesmere Island, Canadian Arctic Archipelago, *Palaeogeogr. Palaeocl.*, 30, 325-347, 1980.

Estrada, S., and Henjes-Kunst, F.: $^{40}Ar$-$^{39}Ar$ and U-Pb dating of Cretaceous continental rift-related magmatism on the northeast Canadian Arctic margin, *Zeitschrift der Deutschen Gesellschaft für Geowissenschaften*, 164 (1), 107-130, 2013.

Evans, D., Sagoo, N., Renema, W., Cotton, L.J., Müller, W., Todd, J.A., Saraswati, P.K., Stassen, P., Ziegler, M., and Pearson, P.N.: Eocene greenhouse climate revealed by coupled clumped isotope-Mg/Ca thermometry, *Proceedings of the National Academy of Sciences*, 115 (6), 1174-1179, http://www.pnas.org/cgi/doi/10.1073/pnas.1714744115, 2018.



Evenchick, C.A., Davis, W.J., Bédard, J.H., Hayward, N., and Friedman, R.M.: Evidence for protracted High Arctic
large igneous province magmatism in the central Sverdrup Basin from stratigraphy, geochronology, and
paleodepths of saucer-shaped sills, *GSA Bulletin*, 127 (9-10), 1366-1390,
https://doi.org/10.1130/b31190.1, 2015.

Ezaki, Y., Kawamura, T., and Nakamura, K.: Kapp Starostin Formation in Spitsbergen: a sedimentary and faunal
record of Late Permian palaeoenvironments in an Arctic region, In: *Pangea: Global Environments and
Resources — Memoir 17*, 647-655, 1994.

## F

Fairchild, I.J.: The Orustdalen Formation of Brøggerhalvøya, Svalbard: A fan delta complex of Dinantian/Namurian
age, *Polar Research*, 1982 (1), 17-34, https://doi.org/10.1111/j.1751-8369.1982.tb00470.x, 1982.

Fairchild, I.J., Fleming, E.J., Bao, H., Benn, D.I., Boomer, I., Dublyansky, Y.V., Halverson, G.P., Hambrey, M.J.,
Hendy, C., McMillan, E.A., Spötl, C., Stevenson, C.T.E., and Wynn, P.M.,: Continental carbonate facies of
a Neoproterozoic panglaciation, north-east Svalbard, *Sedimentology*, 63 (2), 443-497,
https://doi.org/10.1111/sed.12252, 2016.

Faleide, J.I., Tsikalas, F., Breivik, A.J., Mjelde, R., Ritzmann, O., Engen, Ø., Wilson, J., and Eldholm, O.,: Structure
and evolution of the continental margin off Norway and the Barents Sea, *Episodes Journal of International
Geoscience*, 31 (1), 82-91, 2008.

Faleide Jan, I., Pease, V., Curtis, M., Klitzke, P., Minakov, A., Scheck-Wenderoth, M., Kostyuchenko, S., and
Zayonchek, A.: Tectonic implications of the lithospheric structure across the Barents and Kara shelves,
*Geological Society, London, Special Publications*, 460 (1), 285-314, https://doi.org/10.1144/SP460.18,
2018.

Fallatah, M. I., Alnazghah, M., Kerans, C., & Al-Hussaini, A.: Sedimentology and carbon isotope stratigraphy from
the Late Jurassic – Early Cretaceous of the Arabian plate: The Weissert event and the VOICE in the Tethys
Realm? *Marine and Petroleum Geology,* 161, 106670. https://doi.org/10.1016/j.marpetgeo.2023.106670,
2024.


Farabegoli, E., Perri, M.C., and Posenato, R.: Environmental and biotic changes across the Permian–Triassic
boundary in western Tethys: the Bulla parastratotype, Italy, *Global and Planetary Change*, 55 (1-3), 109-
135, http://dx.doi.org/10.1016/j.gloplacha.2006.06.009, 2007.

Franeck, F: Perspectives on the Great Ordovician Biodiversification Event-local to global patterns, (PhD Thesis,
University of Oslo), 2020.



Fischer, A.G.: Climatic oscillations in the biosphere. Biotic crises in ecological and evolutionary time, *Academic Press,* 103-131, 1981.

Fischer, A.G.: Long-term climate oscillations recorded in stratigraphy. Climate in Earth history, *Academy Press, Washington, D.C*, 97-104, 1982.

Fischer, A.G.: The two Phanerozoic supercycles, *In*: *Catastrophes and Earth history* (eds. W.A. Berggren & J.A. Van Couvering), Princetown Univeristy Press, 129-150, https://doi.org/10.1515/9781400853281.129, 1984.

Fortey, R., and Bruton, D.: Cambrian-Ordovician rocks adjacent to Hinlopenstretet, North Ny Friesland, Spitsbergen, *Geological Society of America Bulletin*, 84 (7), 2227-2242, 1973.

Foster, W.J: Palaeoecology of the late Permian mass extinction and subsequent recovery, Unpublished PhD thesis, Plymouth University, 2015.

Foster, W.J., Danise, S., Price, G.D., and Twitchett, R.J.: Subsequent biotic crises delayed marine recovery following the late Permian mass extinction event in northern Italy, *PLoS One*, 12 (3), e0172321, https://doi.org/10.1371/journal.pone.0172321, 2017a.

Foster, W.J., Danise, S., and Twitchett, R.J.: A silicified Early Triassic marine assemblage from Svalbard, *Journal of Systematic Palaeontology*, 15 (10), 851-877, https://doi.org/10.1080/14772019.2016.1245680, 2017b.

Foster, W.J., Lehrmann, D.J., Yu, M., Ji, L., and Martindale, R.C.: Persistent environmental stress delayed the recovery of marine communities in the aftermath of the latest Permian mass extinction. *Paleoceanography and Paleoclimatology*, 33 (4), 338-353, 2018.

Foster, W.J., Hirtz, J.A., Farrell, C., Reistroffer, M., Twitchett, R.J., and Martindale, R.C.: Bioindicators of severe ocean acidification are absent from the end-Permian mass extinction, *Scientific Reports*, 12 (1), 1202, https://doi.org/10.1038/s41598-022-04991-9, 2022.

Foster, W.J., Asatryan, G., Rauzi, S., Botting, J., Buchwald, S., Lazarus, D., Isson, T., Renaudie, J., and Kiessling, W.: Response of Siliceous Marine Organisms to the Permian-Triassic Climate Crisis Based on New
Findings From Central Spitsbergen, Svalbard, *Paleoceanography and Paleoclimatology*, 38 (12), e2023PA004766, 2023.

Frakes, L.A., Francis, J.E., and Syktus, J.I.: Climate modes of the Phanerozoic, *Cambridge University Press, Cambridge*, 274 pp, 1992.

Franks, P.J., Berry, J.A., Lombardozzi, D.L., and Bonan, G.B.: Stomatal Function across Temporal and Spatial
Scales: Deep-Time Trends, Land-Atmosphere Coupling and Global Models, *Plant Physiology*, 174 (2), 583-602, https://doi.org/10.1104/pp.17.00287, 2017.

Friend, P.: Fluviatile sedimentary structures in the Wood Bay series (Devonian) of Spitsbergen, *Sedimentology*, 5 (1), 39-68, 1965.





Friend, P., Harland, W., Rogers, D., Snape, I., and Thornley, R.: Late Silurian and Early Devonian stratigraphy and
probable strike-slip tectonics in northwestern Spitsbergen, *Geological Magazine*, 134 (4), 459-479, 1997.

Friend, P.F.: The Devonian stratigraphy of north and central Spitsbergen. *Proceedings of the Yorkshire Geological Society*, 33 (1), 77-118, https://doi.org/10.1144/pygs.33.1.77, 1961.

Friend, P.F., and Moody-Stuart, M.: Carbonate deposition on the river floodplains of the Wood Bay Formation (Devonian) of Spitsbergen, *Geological Magazine*, 107 (3), 181-195, https://doi.org/10.1017/S0016756800055655, 1970.

Feyling-Hanssen, R.W., and Ulleberg, K.: A Tertiary-Quaternary section at Sarsbukta, Spitsbergen, Svalbard, and its foraminifera, *Polar Research*, 2, 77-106, https://doi.org/10.3402/polar.v2i1.6963, 1984.

Fyhn, M.B.W., and Hopper, J.R.: NE Greenland Composite Tectono-Sedimentary Element, northern Greenland Sea and Fram Strait. *Geological Society, London, Memoirs*, 57, https://doi.org/10.1144/M57-2017-12, 2025.

## G

Gabrielsen, R.H., Kløvjan, O.S., Haugsbø, H., Midbøe, P.S., Nøttvedt, A., Rasmussen, E., and Skott, P.H.: A structural outline of Forlandsundet Graben, Prins Karls Forland, Svalbard, *Norsk Geologisk Tidsskrift*, 72, 105-120, 1992.

Galfetti, T., Hochuli, P.A., Brayard, A., Bucher, H., Weissert, H., and Vigran, J.O.: Smithian-Spathian boundary event: Evidence for global climatic change in the wake of the end-Permian biotic crisis, *Geology*, 35 (4), 291-294, https://doi.org/10.1130/g23117a.1, 2007.

Galloway, J.M., Vickers, M.L., Price, G.D., Poulton, T., Grasby, S.E., Hadlari, T., Beauchamp, B., and Sulphur, K.: Finding the VOICE: organic carbon isotope chemostratigraphy of Late Jurassic – Early Cretaceous Arctic Canada, *Geological Magazine*, 157 (10), 1643-1657, https://doi.org/10.1017/s0016756819001316, 2020.

Galloway, J.M., Fensome, R.A., Swindles, G.T., Hadlari, T., Fath, J., Schröder-Adams, C., Herrle, J.O., and Pugh, A.: Exploring the role of High Arctic Large Igneous Province volcanism on Early Cretaceous Arctic forests, *Cretaceous Research*, 129, 105022, 2022.

Galloway, J.M., Grasby, S.E., Wang, F., Hadlari, T., Dewing, K., Bodin, S., and Sanei, H.: A mercury and trace element geochemical record across Oceanic Anoxic Event 1b in Arctic Canada, *Palaeogeography, Palaeoclimatology, Palaeoecology*, 617, 111490, https://doi.org/10.1016/j.palaeo.2023.111490, 2023.

Galloway, J.M., and Lindström, S.: Impacts of Large-scale Magmatism on Land Plant Ecosystems, *Elements*, 19, 289-295, 2023a.

Galloway, J.M., and Lindström, S.: Wildfire in the geological record: Application of Quaternary methods to deep time studies, *Evolving Earth*, 1, 100025, 2023b.





Galloway, J. M., Hadlari, T., Dewing, K., Poulton, T., Grasby, S. E., Reinhardt, L., Rogov, M., Longman, J., and Vickers, M.: The silent VOICE—Searching for geochemical markers to track the impact of Late Jurassic rift tectonics. *Geochemistry, Geophysics, Geosystems*, *25*(10), e2024GC011490, 2024.

Gastaldo, R.A., DiMichele, W.A., and Pfefferkorn, H.W.: Out of the icehouse into the greenhouse: a late Paleozoic analogue for modern global vegetational change, *GSA Today*, 6 (10), 1-7, 1996.

Gee, D.G., Bogolepova, O.K., and Lorenz, H.: The Timanide, Caledonide and Uralide orogens in the Eurasian high Arctic, and relationships to the palaeo-continents Laurentia, Baltica and Siberia, *Geological Society, London, Memoirs*, 32 (1), 507-520, https://doi.org/10.1144/GSL.MEM.2006.032.01.31, 2006.

Gee, D.G., and Teben'kov, A.: Svalbard: a fragment of the Laurentian margin, *Geological Society, London, Memoirs*, 30 (1), 191-206, 2004.

Gensel, P.G.: The earliest land plants. *Annual Review of Ecology, Evolution, and Systematics,* 39: 459-477, 2008.

Gilmullina, A., Klausen, T.G., Doré, A.G., Rossi, V.M., Suslova, A., Eide, C.H.: Linking sediment supply variations and tectonic evolution in deep time, source-to-sink systems - The Triassic Greater Barents Sea Basin.
*GSA Bulletin* 134(7-8), 1760-1780. 2022.

Gion, A. M., Williams, S. E., and Müller, R. D.: A reconstruction of the Eurekan Orogeny incorporating deformation constraints. *Tectonics,* 36(2), 304-320, 2017.

Gjelberg, J. and Steel, R.: An outline of Lower-Middle Carboniferous sedimentation on Svalbard: Effects of tectonic, climatic and sea level changes in rift basin sequences, *Geology of the North Atlantic Borderlands —*
*Memoir 7,* 543-561, 1981.

Gjelberg, J. and Steel, R.J.: Helvetiafjellet Formation (Barremian-Aptian), Spitsbergen: characteristics of a transgressive succession, *Norwegian Petroleum Society Special Publications. Elsevier*, 571-593 https://doi.org/10.1016/S0928-8937(06)80087-1, 1995.

Glørstad-Clark, E., Faleide, J.I., Lundschien, B.A. and Nystuen, J.P.: Triassic seismic sequence stratigraphy and
paleogeography of the western Barents Sea area. *Marine and Petroleum geology,* 27(7): 1448-1475, https://doi.org/10.1016/j.marpetgeo.2010.02.008, 2010

Gobbett, D.J. and Wilson, C.: The Oslobreen Series, Upper Hecla Hoek of Ny Friesland, Spitsbergen. *Geological Magazine*, 97(6): 441-457, 1960.

Golovneva, L. B.: Palaeogene climates of Spitsbergen. *GFF (Geological Society of Sweden)*, 122(1), 62–
63.https://doi.org/10.1080/11035890001221062, 2000.

Golovneva, L.B., Zolina, A.A. and Spicer, R.A.: The early Paleocene (Danian) climate of Svalbard based on palaeobotanical data. *Papers in Palaeontology,* 9(6): e1533, 2023.

Golovneva, L., and Zolina, A.: The Renardodden flora of Spitsbergen. *Biological Communications*, 68(4), 307–319. https://doi.org/10.21638/spbu03.2023.410, 2023.





Gomes, A.S., Vasconcelos, P.M.: Geochronology of the Parana-Etendeka large igneous province. *Earth Sci. Rev.* 220, 103716. 2021.

Grasby, S.E. and Beauchamp, B.: Intrabasin variability of the carbon-isotope record across the Permian–Triassic transition, Sverdrup Basin, Arctic Canada. *Chemical Geology,* 253(3): 141-150, https://doi.org/10.1016/j.chemgeo.2008.05.005, 2008.

Grasby, S., Sanei, H. & Beauchamp, B.: Catastrophic dispersion of coal fly ash into oceans during the latest Permian extinction. *Nature Geosci* **4**, 104–107. https://doi.org/10.1038/ngeo1069, 2011.

Grasby, S.E., Beauchamp, B., Embry, A. and Sanei, H.: Recurrent Early Triassic Ocean anoxia. *Geology,* 41(2): 175-178, 10.1130/g33599.1, 2013.

Grasby, S.E., Beauchamp, B., Bond, D.P.G., Wignall, P., Talavera, C., Galloway, J.M., Piepjohn, K., Reinhardt, L.
and Blomeier, D.: Progressive environmental deterioration in northwestern Pangea leading to the latest Permian extinction. *GSA Bulletin*, 127(9-10): 1331-1347, 10.1130/b31197.1, 2015a.

Grasby, S.E., Beauchamp, B., Bond, D.P.G., Wignall, P.B. and Sanei, H.: Mercury anomalies associated with three extinction events (Capitanian Crisis, Latest Permian Extinction and the Smithian/Spathian Extinction) in NW Pangea. *Geological Magazine,* 153(2): 285-297, 10.1017/s0016756815000436, 2015b.

Grasby, S.E., Beauchamp, B., Bond, D.P., Wignall, P.B. and Sanei, H.: Mercury anomalies associated with three extinction events (Capitanian crisis, latest Permian extinction and the Smithian/Spathian extinction) in NW Pangea. *Geological magazine,* 153(2): 285-297, 2016a.

Grasby, S.E., Beauchamp, B. and Knies, J.: Early Triassic productivity crises delayed recovery from world's worst mass extinction. *Geology,* 44(9): 779-782, 2016b.

Grasby, S.E., Shen, W., Yin, R., Gleason, J.D., Blum, J.D., Lepak, R.F., Hurley, J.P. and Beauchamp, B.: Isotopic signatures of mercury contamination in latest Permian oceans. *Geology,* 45(1): 55-58. 2017a.

Grasby, S. E., McCune, G. E., Beauchamp, B., and Galloway, J. M.: Lower Cretaceous cold snaps led to widespread glendonite occurrences in the Sverdrup Basin, *Canadian High Arctic. Bulletin*, 129(7-8), 771-787, 2017b.

Grasby, S.E., Knies, J., Beauchamp, B., Bond, D.P.G., Wignall, P. and Sun, Y.: Global warming leads to Early Triassic nutrient stress across northern Pangea. *GSA Bulletin,* 132(5-6): 943-954, 10.1130/b32036.1, 2019a.

Grasby, S.E., Them, T.R., Chen, Z., Yin, R. and Ardakani, O.H.: Mercury as a proxy for volcanic emissions in the geologic record. *Earth-Science Reviews*, 196: 102880, https://doi.org/10.1016/j.earscirev.2019.102880,
2019b.

Grasby, S.E., Liu, X., Yin, R., Ernst, R.E. and Chen, Z.: Toxic mercury pulses into late Permian terrestrial and marine environments. *Geology,* 48(8): 830-833, 10.1130/g47295.1, 2020.



Green, P. F., and Duddy, I. R.: Synchronous exhumation events around the Arctic including examples from Barents Sea and Alaska North Slope. In *Geological Society, London, Petroleum Geology Conference Series* (Vol. 7, No. 1, pp. 633-644). London: The Geological Society of London, 2010.

Green, T., Renne, P.R. and Brenhin Kneller, C.: Continental flood basalts drive Phanerozoic extinctions. *Proceedings of The National Academy of Sciences*, 119 (38), e2120441119, 2022.

Greenwood, D.R., Basinger, J.F. and Smith, R.Y.: How wet was the Arctic Eocene rain forest? Estimates of precipitation from Paleogene Arctic macrofloras. *Geology,* 38(1): 15-18, 10.1130/g30218.1, 2010.

Grossman, E. L., Yancey, T. E., Jones, T. E., Bruckschen, P., Chuvashov, B., Mazzullo, S. J., and Mii, H. S.: Glaciation, aridification, and carbon sequestration in the Permo-Carboniferous: the isotopic record from low latitudes. *Palaeogeography, Palaeoclimatology, Palaeoecology*, 268(3-4), 222-233, 2008.

Grossman, E.L.: Applying Oxygen Isotope Paleothermometry in Deep Time. *The Paleontological Society Papers,* 18: 39-68, 10.1017/S1089332600002540, 2012.

Grotheer, H., Le Métayer, P., Piggott, M., Lindeboom, E., Holman, A., Twitchett, R. and Grice, K.: Occurrence and significance of phytanyl arenes across the Permian-Triassic boundary interval. *Organic Geochemistry,* 104: 42-52, 2017.

Grundvåg, S.-A., Jelby, M.E., Śliwińska, K.K., Nøhr-Hansen, H., Aadland, T., Sandvik, S.E., Tennvassås, I., Engen, T.M. and Olaussen, S.: Sedimentology and palynology of the Lower Cretaceous succession of central Spitsbergen: integration of subsurface and outcrop data. *Norwegian Journal of Geology* https://dx.doi.org/10.17850/njg99-2-02, 2019.

Grundvåg, S.-A. and Olaussen, S.: Sedimentology of the Lower Cretaceous at Kikutodden and Keilhaufjellet, southern Spitsbergen: implications for an onshore–offshore link. *Polar Research,* 36(1): 1302124, 10.1080/17518369.2017.1302124, 2017.

Grundvåg, S.A., Johannessen, E.P.: Helland-Hansen, W. and Plink-Björklund, P., Depositional architecture and evolution of progradationally stacked lobe complexes in the E ocene C entral B asin of S pitsbergen. *Sedimentology,* 61(2): 535-569, https://doi.org/10.1111/sed.12067, 2014.

Grundvåg, S.A., Marin, D., Kairanov, B., Śliwińska, K.K., Nøhr-Hansen, H., Jelby, M.E., Escalona, A. and Olaussen, S.: The Lower Cretaceous succession of the northwestern Barents Shelf: Onshore and offshore correlations. *Marine and Petroleum Geology*, 86: 834-857. https://doi.org/10.1016/j.marpetgeo.2017.06.036, 2017.

Grundvåg, S.A., Helland-Hansen, W., Johannessen, E.P., Eggenhuisen, J., Pohl, F., and Spychala, Y.: Deep-water sand transfer by hyperpycnal flows, the Eocene of Spitsbergen, Arctic Norway. *Sedimentology,* 70(7): 2057-2107, 2023.

Gruszczyński, M., Hałas, S., Hoffman, A. and Małkowski, K.: A brachiopod calcite record of the oceanic carbon and oxygen isotope shifts at the Permian/Triassic transition. *Nature,* 337(6202): 64-68, 1989.




Gröcke, D.R., Price, G.D., Robinson, S.A., Baraboshkin, E.Y., Mutterlose, J. and Ruffell, A.H.: The Upper Valanginian (Early Cretaceous) positive carbon–isotope event recorded in terrestrial plants. *Earth and Planetary Science Letters*, 240(2): 495-509, 2005.

Guarnieri, P.: Pre-break-up palaeostress state along the East Greenland margin. *Journal of the Geological Society*, 172(6): 727-739, 10.1144/jgs2015-053, 2015.

Gudlaugsson, S., Faleide, J., Johansen, S. and Breivik, A.: Late Palaeozoic structural development of the south-western Barents Sea. *Marine and Petroleum Geology,* 15(1): 73-102, https://doi.org/10.1016/S0264-8172(97)00048-2, 1998.

Gussone, N., Ahm, A.-S.C., Lau, K.V. and Bradbury, H.J.: Calcium isotopes in deep time: Potential and limitations. *Chemical Geology,* 544: 119601, https://doi.org/10.1016/j.chemgeo.2020.119601, 2020.

Gutjahr, M., Ridgwell, A., Sexton, P.F., Anagnostou, E., Pearson, P.N., Pälike, H., Norris, R.D., Thomas, E. and Foster, G.L.: Very large release of mostly volcanic carbon during the Palaeocene–Eocene Thermal Maximum. *Nature,* 548(7669): 573-577, 10.1038/nature23646, 2017.


**H**

Haaland, L.C., Slagstad, T., Osmundsen, P.T., Redfield, T.: U-Pb calcite ages date oblique rifting of the Arctic-North Atlantic gateway. *Geology,* 52(8), 615-619, 2024.

Hansen, B. B., Bucher, H. F., Schneebeli-Hermann, E., and Hammer, Ø.: Smithian and Spathian palaeontological
records of the Vikinghøgda Formation in Central Spitsbergen. *Lethaia,* 57(1), 1-15, 2024.

Hallam, A.: A review of Mesozoic climates. *Journal of the Geological Society*, 142(3): 433-445, 1985.

Halverson, G.P., Maloof, A.C. and Hoffman, P.F.: The Marinoan glaciation (Neoproterozoic) in northeast Svalbard. *Basin Research,* 16(3): 297-324, 2004.

Hambrey, M.J.: Late Precambrian diamictites of northeastern Svalbard. *Geological Magazine,* 119(6): 527-551,
10.1017/S0016756800027035, 1982.

Hammer, Ø., Nakrem, H.A., Little, C.T., Hryniewicz, K., Sandy, M.R., Hurum, J.H., Druckenmiller, P., Knutsen, E.M. and Høyberget, M.: Hydrocarbon seeps from close to the Jurassic–Cretaceous boundary, Svalbard. *Palaeogeography, Palaeoclimatology, Palaeoecology,* 306(1-2): 15-26, 2011.

Hammer, Ø., Collignon, M. and Nakrem, H.A.: Organic carbon isotope chemostratigraphy and cyclostratigraphy in
the Volgian of Svalbard. *Norwegian Journal of Geology*, 92, 2012.

Hammer, Ø., Jones, M.T., Schneebeli-Hermann, E., Hansen, B.B. and Bucher, H.: Are Early Triassic extinction events associated with mercury anomalies? A reassessment of the Smithian/Spathian boundary extinction. *Earth-Science Reviews,* 195: 179-190, https://doi.org/10.1016/j.earscirev.2019.04.016, 2019.



Hanken, N.-M. and Nielsen, J.K.: Upper Carboniferous–Lower Permian Palaeoaplysina build-ups on Svalbard: the influence of climate, salinity and sea-level. Geological Society, London, *Special Publications*, 376(1): 269-305, https://doi.org/10.1144/SP376.17, 2013.

Hansen, J. and Holmer, L.E.: Diversity fluctuations and biogeography of Ordovician brachiopod faunas in northeastern Spitsbergen. *Bulletin of Geosciences,* 85(3): 497-504, 2010.

Hansen, B. B., Bucher, H. F., Schneebeli-Hermann, E., and Hammer, Ø.: Smithian and Spathian palaeontological records of the Vikinghøgda Formation in Central Spitsbergen. *Lethaia,* 57(1), 1-15, 2024

Haq, B.U.: Cretaceous eustasy revisited. I 113: 44-58, 10.1016/j.gloplacha.2013.12.007, 2014.

Harding, I.C., Charles, A.J., Marshall, J.E.A., Pälike, H., Roberts, A.P., Wilson, P.A., Jarvis, E., Thorne, R., Morris, E., Moremon, R., Pearce, R.B. and Akbari, S.: Sea-level and salinity fluctuations during the Paleocene–Eocene thermal maximum in Arctic Spitsbergen. *Earth and Planetary Science Letters*, 303(1): 97-107, https://doi.org/10.1016/j.epsl.2010.12.043, 2011.

Harland, W.B. and Wilson, C.: The Hecla Hoek succession in Ny Friesland, Spitsbergen. *Geological Magazine*, 93(4): 265-286, 1956.

Harland, W.B., Cutbill, J., Friend, P.F., Gobbett, D.J., Holliday, D., Maton, P., Parker, J. and Wallis, R.H.: The Billefjorden Fault Zone, Spitsbergen: the long history of a major tectonic lineament, Nor. Polarinst. Skr., 161, 1974.

Harland, W., Herod, K., Wright, A. and Moseley, F.: Ice Ages: Ancient and Modern. *Seel House Press Liverpool,* 189-126, 1975a.

Harland, W.B., Pickton, C.A. and Reynolds, A.B.: Cambridge Svalbard Expedition, 1974. *Polar Record,* 17(109): 383-384, 10.1017/S0032247400032204, 1975b.

Harland, W. and Wright, N.: Alternative hypothesis for the pre-Carboniferous evolution of Svalbard. *Norsk Polarinstitutt Skrifter,* 167: 89-117, 1979.

Harland, W.B.: Svalbard. W.B. Harland (Ed.): Geology of Svalbard, *Geological Society, London, Memoirs* 1997.

Harper, D.T., Suarez, M.B., Uglesich, J., You, H., Li, D. and Dodson, P.: Aptian–Albian clumped isotopes from northwest China: cool temperatures, variable atmospheric pCO2 and regional shifts in the hydrologic cycle. *Climate of the Past,* 17(4), pp.1607-1625, 2021.

Hatleberg, E.: Lower Triassic conodonts and biofacies interpretations: Nepal and Svalbard. *Geologica et Palaeontologica*, 18: 101-125, 1984.

Hauser, N., Reimold, W.U., Cavosie, A.J., Crósta, A.P., Schwarz, W.H., Trieloff, M., Da Silva Maia de Souza, C., Pereira, L.A., Rodrigues, E.N. and Brown, M.: Linking shock textures revealed by BSE, CL, and EBSD with U-Pb data (LA-ICP-MS and SIMS) from zircon from the Araguainha impact structure, Brazil. *Meteoritics and Planetary Science,* 54(10): 2286-2311, https://doi.org/10.1111/maps.13371, 2019.



Head, M.: A palynological investigation of Tertiary strata at Renardodden, West Spitsbergen, *6th International Palynological Conference*, Abstract, pp. 61, 1984.

Henderson, C.M., Baud, A.: Correlation of the Permian-Triassic boundary in Arctic Canada and comparison with Meishan, China. Proceedings, *30th International Geological Congress*. 11,143–152, 1997.

Helland-Hansen, W.: Sedimentation in Paleogene Foreland Basin, Spitsbergen1. *AAPG Bulletin,* 74(3): 260-272, 10.1306/0c9b22bd-1710-11d7-8645000102c1865d, 1990.

Helland-Hansen, W. and Grundvåg, S.A.: The Svalbard Eocene-Oligocene (?) Central Basin succession: Sedimentation patterns and controls. *Basin Research,* 33(1): 729-753, 10.1111/bre.12492, 2021.

Henderson, C.M., Baud, A.: Correlation of the Permian-Triassic boundary in Arctic Canada and comparison with Meishan, China. Proceedings, *30th International Geological Congress*. 11, 143–152, 1997.

Henriksen, E., Bjørnseth, H., Hals, T., Heide, T., Kiryukhina, T., Kløvjan, O., Larssen, G.B., Ryseth, A., Rønning, K. and Sollid, K.: Chapter 17 Uplift and erosion of the greater Barents Sea: impact on prospectivity and petroleum systems. *Geological Society, London, Memoirs*, 35(1): 271-281, 2011a.

Henriksen, E., Ryseth, A.E., Larssen, G.B., Heide, T., Rønning, K., Sollid, K. and Stoupakova, A.V.: Chapter 10 Tectonostratigraphy of the greater Barents Sea: implications for petroleum systems. *Geological Society, London, Memoirs*, 35(1): 163-195, 10.1144/M35.10, 2011b.

Herrle, J.O., Schröder-Adams, C.J., Davis, W., Pugh, A.T., Galloway, J.M. and Fath, J.: Mid-Cretaceous High Arctic stratigraphy, climate, and oceanic anoxic events. *Geology,* 43(5): 403-406, 2015.

Hesselbo, S.P., Gröcke, D.R., Jenkyns, H.C., Bjerrum, C.J., Farrimond, P., Morgans Bell, H.S. and Green, O.R.: Massive dissociation of gas hydrate during a Jurassic oceanic anoxic event. *Nature,* 406(6794): 392-395, 2000.

Hesselbo, S.P., Jenkyns, H.C., Duarte, L.V. and Oliveira, L.C.: Carbon-isotope record of the Early Jurassic (Toarcian) Oceanic Anoxic Event from fossil wood and marine carbonate (Lusitanian Basin, Portugal). *Earth and Planetary Science Letters,* 253(3-4): 455-470, https://doi.org/10.1016/j.epsl.2006.11.009, 2007.

Heyn, B. H., Shephard, G. E., and Conrad, C. P.: Prolonged multi-phase magmatism due to plume-lithosphere interaction as applied to the High Arctic Large Igneous Province. *Geochemistry, Geophysics, Geosystems*, 25, e2023GC011380. https://doi.org/10.1029/2023GC011380, 2024.

Hjálmarsdóttir, H.R., Hammer, Ø., Nagy, J. and Grundvåg, S.-A.: Foraminiferal stratigraphy and palaeoenvironment of a storm-influenced marine shelf: Upper Aptian–lower Albian, Svalbard, Arctic Norway. *Cretaceous Research,* 130: 105033, 2022.

Hjelstuen, B. O., Elverhøi, A., and Faleide, J. I.: Cenozoic erosion and sediment yield in the drainage area of the Storfjorden Fan. *Global and Planetary Change*, *12*(1-4), 95-117, 1996.



Hochuli, P.A., Hermann, E., Vigran, J.O., Bucher, H. and Weissert, H.: Rapid demise and recovery of plant ecosystems across the end-Permian extinction event. Global and Planetary Change, 74(3): 144-155, https://doi.org/10.1016/j.gloplacha.2010.10.004, 2010.

Hoel, A. and Orvin, A.K.: 1937. Das Festungsprofil auf Spitzbergen. I, Karbon-Kreide: Vermessungsresultate.

Hoffman, P.F. and Schrag, D.P.: The snowball Earth hypothesis: testing the limits of global change. *Terra nova,* 2065 14(3): 129-155, 2002.

Hofmann, R., Goudemand, N., Wasmer, M., Bucher, H., and Hautmann, M.: New trace fossil evidence for an early recovery signal in the aftermath of the end-Permian mass extinction. *Palaeogeography, Palaeoclimatology, Palaeoecology*, *310*(3-4), 216-226, 2011.

Hofmann, R., Hautmann, M., and Bucher, H.: A new paleoecological look at the Dinwoody Formation (Lower 2070 Triassic, Western USA): intrinsic versus extrinsic controls on ecosystem recovery after the end-Permian mass extinction. *Journal of Paleontology,* 87(5), 854-880, 2013.

Hofmann, R., Hautmann, M., Brayard, A., Nützel, A., Bylund, K. G., Jenks, J. F., Vennin, E., Oliver, N., and Bucher, H., Recovery of benthic marine communities from the end-P ermian mass extinction at the low latitudes of eastern Panthalassa. *Palaeontology*, 57(3), 547-589, 2014.

Hofmann, R., Hautmann, M., and Bucher, H., Recovery dynamics of benthic marine communities from the Lower Triassic Werfen Formation, northern Italy. *Lethaia,* 48(4), 474-496, 2015.

Holland, M.M. and Bitz, C.M.: Polar amplification of climate change in coupled models. *Climate dynamics*, 21(3-4): 221-232, 10.1007/s00382-003-0332-6, 2003.

Holliday, D. and Cutbill, J.: The Ebbadalen Formation (Carboniferous), Spitsbergen. *Proceedings of the Yorkshire* 2080 *Geological Society*, 39(1): 1-32, 1972.

Holmden, C., Creaser, R.A., Muehlenbachs, K., Leslie, S.A. and Bergström, S.M.: Isotopic evidence for geochemical decoupling between ancient epeiric seas and bordering oceans: Implications for secular curves. *Geology,* 26(6): 567-570, 10.1130/0091-7613(1998)026, 2.3.Co;2, 1998.

Hönisch, B., Ridgwell, A., Schmidt, D., Thomas, E., Gibbs, S., Sluijs, A., Zeebe, R., Kump, L., Martindale, R.C., 2085 Greene, S.E., Kiessling, W., Ries, J., Zachos, J.C., Royer, D., Barker, S., Marchitto, T.M.Jr., Moyer, R., Pelejero, C., Ziveri, P., Foster, G., Williams, B.: The Geological Record of Ocean Acidification. *Science,* 335(6072): 1058-1063, 2012.

Horota, R.K., Rossa, P., Marques, A., Gonzaga, L., Senger, K., Cazarin, C.L., Spigolon, A. and Veronez, M.R.: An Immersive Virtual Field Experience Structuring Method for Geoscience Education. *IEEE Transactions on* 2090 *Learning Technologies*, 16(1): 121-132, 2022.

Hochuli, P.A., Schneebeli-Hermann, E., Mangerud, G. and Bucher, H.: Evidence for atmospheric pollution across the Permian-Triassic transition. *Geology,* 45(12): 1123-1126, 2017.



Hovikoski, J., Fyhn, M.B.W., Nøhr-Hansen, H., Hopper, J.R., Andrews, S., Barham, M., Nielsen, L.H., Bjerager, M., Bojesen-Koefoed, J., Lode, S., Sheldon, E., Uchman, A., Skorstengaard, P.R., and Alsen, P.: Paleocene-Eocene volcanic segmentation of the Norwegian-Greenland seaway reorganized high-latitude ocean circulation. *Communications Earth and Environment*, 2: 172, 2021.

Hryniewicz, K., Nakrem, H.A., Hammer, Ø., Little, C.T., Kaim, A., Sandy, M.R. and Hurum, J.H.: The palaeoecology of the latest Jurassic–earliest Cretaceous hydrocarbon seep carbonates from Spitsbergen, Svalbard. *Lethaia,* 48(3): 353-374, 2015.

Hurum, J.H., Milàn, J., Hammer, Ø., Midtkandal, I., Amundsen, H. and Sæther, B.: Tracking polar dinosaurs-new finds from the Lower Cretaceous of Svalbard. *Norwegian Journal of Geology.* 86(4), 2006.

Hurum, J.H., Nakrem, H.A., Hammer, Ø., Knutsen, E.M., Druckenmiller, P.S., Hryniewicz, K. and Novis, L.K.: An Arctic Lagerstätte–the Slottsmøya Member of the Agardhfjellet Formation (Upper Jurassic–Lower Cretaceous) of Spitsbergen. *Norwegian Journal of Geology*, 92, 2012.

Hurum, J.H., Roberts, A.J., Nakrem, H.A., Stenløkk, J.A. and Mørk, A.: The first recovered ichthyosaur from the Middle Triassic of Edgeøya, Svalbard. *Norwegian Petroleum Directorate Bulletin*, 11: 97-110, 2014.

Hurum, J.H., Druckenmiller, P.S., Hammer, Ø., Nakrem, H.A. and Olaussen, S.: The theropod that wasn't: an ornithopod tracksite from the Helvetiafjellet Formation (Lower Cretaceous) of Boltodden, Svalbard. *Geological Society, London, Special Publications,* 434(1): 189-206, https://doi.org/10.1144/SP434.10, 2016a.

Hurum, J.H., Roberts, A.J., Dyke, G.J., Grundvåg, S.-A., Nakrem, H.A., Midtkandal, I., Sliwinska, K. and Olaussen, S.: Bird or maniraptoran dinosaur? A femur from the Albian strata of Spitsbergen. *Palaeontologia Polonica*, 67: 137-47, 2016b.

Hurum, J.H., Engelschiøn, V.S., Økland, I.H., Bratvold, J., Ekeheien, C., Roberts, A.J., Delsett, L.L., Hansen, B.B., Mørk, A. and Nakrem, H.A.: The history of exploration and stratigraphy of the Early to Middle Triassic vertebrate-bearing strata of Svalbard (Sassendalen Group, Spitsbergen*). Norwegian Journal of Geology*, 98(2): 165-174, 2018.

Hutchinson, D.K., Coxall, H.K., Lunt, D.J., Steinthorsdottir, M., De Boer, A.M., Baatsen, M., von der Heydt, A., Huber, M., Kennedy-Asser, A.T. and Kunzmann, L.: The Eocene–Oligocene transition: a review of marine and terrestrial proxy data, models and model–data comparisons. Climate of the Past, 17(1): 269-315, 2021.

Hutchinson, D.K., Coxall, H.K., O'Regan, M., Nilsson, J., Caballero, R. and de Boer, A.M.: Arctic closure as a trigger for Atlantic overturning at the Eocene-Oligocene Transition. *Nature Communications*, 10(1): 3797, 10.1038/s41467-019-11828-z, 2019.

Hüneke, H., Joachimski, M., Buggisch, W. and Lützner, H.: Marine carbonate facies in response to climate and nutrient level: The upper carboniferous and permian of central spitsbergen (Svalbard). *Facies,* 45(1): 93-135, 10.1007/BF02668107, 2001.



Høy, T. and Lundschien, B.: Triassic deltaic sequences in the northern Barents Sea. Geological society, London, memoirs, 35(1): 249-260, https://doi.org/10.1144/M35.15, 2011.

**I**

Inglis, G.N., Bragg, F., Burls, N.J., Cramwinckel, M.J., Evans, D., Foster, G.L., Huber, M., Lunt, D.J., Siler, N. and Steinig, S.: Global mean surface temperature and climate sensitivity of the early Eocene Climatic Optimum (EECO), Paleocene–Eocene Thermal Maximum (PETM), and latest Paleocene. *Climate of the Past*, 16(5): 1953-1968, 10.5194/cp-16-1953-2020, 2020.

Ingólfsson, Ó., and Landvik, J. Y.: The Svalbard–Barents Sea ice-sheet–Historical, current and future perspectives.
*Quaternary Science Reviews*, *64*, 33-60, 2013.

Isbell, J.L., Fraiser, M.L. and Henry, L.C.: Examining the Complexity of Environmental Change during the Late Paleozoic and Early Mesozoic. *PALAIOS,* 23(5): 267-269, 10.2110/palo.2008.S03, 2008.

Ivanov, A.V., He, H., Yan, L., Ryabov, V.V., Shevko, A.Y., Palesskii, S.V. and Nikolaeva, I.V.: Siberian Traps large igneous province: Evidence for two flood basalt pulses around the Permo-Triassic boundary and in the
Middle Triassic, and contemporaneous granitic magmatism. *Earth-Science Reviews,* 122: 58-76, https://doi.org/10.1016/j.earscirev.2013.04.001, 2013.

**J**

Jakobsson, M., Backman, J., Rudels, B., Nycander, J., Frank, M., Mayer, L., Jokat, W., Sangiorgi, F., O'Regan, M., and Brinkhuis, H.: The early Miocene onset of a ventilated circulation regime in the Arctic Ocean, *Nature,*
447, 986–990, 2007.

Janocha, J., Wesenlund, F., Thießen, O., Grundvåg, S.-A., Koehl, J.-B., and Johannessen, E. P.: Petroleum geochemistry of Upper Paleozoic strata on Bjørnøya, western Barents Shelf, *Mar. Petrol. Geol.,* 163, 106768, https://doi.org/10.1016/j.marpetgeo.2024.106768 , 2024.

Jansen, E., Overpeck, J., Briffa, K. R., Duplessy, J.-C., Joos, F., Masson-Delmotte, V., Olago, D., Otto-Bliesner,
B., Peltier, W. R., Rahmstorf, S., Ramesh, R., Raynaud, D., Rind, D., Solomina, O., Villalba, R., and Zhang, D.: *Palaeoclimate, in: Climate Change 2007: The Physical Science Basis. Contribution of Working Group I to the Fourth Assessment Report of the Intergovernmental Panel on Climate Change, edited by: Solomon, S., Qin, D., Manning, M., Chen, Z., Marquis, M., Averyt, K. B., Tignor, M., and Miller, H. L., Cambridge University Press*, Cambridge, United Kingdom and New York, NY, USA, 2007.

Jelby, M. E., Grundvåg, S. A., Helland-Hansen, W., Olaussen, S., and Stemmerik, L.: Tempestite facies variability and storm-depositional processes across a wide ramp: Towards a polygenetic model for hummocky cross-stratification, *Sedimentology*, 67, 742–781, 2020a.



Jelby, M. E., Śliwińska, K. K., Koevoets, M. J., Alsen, P., Vickers, M. L., Olaussen, S., and Stemmerik, L.: Arctic reappraisal of global carbon-cycle dynamics across the Jurassic–Cretaceous boundary and Valanginian Weissert Event, *Palaeogeogr. Palaeoclimatol. Palaeoecol.*, 555, https://doi.org/10.1016/j.palaeo.2020.109847, 2020b.

Jelby, M. E., Grundvåg, S. A., Śliwińska, K. K., Alsen, P., Vickers, M. L., Olaussen, S., and Stemmerik, L.: Lower Cretaceous holostratigraphy in Svalbard: the Arctic key piece of the Boreal basin puzzle, *Geol. Soc. Lond. Spec. Publ.,* 545, SP545-2023, 2025.

Jenkyns, H.: Cretaceous anoxic events: from continents to oceans, J. *Geol. Soc. Lond.,* 137, 171–188, 1980.

Jenkyns, H. C.: Carbon-isotope stratigraphy and paleoceanographic significance of the Lower Cretaceous shallow-water carbonates of Resolution Guyot, Mid-Pacific Mountains, *Proc. Ocean Drill. Program Sci. Results*, 99–104, 1995.

Jenkyns, H. C.: Geochemistry of oceanic anoxic events. *Geochemistry, Geophysics, Geosystems*, *11*(3), 2010.

Jenkyns, H. C., Schouten-Huibers, L., Schouten, S., and Sinninghe Damsté, J. S.: Warm Middle Jurassic-Early Cretaceous high-latitude sea-surface temperatures from the Southern Ocean, *Clim. Past*, 8, 215–226, 2012.

Jin, Y., Wang, Y., Wang, W., Shang, Q., Cao, C., and Erwin, D. H.: Pattern of marine mass extinction near the Permian-Triassic boundary in South China, *Science,* 289, 432–436, 2000.

Jin, X., Tomimatsu, Y., Yin, R., Onoue, T., Franceschi, M., Grasby, S. E., and Rigo, M.: Climax in Wrangellia LIP activity coincident with major Middle Carnian (Late Triassic) climate and biotic changes: Mercury isotope evidence from the Panthalassa pelagic domain, *Earth Planet. Sci. Lett.*, 607, 118075, 2023.

Joachimski, M. M., Alekseev, A. S., Grigoryan, A., and Gatovsky, Y. A.: Siberian Trap volcanism, global warming and the Permian-Triassic mass extinction: New insights from Armenian Permian-Triassic sections, *GSA Bull.*, 132, 427–443, https://doi.org/10.1130/b35108.1, 2019.

Joachimski, M. M., Lai, X., Shen, S., Jiang, H., Luo, G., Chen, B., Chen, J., and Sun, Y.: Climate warming in the latest Permian and the Permian–Triassic mass extinction, *Geology,* 40, 195–198, https://doi.org/10.1130/g32707.1, 2012.

Joachimski, M. M., Alekseev, A. S., Grigoryan, A., and Gatovsky, Y. A.: Siberian Trap volcanism, global warming and the Permian-Triassic mass extinction: New insights from Armenian Permian-Triassic sections, *GSA Bull.*, 132, 427–443, 2020.

Jochmann, M. M., Augland, L. E., Lenz, O., Bieg, G., Haugen, T., Grundvåg, S. A., Jelby, M. E., Midtkandal, I., Dolezych, M., and Hjálmarsdóttir, H. R.: Sylfjellet: a new outcrop of the Paleogene Van Mijenfjorden Group in Svalbard, *Arktos,* 6, 17–38, https://doi.org/10.1007/s41063-019-00072-w, 2020.



Johannessen, E. P., Henningsen, T., Bakke, N. E., Johansen, T. A., Ruud, B. E., Riste, P., Elvebakk, H., Jochmann, M., Elvebakk, G., and Woldengen, M. S.: Palaeogene clinoform succession on Svalbard expressed in outcrops, seismic data, logs and cores, *First Break,* 29, https://doi.org/10.3997/1365-2397.2011004, 2011.

Johannessen, E. P., and Steel, R. J.: Shelf-margin clinoforms and prediction of deepwater sands, *Basin Res.,* 17, 521–550, 2005.

Johannessen, E. P., and Steel, R. J.: Mid-Carboniferous extension and rift-infill sequences in the Billefjorden Trough, Svalbard, *Nor. Geol. Tidsskr.,* 72, 35–48, 1992.

Johansson, Å., Gee, D. G., Larionov, A. N., Ohta, Y., and Tebenkov, A. M.: Grenvillian and Caledonian evolution of eastern Svalbard–a tale of two orogenies, *Terra Nova,* 17, 317–325, 2005.

Johansson, Å., Larionov, A. N., Gee, D. G., Ohta, Y., Tebenkov, A. M., and Sandelin, S.: Grenvillian and Caledonian
tectono-magmatic activity in northeasternmost Svalbard, *Geol. Soc. Lond. Mem.,* 30, 207–232, https://doi.org/10.1144/GSL.MEM.2004.030.01.17, 2004.

Jokat, W., and Herter, U.: Jurassic failed rift system below the Filchner-Ronne-Shelf, Antarctica: New evidence from geophysical data, *Tectonophysics,* 688, 65–83, 2016.

Jones, M. T., Augland, L. E., Shephard, G. E., Burgess, S. D., Eliassen, G. T., Jochmann, M. M., Friis, B., Jerram,
D. A., Planke, S., and Svensen, H. H.: Constraining shifts in North Atlantic plate motions during the Palaeocene by U-Pb dating of Svalbard tephra layers, *Sci. Rep.,* 7, 6822, https://doi.org/10.1038/s41598-017-06170-7, 2017.

Jones, M. T., Jerram, D. A., Svensen, H. H., and Grove, C.: The effects of large igneous provinces on the global carbon and sulphur cycles, *Palaeogeogr. Palaeoclimatol. Palaeoecol.,* 441, 4–21,
https://doi.org/10.1016/j.palaeo.2015.06.042, 2016.

Jones, M. T., Percival, L. M. E., Stokke, E. W., Frieling, J., Mather, T. A., Riber, L., Schubert, B. A., Schultz, B., Tegner, C., Planke, S., and Svensen, H. H.: Mercury anomalies across the Palaeocene–Eocene Thermal Maximum, *Clim. Past,* 15, 217–236, https://doi.org/10.5194/cp-15-217-2019, 2019.

Jones, M. T., Stokke, E. W., Rooney, A. D., Frieling, J., Pogge von Strandmann, P. A. E., Wilson, D. J., Svensen,
H. H., Planke, S., Adatte, T., Thibault, N., Vickers, M. L., Mather, T. A., Tegner, C., Zuchuat, Z., and Schultz, B. P.: Tracing North Atlantic volcanism and seaway connectivity across the Paleocene-Eocene Thermal Maximum (PETM), *Clim. Past,* 19, 1623–1652, https://doi.org/10.5194/cp-19-1623-2023, 2023.

## K

Kanat, L., and Morris, A.: A working hypothesis for central western Oscar II Land, Spitsbergen, *Nor. Polarinst. Skr.,* 190, 1988.

Kasbohm, J., Schoene, B., and Burgess, S.: Radiometric constraints on the timing, tempo, and effects of large igneous province emplacement, in: Large Igneous Provinces: A Driver of Global Environmental and Biotic



Changes, edited by: Ernst, R. E., Dickson, A. J., and Bekker, A., *Am. Geophys. Union, John Wiley Sons*, Inc., 27–82, 2021.

Kear, B. P., Engelschiøn, V. S., Hammer, Ø., Roberts, A. J., and Hurum, J. H.: Earliest Triassic ichthyosaur fossils push back oceanic reptile origins, *Curr. Biol.,* 33, R178–R179, 2023.

Kele, S., Breitenbach, S. F., Capezzuoli, E., Meckler, A. N., Ziegler, M., Millan, I. M., Kluge, T., Deák, J., Hanselmann, K., John, C. M., and Yan, H.: Temperature dependence of oxygen-and clumped isotope fractionation in carbonates: a study of travertines and tufas in the 6–95°C temperature range, *Geochim. Cosmochim. Acta,* 168, 172–192, 2015.

Kellogg, H. E.: Tertiary stratigraphy and tectonism in Svalbard and continental drift, *AAPG Bull.,* 59, 465–485, 1975.

Kenrick, P., Wellman, C. H., Schneider, H., and Edgecombe, G. D.: A timeline for terrestrialization: consequences for the carbon cycle in the Palaeozoic, *Philos. Trans. R. Soc. B-Biol. Sci.,* 367, 519–536, 2012.

Kidder, D. L., and Worsley, T. R.: Causes and consequences of extreme Permo-Triassic warming to globally equable climate and relation to the Permo-Triassic extinction and recovery*, Palaeogeogr. Palaeoclimatol. Palaeoecol.*, 203, 207–237, https://doi.org/10.1016/S0031-0182(03)00667-9, 2004.

Kierulf, H. P., Kohler, J., Boy, J. P., Geyman, E. C., Mémin, A., Omang, O. C., and Steffen, R.: Time-varying uplift in Svalbard—an effect of glacial changes, *Geophys. J.* Int., 231, 1518–1534, 2022.

Kim, S.-T., and O'Neil, J. R.: Equilibrium and nonequilibrium oxygen isotope effects in synthetic carbonates, *Geochim. Cosmochim. Acta,* 61, 3461–3475, 1997.

Kingsbury, C. G., Kamo, S. L., Ernst, R. E., Söderlund, U., and Cousens, B. L.: U-Pb geochronology of the plumbing system associated with the Late Cretaceous Strand Fiord Formation, Axel Heiberg Island, Canada: part of the 130-90 Ma High Arctic large igneous province, *J. Geodyn.,* 118, 106–117, 2018.

Klausen, T. G., Müller, R., Slama, J., and Helland-Hansen, W.: Evidence for Late Triassic provenance areas and Early Jurassic sediment supply turnover in the Barents Sea Basin of northern Pangea, *Lithosphere,* 9, 14–28, 2017.

Klausen, T. G., Nyberg, B., and Helland-Hansen, W.: The largest delta plain in Earth's history, *Geology*, 47, 470–474, https://doi.org/10.1130/G45507.1, 2019.

Klausen, T. G., Paterson, N. W., and Benton, M. J.: Geological control on dinosaurs' rise to dominance: Late Triassic ecosystem stress by relative sea level change, *Terra Nova,* 32, 434–441, https://doi.org/10.1111/ter.12480, 2020.

Kleinspehn, K. L., and Teyssier, C.: Oblique rifting and the Late Eocene–Oligocene demise of Laurasia with inception of Molloy Ridge: Deformation of Forlandsundet Basin, Svalbard, *Tectonophysics,* 693, 363–377, 2016.

Knag, G.: Gipshuken-og Kapp Starostin formasjonen, mellom til øvre Perm, langs vestkysten av Svalbard*, Cand. real., Univ. i Bergen*, 1980.



Knoll, A. H., and Swett, K.: Micropaleontology across the Precambrian—Cambrian boundary in Spitsbergen, *J. Paleontol.*, 61, 898–926, https://doi.org/10.1017/S0022336000029292, 1987.

Koevoets, M. J., Abay, T. B., Hammer, Ø., and Olaussen, S.: High-resolution organic carbon–isotope stratigraphy of the Middle Jurassic–Lower Cretaceous Agardhfjellet Formation of central Spitsbergen, Svalbard, *Palaeogeogr. Palaeoclimatol. Palaeoecol.*, 449, 266–274, https://doi.org/10.1016/j.palaeo.2016.02.029, 2016.

Koevoets, M. J., Hammer, Ø., Olaussen, S., Senger, K., and Smelror, M.: Integrating subsurface and outcrop data
of the Middle Jurassic to Lower Cretaceous Agardhfjellet Formation in central Spitsbergen, *Nor. Geol. Tidsskr.*, 98, https://doi.org/10.17850/njg98-4-01, 2018.

Koevoets, M., Hammer, Ø., and Little, C. T.: Palaeoecology and palaeoenvironments of the Middle Jurassic to lowermost Cretaceous Agardhfjellet Formation (Bathonian-Ryazanian), Spitsbergen, Svalbard, *Nor. J. Geol.,* 99, https://doi.org/10.17850/njg99-1-02, 2019.

Korte, C., Jasper, T., Kozur, H. W., and Veizer, J.: δ18O and δ13C of Permian brachiopods: a record of seawater evolution and continental glaciation, *Palaeogeogr. Palaeoclimatol. Palaeoecol.,* 224, 333–351, 2005.

Korte, C., Jasper, T., Kozur, H. W., and Veizer, J.: 87Sr/86Sr record of Permian seawater, *Palaeogeogr. Palaeoclimatol. Palaeoecol.*, 240, 89–107, https://doi.org/10.1016/j.palaeo.2006.03.047, 2006.

Korte, C., and Kozur, H. W.: Carbon-isotope stratigraphy across the Permian–Triassic boundary: A review*, J. Asian
Earth Sci.,* 39, 215–235, https://doi.org/10.1016/j.jseaes.2010.01.005, 2010.

Krajewski, K. P.: The Botneheia Formation (Middle Triassic) in Edgeøya and Barentsøya, Svalbard: lithostratigraphy, facies, phosphogenesis, paleoenvironment, *Pol. Polar Res.,* 29, 319–364, 2008.

Krajewski, K. P.: Organic matter–apatite–pyrite relationships in the Botneheia Formation (Middle Triassic) of eastern Svalbard: Relevance to the formation of petroleum source rocks in the NW Barents Sea shelf, *Mar.
Petrol. Geol.,* 45, 69–105, 2013.

Kristoffersen, Y.: On the tectonic evolution and paleoceanographic significance of the Fram Strait gateway, in: *Geological history of the polar oceans: Arctic versus Antarctic, edited by: Bleil, U., and Thiede, J., Springer, Dordrecht*, 63–76, 1990.

Kristoffersen, Y., and Husebye, E. S.: Multi-channel seismic reflection measurements in the Eurasian Basin, Arctic
Ocean, from US ice station FRAM-IV, *Tectonophysics,* 114, 103–115, 1985.

Kristoffersen, Y., Ohta, Y., and Hall, J. K.: On the origin of the Yermak Plateau north Svalbard, Arctic Ocean, *Nor. J. Geol.,* 100, 1–33, 2020.

Kröger, B., Finnegan, S., Franeck, F., and Hopkins, M. J.: The Ordovician Succession Adjacent to Hinlopenstretet, Ny Friesland, Spitsbergen, *Am. Mus. Novit.*, 3882, 1–28, 2017.

Kump, L. R., Bralower, T. J., and Ridgwell, A.: Ocean acidification in deep time, *Oceanography*, 22, 94–107, 2009.

Kvaček, Z., and Manum, S. B.: Ferns of the Spitsbergen Palaeogene, Palaeontogr. Abt. B, 230, 169–181, 1993.



Kvaček, Z., Manum, S. B., and Boulter, M. C.: Angiosperms from the Palaeogene of Spitsbergen, including an unfinished work by A. G. Nathorst, *Palaeontogr.* Abt. B, 232, 103–128, 1994.

**L**


Labandeira, C. C.: The four phases of plant-arthropod associations in deep time, *Geol. Acta*, 4, 409–438, 2006.

Large, D. J. and Marshall, C.: Use of carbon accumulation rates to estimate the duration of coal seams and the influence of atmospheric dust deposition on coal composition, *Geol. Soc. Lond. Spec. Publ.,* 404, 303–315, 2015.

Large, D. J., Marshall, C., Jochmann, M., Jensen, M., Spiro, B. F., and Olaussen, S.: Time, Hydrologic Landscape, and the Long-Term Storage of Peatland Carbon in Sedimentary Basins*, J. Geophys. Res.-Earth Surf.*, 126, e2020JF005762, 2021.

Lasabuda, A., Laberg, J. S., Knutsen, S. M., and Safronova, P.: Cenozoic tectonostratigraphy and pre-glacial erosion: A mass-balance study of the northwestern Barents Sea margin, Norwegian Arctic, *J. Geodyn.*,
119, 149–166, 2018.

Lasabuda, A. P. E., Johansen, N. S., Laberg, J. S., Faleide, J. I., Senger, K., Rydningen, T. A., Patton, H., Knutsen, S.-M., and Hanssen, A.: Cenozoic uplift and erosion of the Norwegian Barents Shelf – A review, *Earth-Sci. Rev.,* 217, 103609, https://doi.org/10.1016/j.earscirev.2021.103609, 2021.

Lauritzen, O. and Worsley, D.: Observations of the Upper Palaeozoic stratigraphy of the Ny Friesland area*, Norsk
Polarinstitutt* A51, 41 pp., 1975.

Lawver, L., Müller, R., Srivastava, S., and Roest, W.: The opening of the Arctic Ocean, in: *Geological history of the polar oceans: Arctic versus Antarctic, edited by: Bleil, U. and Thiede, J., Springer, Dordrecht*, 29–62, 1990a.

Lawver, L., Scotese, C., Grantz, A., Johnson, L., and Sweeney, J.: A review of tectonic models for the evolution of
the Canada Basin, in: The Geology of North America, edited by: Grantz, A., Johnson, G. L., and Sweeney, J. F., *Geological Society of America, Boulder*, CO, 593–618, 1990b.

Lee, C., Love, G. D., Hopkins, M. J., Kröger, B., Franeck, F., and Finnegan, S.: Lipid biomarker and stable isotopic profiles through Early-Middle Ordovician carbonates from Spitsbergen, Norway, *Org. Geochem.,* 131, 5–18, https://doi.org/10.1016/j.orggeochem.2019.02.008, 2019.

Lee, S., Shi, G. R., Nakrem, H. A., Woo, J., and Tazawa, J.-I.: Mass extinction or extirpation: Permian biotic turnovers in the northwestern margin of Pangea, *GSA Bull.*, 134, 2399–2414, https://doi.org/10.1130/b36227.1, 2022.





Leever, K. A., Gabrielsen, R. H., Faleide, J. I., and Braathen, A.: A transpressional origin for the West Spitsbergen fold-and-thrust belt: Insight from analog modelling, *Tectonics*, 30, TC2014, https://doi.org/10.1029/2010TC002753, 2011.

Lehnert, O., Stouge, S., and Brandl, P. A.: Conodont biostratigraphy in the Early to Middle Ordovician strata of the Oslobreen Group in Ny Friesland, Svalbard, *Z. Dtsch. Ges. Geowiss.*, 164, 149–172, 2013.

Leu, M., Schneebeli-Hermann, E., Hammer, Ø., Lindemann, F. J., and Bucher, H.: Spatiotemporal dynamics of nektonic biodiversity and vegetation shifts during the Smithian–Spathian transition: conodont and palynomorph insights from Svalbard, *Lethaia,* 57, 1–19, 2024.

Lewis, S. L., and Maslin, M. A.: Defining the anthropocene, *Nature,* 519, 171–180, https://doi.org/10.1038/nature14258, 2015.

Lini, A., Weissert, H., and Erba, E.: The Valanginian carbon isotope event: a first episode of greenhouse climate conditions during the Cretaceous, *Terra Nova,* 4, 374–384, https://doi.org/10.1111/j.1365-3121.1992.tb00826.x, 1992.

Lipinski, M., Warning, B., & Brumsack, H. J.: Trace metal signatures of Jurassic/Cretaceous black shales from the Norwegian Shelf and the Barents Sea. *Palaeogeography, Palaeoclimatology, Palaeoecology*, *190*, 459-475, https://doi.org/10.1016/S0031-0182(02)00619-3, 2003.

Liu, Z., Pagani, M., Zinniker, D., DeConto, R., Huber, M., Brinkhuis, H., Shah, S. R., Leckie, R. M., and Pearson, A.: Global Cooling During the Eocene-Oligocene Climate Transition, *Science,* 323, 1187–1190, https://doi.org/10.1126/science.1166368, 2009.

Lopes, G., Mangerud, G., and Clayton, G.: The palynostratigraphy of the Mississippian Birger Johnsonfjellet section, Spitsbergen, Svalbard, *Palynology*, 43, 631–649, 2019.

Lord, G. S., Johansen, S. K., Støen, S. J., and Mørk, A.: Facies development of the Upper Triassic succession on Barentsøya, Wilhelmøya and NE Spitsbergen, Svalbard, *Nor. J. Geol.*, 97, https://doi.org/10.17850/njg97-1-03, 2017.

Lord, G. S., Mørk, A., Haugen, T., Boxaspen, M. A., Husteli, B., Forsberg, C. S., and Olaussen, S.: Stratigraphy and palaeosol profiles of the Upper Triassic Isfjorden Member, Svalbard, *Nor. J. Geol.,* 2022.

Lourens, L. J., Sluijs, A., Kroon, D., Zachos, J. C., Thomas, E., Röhl, U., Bowles, J., and Raffi, I.: Astronomical pacing of late Palaeocene to early Eocene global warming events, *Nature,* 435, 1083–1087, https://doi.org/10.1038/nature03814, 2005.

Ludwig, P.: The marine transgression in the Middle Carboniferous of Brøggerhalvøya (Svalbard), *Polar Res.,* 9, 65–76, https://doi.org/10.3402/polar.v9i1.6779, 1991.



Lundschien, B. A., Høy, T., and Mørk, A.: Triassic hydrocarbon potential in the Northern Barents Sea; integrating Svalbard and stratigraphic core data, *Nor. Pet. Dir. Bull.,* 11, 3–20, 2014.

Luo, G., Yang, H., Algeo, T. J., Hallmann, C., and Xie, S.: Lipid biomarkers for the reconstruction of deep-time environmental conditions, *Earth-Sci. Rev.,* 189, 99–124, https://doi.org/10.1016/j.earscirev.2018.03.005, 2019.

Lüthje, C. J., Milàn, J., and Hurum, J. H.: Paleocene tracks of the mammal pantodont genus Titanoides in coal-bearing strata, Svalbard, Arctic Norway, J. Vertebr. *Paleontol.*, 30, 521–527, 2010.

Lüthje, C. J., Nichols, G., and Jerrett, R.: Sedimentary facies and reconstruction of a transgressive coastal plain with coal formation, Paleocene, Spitsbergen, Arctic Norway, *Nor. J. Geol.*, 100, 2010.

## M

Macdonald, F.A.:. Deep-Time paleoclimate proxies. *AGU Advances,* 1(3), https://doi.org/10.1029/2020av000244, 2020.

Mackey, T.J., Jost, A.B., Creveling, J.R. and Bergmann, K.D.: A Decrease to Low Carbonate Clumped Isotope Temperatures in Cryogenian Strata. *AGU Advances,* 1(3), e2019AV000159, https://doi.org/10.1029/2019AV000159, 2020.

Maher Jr, H. D., Braathen, A., Bergh, S., Dallmann, W., and Harland, W. B.:. Tertiary or Cretaceous age for Spitsbergen's fold-thrust belt on the Barents Shelf. *Tectonics,* 14(6), 1321-1326, https://doi.org/10.1029/95TC01257, 1995.

Maher, Jr., Hays, T., Shuster, R. and Mutrux, J.:. Petrography of Lower Cretaceous sandstones on Spitsbergen. *Polar Research,* 23(2): 147-165, https://doi.org/10.3402/polar.v23i2.6276, 2004.

Maher, J.H.D.: Manifestations of the Cretaceous High Arctic Large Igneous Province in Svalbard. The Journal of Geology, 109(1): 91-104, https://doi.org/10.1086/317960, 2001.

Mahoney, J., Storey, M., Duncan, R., Spencer, K. and Pringle, M.: Geochemistry and age of the Ontong Java Plateau, in: The Mesozoic Pacific: Geology, tectonics, and volcanism, edited by Pringle, M.S., Sager, W.W., Sliter W.V. and Stein, S., Wiley, Washington, D. C: *American Geophysical Union,* 233-261, https://doi.org/10.1029/GM077p0233 1993.

Majka, J. and Kośmińska, K.: Magmatic and metamorphic events recorded within the Southwestern Basement Province of Svalbard. Arktos 3, 5, https://doi.org/10.1007/s41063-017-0034-7, 2017.

Major, H., Nagy, J., Haremo, P., Dallmann, W.K., Andersen, A., Salvigsen, O.: Adventdalen. In: Geological Map Svalbard 1:100 000, *Sheet C9G. Norsk Polarinstitutt,* 1992.



Mangerud, G. and Konieczny, R.: Palynology of the Permian succession of Spitsbergen, Svalbard. *Polar Research*, 12(1): 65-93, https://doi.org/10.3402/polar.v12i1.6704, 1993.

Mangerud, G. and Konieczny, R.M.: Palynological investigations of Permian rocks from Nordaustlandet, Svalbard. *Polar Research*, 9(2), 155-167, https://doi.org/10.3402/polar.v9i2.6788, 1991.

Manum, S.: Some dinoflagellates and hystrichosphaerids from the Lower Tertiary of Spitsbergen. *Nytt Magasin for Botanik* 8: 17-25, 1960.

Manum, S.: Studies in the Tertiary Flora of Spitsbergen: With Notes on Tertiary Floras of Ellesmere Island,
Greenland, and Iceland, a Palynological Investigation. *Norsk Polarinstitutt Skrifter*, 125. Pp. 128 + 21 plates, Norsk Polarinstitutt, Oslo University Press, 127 pages, 1962.

Manum, S.B. and Throndsen, T.:Rank of coal and dispersed organic matter and its geological bearing in the Spitsbergen Tertiary. *Norsk Polarinstitutt Årbok*, 1977: 159-177, 1978.

Manum, S.B. and Throndsen, T.: Age of Tertiary formations on Spitsbergen. *Polar Research*, 4(2): 103-131,
https://doi.org/10.3402/polar.v4i2.6927, 1986.

Marin, D., Escalona, A., Śliwińska, K.K., Nøhr-Hansen, H. and Mordasova, A.:Sequence stratigraphy and lateral variability of Lower Cretaceous clinoforms in the southwestern Barents Sea. *AAPG Bulletin,* 101(9): 1487-1517, https://doi.org/10.1306/10241616010, 2017.

Marshall, C.J.: Palaeogeographic development and economic potential of the coal-bearing Palaeocene Todalen
Member, Spitsbergen, *PhD University of Nottingham*, 360 pp., https://eprints.nottingham.ac.uk/13794/, 2013.

Marshall, C., Large, D. J., Meredith, W., Snape, C. E., Uguna, C., Spiro, B. F., Orheim, A., Jochmann, M., Mokogwu, I., Wang, Y., and Friis, B.: Geochemistry and petrology of Palaeocene coals from Spitsbergen— Part 1: Oil potential and depositional environment. *International Journal of Coal Geology*, *143*, 22-33,
https://doi.org/10.1016/j.coal.2015.03.006, 2015.

Martinez, M., Aguirre-Urreta, B., Dera, G., Lescano, M., Omarini, J., Tunik, M., O'Dogherty, L., Aguado, R., Company, M. and Bodin, S.: Synchrony of carbon cycle fluctuations, volcanism and orbital forcing during the Early Cretaceous. *Earth-Science Reviews,* 239, 104356, https://doi.org/10.1016/j.earscirev.2023.104356 , 2023.

Matysik, M., Stemmerik, L., Olaussen, S. and Brunstad, H.: Diagenesis of spiculites and carbonates in a Permian temperate ramp succession–Tempelfjorden Group, Spitsbergen, Arctic Norway. *Sedimentology,* 65(3): 745-774, https://doi.org/10.1111/sed.12404, 2018.

Matthiessen, J.: Biostratigraphie tertiärer Ablagerungen (Paläozän) Am Van Keulenfjord (Spitzbergen) nach Dinoflagellaten-Zysten. *(Diploma thesis),* Christian-Albrechts-Universität zu Kiel, Kiel, 94 pp., 1986.



Maxwell, E.E. and Kear, B.P.::Triassic ichthyopterygian assemblages of the Svalbard archipelago: a reassessment of taxonomy and distribution. *Journal of the Geological Society of Sweden* 135, 85-94, https://doi.org/10.1080/11035897.2012.759145, 2013.

McArthur, J.M., Mutterlose, J., Price, G.D., Rawson, P.F., Ruffell, A. and Thirlwall, M.F.:Belemnites of Valanginian, Hauterivian and Barremian age: Sr-isotope stratigraphy, composition ($^{87}$Sr/$^{86}$Sr, $\delta^{13}$C, $\delta^{18}$O, Na, Sr, Mg),
and palaeo-oceanography. Palaeogeography, Palaeoclimatology, Palaeoecology, 202(3-4), 253-272, https://doi.org/10.1016/S0031-0182(03)00638-2 2004.McArthur, J. M., Janssen, N. M. M., Reboulet, S., Leng, M. J., Thirlwall, M. F., and Van de Schootbrugge, B.:  Palaeotemperatures, polar ice-volume, and isotope stratigraphy (Mg/Ca, $\delta^{18}$O, $\delta^{13}$C, $^{87}$Sr/$^{86}$Sr): the early cretaceous (Berriasian, Valanginian, Hauterivian). *Palaeogeography, Palaeoclimatology, Palaeoecology*, 248(3-4), 391-430,
https://doi.org/10.1016/j.palaeo.2006.12.0152007, 2007.

McCann Andrew, J.:Deformation of the Old Red Sandstone of NW Spitsbergen; links to the Ellesmerian and Caledonian orogenies. *Geological Society, London, Special Publications,* 180(1): 567-584, https://doi.org/10.1144/GSL.SP.2000.180.01.30, 2000.

McClelland*, W.C., von Gosen*, W., Piepjohn*, K.: Tonian and Silurian magmatism in Nordaustlandet: Svalbard's
place in the Caledonian orogen, in: *Circum-Arctic Structural Events: Tectonic Evolution of the Arctic Margins and Trans-Arctic Links with Adjacent Orogens, edited by: Piepjohn, K., Strauss, J.V., Reinhardt, L. and McClelland, W.C.: Boulder, Colorado, Geological Society of America, Special Paper* 541, 63 – 80, https://doi.org/10.1130/2018.2541(04). 2019.

McDannell, K.T. and Flowers, R.M.:Vestiges of the Ancient: Deep-Time Noble Gas Thermochronology. *Elements,*
16(5): 325-330, https://doi.org/10.2138/gselements.16.5.325, 2020.

McInerney, F. A., and Wing, S. L.: The Paleocene-Eocene Thermal Maximum: a perturbation of carbon cycle, climate, and biosphere with implications for the future. *Annual Review of Earth and Planetary Sciences,* 39: 489-516, https://doi.org/10.1146/annurev-earth-040610-133431, 2011.

McKerrow, W.S., Scotese, C.R. and Brasier, M.D.: Early Cambrian continental reconstructions. *Journal of the*
*Geological Society, London*, 149, 599-606, https://doi.org/10.1144/gsjgs.149.4.0599, 1992.

Meissner, P., Mutterlose, J., and Bodin, S.: Latitudinal temperature trends in the northern hemisphere during the Early Cretaceous (Valanginian–Hauterivian). *Palaeogeography, Palaeoclimatology, Palaeoecology*, 424, 17-39, https://doi.org/10.1016/j.palaeo.2015.02.003, 2015.

Menegatti, A.P., Weissert, H., Brown, R.S., Tyson, R.V., Farrimond, P., Strasser, A. and Caron, M.:High-resolution
$\delta^{13}$C stratigraphy through the early Aptian "Livello Selli" of the Alpine Tethys. *Paleoceanography,* 13(5): 530-545, 1998.

Michelsen, J.K. and Khorasani, G.K.: A regional study on coals from Svalbard: organic facies, maturity and thermal history. *Bulletin de la Société Géologique de France,* 162 (2): 385–397, 1991.




Midtkandal, I. and Nystuen, J.: Depositional architecture of a low-gradient ramp shelf in an epicontinental sea: The
lower Cretaceous of Svalbard. *Basin Research,* 21(5): 655-675, https://doi.org/10.1111/j.1365-2117.2009.00399.x., 2009.

Midtkandal, I., Nystuen, J.P., Nagy, J., Mørk, A.: Lower Cretaceous lithostratigraphy across a regional subaerial
        unconformity in Spitsbergen: the Rurikfjellet and Helvetiafjellet formations. *Nor. J. Geol. 88* (4), 287–304,
        2008.

Midtkandal, I., Svensen, H.H., Planke, S., Corfu, F., Polteau, S., Torsvik, T.H., Faleide, J.I., Grundvåg, S.-A.,
        Selnes, H., Kürschner, W. and Olaussen, S.: The Aptian (Early Cretaceous) oceanic anoxic event (OAE1a)
        in Svalbard, Barents Sea, and the absolute age of the Barremian-Aptian boundary. *Palaeogeography,
        Palaeoclimatology, Palaeoecology*, 463: 126-135, https://doi.org/10.1016/j.palaeo.2016.09.023, 2016.

Mii, H.-s., Grossman, E.L. and Yancey, T.E.: Stable carbon and oxygen isotope shifts in Permian seas of West
Spitsbergen-Global change or diagenetic artifact? *Geology,* 25(3): 227-230, https://doi.org/10.1130/0091-7613(1997)025<0227:SCAOIS>2.3.CO;2, 1997.

Miller, K.G., Kominz, M.A., Browning, J.V., Wright, J.D., Mountain, G.S., Katz, M.E., Sugarman, P.J., Cramer, B.S.,
        Christie-Blick, N. and Pekar, S.F.: The Phanerozoic record of global sea-level change. *Science* 310,1293-1298(2005). https://doi.org/10.1126/science.1116412, 2005.

Mueller, S., Hounslow, M. W., and Kürschner, W. M.:.Integrated stratigraphy and palaeoclimate history of the
        Carnian Pluvial Event in the Boreal realm; new data from the Upper Triassic Kapp Toscana Group in central
        Spitsbergen (Norway). *Journal of the Geological Society*, *173*(1), 186-202, 2016

Müller, R. D., Cannon, J., Qin, X., Watson, R. J., Gurnis, M., Williams, S., Pfaffelmoser, T., Seton, M., Russell, S.
        H. J. ,Zahirovic S.:. GPlates: Building a virtual Earth through deep time. *Geochemistry, Geophysics,*
*Geosystems,* 19, 2243-2261. https://doi.org/10.1029/2018GC007584, 2018.

Minakov, A., Mjelde, R., Faleide, J.I., Flueh, E.R., Dannowski, A. and Keers, H.: Mafic intrusions east of Svalbard
        imaged by active-source seismic tomography. Tectonophysics, 518: 106-118,  2012.

Montañez, I.P. and Poulsen, C.J.: The Late Paleozoic ice age: an evolving paradigm. *Annual Review of Earth and
        Planetary Sciences,* 41: 629-656, 2013.

Montañez, I.P., Tabor, N.J., Niemeier, D., DiMichele, W.A., Frank, T.D., Fielding, C.R., Isbell, J.L., Birgenheier,
        L.P. and Rygel, M.C.: $CO_2$-forced climate and vegetation instability during Late Paleozoic deglaciation.
        *Science,* 315(5808): 87-91, https://doi.org/10.1126/science.1134207, 2007.

Moody-Stuart, M.: High- and low-sinuosity stream deposits, with examples from the Devonian of Spitsbergen.
        *Journal of Sedimentary Research,* 36(4): 1102-1117, https://doi.org/10.1306/74D71609-2B21-11D7-8648000102C1865D, 1966.



Morgans-Bell, H.S., Coe, A.L., Hesselbo, S.P., Jenkyns, H.C., Weedon, G.P., Marshall, J.E.A., Tyson, R.V. and Williams, C.J.: Integrated stratigraphy of the Kimmeridge Clay Formation (Upper Jurassic) based on exposures and boreholes in south Dorset, UK. Geological Magazine, 138(5), 511-539, 2001

Mulrooney, M., Larsen, L., Rismyhr, B., Van Stappen, J., Senger, K., Braathen, A., Mørk, M., Olaussen, S., Cnudde, V. and Ogata, K.: Fluid flow properties of a potential unconventional CO2 storage unit in central Spitsbergen: the Upper Triassic to Middle Jurassic Wilhelmøya Subgroup. *Norwegian Journal of Geology,* 99: 85-116, 2019.

Mulrooney, M.J., Larsen, L., Van Stappen, J., Rismyhr, B., Senger, K., Braathen, A., Olaussen, S., Mørk, M.B.E., Ogata, K. and Cnudde, V.: Fluid flow properties of the Wilhelmøya Subgroup, a potential unconventional CO2 storage unit in central Spitsbergen. *Norwegian Journal of Geology*, 99: 85-116, 2018.

Myhre, P.I., Corfu, F. and Andresen, A.: Caledonian anatexis of Grenvillian crust: a U/Pb study of Albert I Land, NW Svalbard. *Norwegian Journal of Geology*, 89(173): e191, 2008.

Müller, R., Klausen, T.G., Faleide, J.I., Olaussen, S., Eide, C.H. and Suslova, A.:. Linking regional unconformities in the Barents Sea to compression-induced forebulge uplift at the Triassic-Jurassic transition. *Tectonophysics,* 765: 35-51, 2019

Müller, R.D. and Spielhagen, R.F.: Evolution of the Central Tertiary Basin of Spitsbergen: towards a synthesis of sediment and plate tectonic history. *Palaeogeography, Palaeoclimatology, Palaeoecology*, 80(2): 153-172, 1990.

Mørk, A., Knarud, R. and Worsley, D.: Depositional and diagenetic environments of the Triassic and Lower Jurassic succession of Svalbard. Arctic Geology and Geophysics: *Proceedings of the Third International Symposium on Arctic Geology — Memoir 8,* 371-398, 1982.

Mørk, A. and Bjorøy, M.: Mesozoic source rocks on Svalbard, Petroleum Geology of the North European Margin: *Proceedings of the North European Margin Symposium (NEMS'83), organized by the Norwegian Petroleum Society* and held at the Norwegian Institute of Technology (NTH) in Trondheim 9–11 May, 1983. Springer, pp. 371-382, 1984

Mørk, A., Embry, A.F. and Weitschat, W.: Triassic transgressive-regressive cycles in the Sverdrup Basin, Svalbard and the Barents Shelf, Correlation in Hydrocarbon Exploration: *Proceedings of the conference Correlation in Hydrocarbon Exploration organized by the Norwegian Petroleum Society* and held in Bergen, Norway, 3–5 October 1988. Springer, pp. 113-130, 1989.

Mørk, A., Vigran, J.O., Korchinskaya, M.V., Pchelina, T.M., Fefilova, L.A., Vavilov, M.N. and Weitschat, W., a Triassic rocks in Svalbard, the Arctic Soviet islands and the Barents Shelf: bearing on their correlations. *In: Norwegian Petroleum Society Special Publications, edited by:. T.O. Vorren, E. Bergsager, Ø.A. Dahl-Stamnes, E. Holter, B. Johansen, E. Lie and T.B. Lund , Elsevier*, 457-479, https://doi.org/10.1016/B978-0-444-88943-0.50033-2, 1993



Mørk, A., Dallmann, W., Dypvik, H., Johannessen, E., Larssen, G., Nagy, J., Nøttvedt, A., Olaussen, S., Pchelina, T. and Worsley, D.: Mesozoic lithostratigraphy. *Lithostratigraphic lexicon of Svalbard. Upper Palaeozoic to Quaternary bedrock. Review and recommendations for nomenclature use*: 127-214, 1999a.

Mørk, A., Elvebakk, G., Forsberg, A.W., Hounslow, M.W., Nakrem, H.A., Vigran, J.O. and Weitschat, W.: The type section of the Vikinghogda Formation: a new Lower Triassic unit in central and eastern Svalbard. *Polar Research,* 18(1): 51-82, 1999b.


Mørk, A., and Worsley, D.:. Triassic of Svalbard and the Barents shelf. In *NGF Abstracts and proceedings* (Vol. 3, pp. 23-29), 2006.

Mørk, A. and Grundvåg, S.:Festningen-A 300-million-year journey through shoreline exposures of the Carboniferous and Mesozoic in 7 kilometers. *Geological Society of Norway— geological guides* 2020-7,
https://www.geologi.no/images/GeologiskeGuider/Festningen_red.pdf , 2020.

# N

Nabbefeld, B., Grice, K., Twitchett, R. J., Summons, R. E., Hays, L., Böttcher, M. E., and Asif, M.: An integrated
biomarker, isotopic and palaeoenvironmental study through the Late Permian event at Lusitaniadalen, Spitsbergen, *Earth Planet. Sci. Lett.,* 291, 84–96, 2010.

Naber, T. V., Grasby, S. E., Cuthbertson, J. P., Rayner, N., and Tegner, C.: New constraints on the age, geochemistry, and environmental impact of High Arctic Large Igneous Province magmatism: Tracing the
extension of the Alpha Ridge onto Ellesmere Island, Canada, *GSA Bull.,* 133, 1695–1711, 2021.

Nagy, J.: Delta-influenced foraminiferal facies and sequence stratigraphy of Paleocene deposits in Spitsbergen, *Palaeogeogr. Palaeoclimatol. Palaeoecol.*, 222, 161–179, https://doi.org/10.1016/j.palaeo.2005.03.014, 2005.


Nagy, J., and Berge, S. H.: Micropalaeontological evidence of brackish water conditions during deposition of the Knorringfjellet Formation, Late Triassic–Early Jurassic, Spitsbergen, *Polar Res.,* 27, 413–427, https://doi.org/10.1111/j.1751-8369.2007.00038.x, 2008.

Nagy, J., Jargvoll, D., Dypvik, H., Jochmann, M., and Riber, L.: Environmental changes during the Paleocene–
Eocene Thermal Maximum in Spitsbergen as reflected by benthic foraminifera, *Polar Res.,* 32, 19737, https://doi.org/10.3402/polar.v32i0.19737, 2013.



Nagy, J., and Naoroz, M.: Changing depositional environments reflected by foraminifera in a transgressive-regressive sequence of the Lower Cretaceous on Spitsbergen, *Palaeogeogr. Palaeoclimatol. Palaeoecol.*, 511, 144–167, 2018.

Nakrem, H. A., and Mørk, A.: New Early Triassic Bryozoa (Trepostomata) from Spitsbergen, with some remarks on the stratigraphy of the investigated horizons, *Geol. Mag.,* 128, 129–140, 1991.

Nakrem, H. A., Orchard, M. J., Weitschat, W., Hounslow, M. W., Beatty, T. W., and Mørk, A.: Triassic conodonts from Svalbard and their Boreal correlations, *Polar Res.,* 27, 523–539, https://doi.org/10.1111/j.1751-8369.2007.00038.x, 2008.

Nance, R. D., Worsley, T. R., and Moody, J. B.: The supercontinent cycle, *Sci. Am.,* 259, 72–79, 1988.

Nicolaisen, J., Elvebakk, G., Ahokas, J., Bojesen-Koefoed, J., Olaussen, S., Rinna, J., Skeie, J., and Stemmerik, L.: Characterization of upper Palaeozoic organic-rich units in Svalbard: Implications for the petroleum systems of the Norwegian Barents shelf, *J. Petrol. Geol.,* 42, 59–78, 2019.

Nielsen, J. K., Błażejowski, B., Gieszcz, P., and Nielsen, J. K.: Carbon and oxygen isotope records of Permian
brachiopods from relatively low and high palaeolatitudes: climatic seasonality and evaporation, *Geol. Soc. Lond. Spec. Publ.*, 376, 387–406, 2013.

Nøttvedt, A.: Askeladden delta sequence (Paleocene) on Spitsbergen sedimentation and controls on delta formation, *Polar Res.,* 3, 21–48, 1985.

Nøttvedt, A., Cecchi, M., Gjelberg, J., Kristensen, S., Lønøy, A., Rasmussen, A., Rasmussen, E., Skott, P., and
Van Veen, P.: Svalbard-Barents Sea correlation: a short review, *Nor. Pet. Soc. Spec. Publ.,* 2, 363–375, 1993.

## O

Oakey, G. N., and Chalmers, J. A.: A new model for the Paleogene motion of Greenland relative to North America: Plate reconstructions of the Davis Strait and Nares Strait regions between Canada and Greenland, J.
*Geophys. Res.-Sol. Ea.,* 117, https://doi.org/10.1029/2011jb008942, 2012.

Ogata, K., Senger, K., Braathen, A., Tveranger, J., and Olaussen, S.: The importance of natural fractures in a tight reservoir for potential CO2 storage: a case study of the upper Triassic-middle Jurassic Kapp Toscana Group (Spitsbergen, Arctic Norway), *Geol. Soc. Lond. Spec. Publ.*, 374, 395–415, 2014.

Ogata, K., Senger, K., Braathen, A., Tveranger, J., and Olaussen, S.: Fracture systems and mesoscale structural
patterns in the siliciclastic Mesozoic reservoir-caprock succession of the Longyearbyen CO2 Lab project: Implications for geological CO2 sequestration in Central Spitsbergen, Svalbard, *Nor. J. Geol.*, 94, 121–154, 2014b.

Ogata, K., Weert, A., Betlem, P., Birchall, T., and Senger, K.: Shallow and deep subsurface sediment remobilization and intrusion in the Middle Jurassic to Lower Cretaceous Agardhfjellet Formation (Svalbard), *Geosphere,*
19, 801–822, 2023.



Ogden, D. E., and Sleep, N. H.: Explosive eruption of coal and basalt and the end-Permian mass extinction, Proc. Natl. Acad. Sci. USA, 109, 59–62, https://doi.org/10.1073/pnas.1118675109, 2012.

Olaussen, S., Larssen, G. B., Helland-Hansen, W., Johannessen, E. P., Nøttvedt, A., Riis, F., Rismyhr, B., Smelror, M., and Worsley, D.: Mesozoic strata of Kong Karls Land, Svalbard, Norway; a link to the northern Barents Sea basins and platforms, *Nor. J. Geol.,* 98, 2018.

Olaussen, S., Senger, K., Braathen, A., Grundvåg, S.-A., and Mørk, A.: You learn as long as you drill; research synthesis from the Longyearbyen CO2 Laboratory, Svalbard, Norway, *Nor. J. Geol.*, 99, 157–187, 2019.

Olaussen, S., Grundvåg, S.-A., Senger, K., Anell, I., Betlem, P., Braathen, A., Dallmann, W., Jochmann, M., Johannessen, E. P., Lord, G., Mørk, A., Osmundsen, P. T., Smyrak-Sikora, A., and Stemmerik, L.: Svalbard Composite Tectono-Stratigraphic Element, Barents Sea, *Geol. Soc. Lond. Mem.*, 57, https://doi.org/10.1144/M57-2021-36, 2023.

Olempska, E., and Błaszyk, J.: Ostracods from Permian of Spitsbergen, *Pol. Polar Res.*, 3–20, 1996.

Oordt, A. J., Soreghan, G. S., Stemmerik, L., and Hinnov, L. A.: A record of dust deposition in northern, mid-latitude Pangaea during peak icehouse conditions of the late Paleozoic ice age, *J. Sediment. Res.*, 90, 337–363, https://doi.org/10.2110/jsr.2020.15, 2020.

O'Regan, M., Williams, C. J., Frey, K. E., and Jakobsson, M.: A synthesis of the long-term paleoclimatic evolution of the Arctic, *Oceanography*, 24, 66–80, 2011.

## P

Pankhurst, M. J., Stevenson, C. J., and Coldwell, B. C.: Meteorites that produce K-feldspar-rich ejecta blankets correspond to mass extinctions, *J. Geol. Soc. Lond.,* 179, jgs2021-055, https://doi.org/10.1144/jgs2021-055, 2022.

Park, J., Stein, H. J., Hannah, J. L., Georgiev, S. V., Hammer, Ø., and Olaussen, S.: Paleoenvironment in the circum-Arctic region from the Middle Jurassic to Lower Cretaceous: Trace element and stable isotope geochemistry of the Agardhfjellet Formation, Svalbard, *Palaeogeogr. Palaeoclimatol. Palaeoecol.,* 112333, 2024.

Parkinson, I. J., Schaefer, B. F., and Arculus, R. J.: A lower mantle origin for the world's biggest LIP? A high precision Os isotope isochron from Ontong Java Plateau basalts drilled on ODP Leg 192, *Geochim. Cosmochim. Acta,* 66, A580, 2002.

Patterson, W. P., and Walter, L. M.: Depletion of 13C in seawater ΣC02 on modern carbonate platforms: Significance for the carbon isotopic record of carbonates, *Geology*, 22, 885–888, https://doi.org/10.1130/0091-7613(1994)022<0885:Docisc>2.3.Co;2, 1994.



Paterson, N. W., Mangerud, G., Cetean, C. G., Mørk, A., Lord, G. S., Klausen, T. G., and Mørkved, P. T.: A multidisciplinary biofacies characterisation of the Late Triassic (late Carnian–Rhaetian) Kapp Toscana Group on Hopen, Arctic Norway, *Palaeogeogr. Palaeoclimatol. Palaeoecol.,* 464, 16–42, 2016.

Paterson, N. W., and Mangerud, G.: A revised palynozonation for the Middle–Upper Triassic (Anisian–Rhaetian) *Series of the Norwegian Arctic, Geol. Mag.,* 157, 1568–1592, https://doi.org/10.1017/s0016756819000906, 2019.

Payne, J. L., Lehrmann, D. J., Wei, J., Orchard, M. J., Schrag, D. P., and Knoll, A. H.: Large perturbations of the carbon cycle during recovery from the end-Permian extinction, *Science,* 305, 506–509, 2004.

Payne, J. L., and Kump, L. R.: Evidence for recurrent Early Triassic massive volcanism from quantitative interpretation of carbon isotope fluctuations, *Earth Planet. Sci. Lett.,* 256, 264–277, 2007.

Pearson, P. N., and Palmer, M. R.: Atmospheric carbon dioxide concentrations over the past 60 million years, *Nature*, 406, 695–699, 2000.

Penn, J. L., Deutsch, C., Payne, J. L., and Sperling, E. A.: Temperature-dependent hypoxia explains biogeography and severity of end-Permian marine mass extinction, *Science,* 362, eaat1327, 2018.

Peppe, D. J., Royer, D. L., Cariglino, B., Oliver, S. Y., Newman, S., Leight, E., Enikolopov, G., Fernandez-Burgos, M., Herrera, F., and Adams, J. M.: Sensitivity of leaf size and shape to climate: global patterns and paleoclimatic applications, New Phytol., 190, 724–739, 2011.

Percival, L., Tedeschi, L. R., Creaser, R., Bottini, C., Erba, E., Giraud, F., Svensen, H., Savian, J., Trindade, R., and Coccioni, R.: Determining the style and provenance of magmatic activity during the Early Aptian Oceanic Anoxic Event (OAE 1a), *Global Planet. Change*, 200, 103461, 2021.

Petersen, T. G., Thomsen, T. B., Olaussen, S., and Stemmerik, L.: Provenance shifts in an evolving Eurekan foreland basin: the Tertiary Central Basin, Spitsbergen, *J. Geol. Soc. Lond.,* 173, 634–648, 2016.

Petrov, O., Morozov, A., Shokalsky, S., Kashubin, S., Artemieva, I. M., Sobolev, N., Petrov, E., Ernst, R. E., Sergeev, S., and Smelror, M.: Crustal structure and tectonic model of the Arctic region, *Earth-Sci. Rev.*, 154, 29–71, https://doi.org/10.1016/j.earscirev.2015.11.013, 2016.

Pettersson, C. H., Pease, V., and Frei, D.: U–Pb zircon provenance of metasedimentary basement of the Northwestern Terrane, Svalbard: Implications for the Grenvillian–Sveconorwegian orogeny and development of Rodinia, *Precambrian Res.,* 175, 206–220, 2009.

Peucker-Ehrenbrink, B., and Ravizza, G.: The marine osmium isotope record, *Terra Nova,* 12, 205–219, https://doi.org/10.1046/j.1365-3121.2000.00295.x, 2000.

Piepjohn, K.: The Svalbardian-Ellesmerian deformation of the Old Red Sandstone and the pre-Devonian basement in NW Spitsbergen (Svalbard), *Geol. Soc. Lond. Spec. Publ.,* 180, 585–601, https://doi.org/10.1144/GSL.SP.2000.180.01.31, 2000.



Piepjohn, K., Brinkmann, L., Grewing, A., and Kerp, H.: New data on the age of the uppermost ORS and the lowermost post-ORS strata in Dickson Land (Spitsbergen) and implications for the age of the Svalbardian deformation, *Geol. Soc. Lond. Spec. Publ.*, 180, 603–609, https://doi.org/10.1144/GSL.SP.2000.180.01.32, 2000.

Piepjohn, K., and Dallmann, W. K.: Stratigraphy of the uppermost Old Red Sandstone of Svalbard (Mimerdalen Subgroup), *Polar Res.,* 33, 19998, https://doi.org/10.3402/polar.v33.19998, 2014.

Piepjohn, K., and von Gosen, W.: Structural transect through Ellesmere Island (Canadian Arctic): superimposed Palaeozoic Ellesmerian and Cenozoic Eurekan deformation, *Geol. Soc. Lond. Spec. Publ.*, 460, 33–56, 2018.

Piepjohn, K., von Gosen, W., Tessensohn, F., Reinhardt, L., McClelland, W. C., Dallmann, W., Gaedicke, C., and Harrison, J. C.: Tectonic map of the Ellesmerian and Eurekan deformation belts on Svalbard, North Greenland, and the Queen Elizabeth Islands (Canadian Arctic), *Arktos,* 1, 12, https://doi.org/10.1007/s41063-015-0015-7, 2015.

Piepjohn, K., von Gosen, W., and Tessensohn, F.: The Eurekan deformation in the Arctic: an outline, *J. Geol. Soc.*
*Lond.*, 173, 1007–1024, 2016.

Pietsch, C., and Bottjer, D. J.: The importance of oxygen for the disparate recovery patterns of the benthic macrofauna in the Early Triassic, *Earth-Sci. Rev.,* 137, 65–84, 2014.

Podlaha, O. G., Mutterlose, J., and Veizer, J.: Preservation of delta 18O and delta 13C in belemnite rostra from the Jurassic/Early Cretaceous successions, *Am. J. Sci.,* 298, 324–347, 1998.

Pogge von Strandmann, P. A. E., Jones, M. T., West, A. J., Murphy, M. J., Stokke, E. W., Tarbuck, G., Wilson, D. J., Pearce, C. R., and Schmidt, D. N.: Lithium isotope evidence for enhanced weathering and erosion during the Paleocene-Eocene Thermal Maximum, *Sci. Adv.,* 7, eabh4224, https://doi.org/10.1126/sciadv.abh4224, 2021.

Polteau, S., Hendriks, B. W., Planke, S., Ganerød, M., Corfu, F., Faleide, J. I., Midtkandal, I., Svensen, H. S., and
2675 Myklebust, R.: The Early Cretaceous Barents Sea Sill Complex: distribution, 40Ar/39Ar geochronology, and implications for carbon gas formation, *Palaeogeogr. Palaeoclimatol. Palaeoecol.,* 441, 83–95, 2016.

Polteau, S., Planke, S., Faleide, J. I., Svensen, H., and Myklebust, R.: The Cretaceous high Arctic large igneous province, *EGU Gen. Assem. Conf. Abstr.*, 13216, 2010.

Pott, C.: The Upper Triassic Flora of Svalbard, *Acta Palaeontol. Pol.,* 59, 709–740, 2012.

Price, G. D.: The evidence and implications of polar ice during the Mesozoic, *Earth-Sci. Rev.,* 48, 183–210, 1999.

Price, G. D., Bajnai, D., and Fiebig, J.: Carbonate clumped isotope evidence for latitudinal seawater temperature gradients and the oxygen isotope composition of Early Cretaceous seas, *Palaeogeogr. Palaeoclimatol. Palaeoecol.*, 552, 109777, 2020.



Price, G. D., Főzy, I., and Pálfy, J.: Carbon cycle history through the Jurassic–Cretaceous boundary: A new global δ13C stack, *Palaeogeogr. Palaeoclimatol. Palaeoecol.*, 451, 46–61, https://doi.org/10.1016/j.palaeo.2016.03.016, 2016.

Price, G. D., Hart, M. B., Wilby, P. R., and Page, K. N.: Isotopic analysis of Jurassic (Callovian) mollusks from the Christian Malford Lagerstätte (UK): Implications for ocean water temperature estimates based on belemnoids, *Palaios,* 30, 645–654, 2015.

Price, G. D., and Passey, B. H.: Dynamic polar climates in a greenhouse world: Evidence from clumped isotope thermometry of Early Cretaceous belemnites, *Geology,* 41, 923–926, 2013.

Price, G. D., and Nunn, E. V.: Valanginian isotope variation in glendonites and belemnites from Arctic Svalbard: Transient glacial temperatures during the Cretaceous greenhouse, *Geology,* 38, 251–254, https://doi.org/10.1130/g30593.1, 2010.

Price, G. D., and Rogov, M. A.: An isotopic appraisal of the Late Jurassic greenhouse phase in the Russian Platform, *Palaeogeogr. Palaeoclimatol. Palaeoecol.,* 273, 41–49, https://doi.org/10.1016/j.palaeo.2008.11.011, 2009a.

Price, G. D., and Rogov, M. A.: An isotopic appraisal of the Late Jurassic greenhouse phase in the Russian Platform, *Palaeogeogr. Palaeoclimatol. Palaeoecol.*, 273, 41–49, 2009b.

Price, G. D., and Mutterlose, J.: Isotopic signals from late Jurassic–early Cretaceous (Volgian–Valanginian) sub-Arctic belemnites, Yatria River, Western Siberia, *J. Geol. Soc. Lond.,* 161, 959–968, https://doi.org/10.1144/0016-764903-169, 2004.

Price, G. D.: New constraints upon isotope variation during the early Cretaceous (Barremian–Cenomanian) from the Pacific Ocean, *Geol. Mag.,* 140, 513–522, 2003.

Pucéat, E., Lécuyer, C., Sheppard, S. M., Dromart, G., Reboulet, S., and Grandjean, P.: Thermal evolution of Cretaceous Tethyan marine waters inferred from oxygen isotope composition of fish tooth enamels, *Paleoceanography*, 18, 2003.

Pörtner, H.-O., Roberts, D. C., Adams, H., Adler, C., Aldunce, P., Ali, E., Begum, R. A., Betts, R., Kerr, R. B., Biesbroek, R., Birkmann, J., Bowen, K., Castellanos, E. C., Constable, A., Cramer, W., Dodman, D., Eriksen, S. H., Fischlin, A., Garschagen, M., Glavovic, B., Gilmore, E., Haasnoot, M., Harper, S., Hasegawa, T., Hayward, B., Hirabayashi, Y., Howden, M., Kalaba, K., Kiessling, W., Lasco, R., Lawrence, J., Lemos, M. F., Lempert, R., Ley, D., Lissner, T., Lluch-Cota, S., Loeschke, S., Lucatello, S., Luo, Y., Mackey, B., Maharaj, S., Mendez, C., Mintenbeck, K., Moncassim Vale, M., Morecroft, M. D., Mukherji, A., Mycoo, M., Mustonen, T., Nalau, J., Okem, A., Ometto, J. P., Parmesan, C., Pelling, M., Pinho, P., Poloczanska, E., Racault, M.-F., Reckien, D., Pereira, J., Revi, A., Rose, S., Sanchez-Rodriguez, R., Schipper, E. L. F., Schmidt, D., Schoeman, D., Shaw, R., Singh, C., Solecki, W., Stringer, L., Thomas, A., Totin, E., Trisos, C., Viner, D., van Aalst, M., Wairiu, M., Warren, R., Yanda, P., and Zaiton Ibrahim, Z.:



*Climate change 2022: Impacts, adaptation and vulnerability, IPCC Sixth Assessment Report*, https://edepot.wur.nl/565644, 2022.

## Q

Quattrini, A. M., Rodríguez, E., Faircloth, B. C., Cowman, P. F., Brugler, M. R., Farfan, G. A., Hellberg, M. E., Kitahara, M. V., Morrison, C. L., Paz-García, D. A., Reimer, J. D., and McFadden, C. S.: Palaeoclimate ocean conditions shaped the evolution of corals and their skeletons through deep time, *Nat. Ecol. Evol.,* 4,
1531–1538, https://doi.org/10.1038/s41559-020-01291-1, 2020.

## R

Rantanen, M., Karpechko, A. Y., Lipponen, A., Nordling, K., Hyvärinen, O., Ruosteenoja, K., Vihma, T., and Laaksonen, A.: The Arctic has warmed nearly four times faster than the globe since 1979, *Commun. Earth*
*Environ.*, 3, 1–10, https://doi.org/10.1038/s43247-022-00498-3, 2022.

Rauzi, S., Foster, W. J., Takahashi, S., Hori, R. S., Beaty, B. J., Tarhan, L. G., and Isson, T.: Lithium isotopic evidence for enhanced reverse weathering during the Early Triassic warm period*, Proc. Natl. Acad. Sci. USA,* 121, e2318860121, 2024.

Reichow, M. K., Pringle, M. S., Al'Mukhamedov, A. I., Allen, M., Andreichev, V. L., Buslov, M. M., Davies, C.,
Fedoseev, G. S., Fitton, J. G., and Inger, S.: The timing and extent of the eruption of the Siberian Traps large igneous province: Implications for the end-Permian environmental crisis, *Earth Planet. Sci. Lett.*, 277, 9–20, 2009.

Retallack, G. J., and Conde, G. D.: Deep time perspective on rising atmospheric $CO_2$, *Global Planet. Change*, 189, 103177, https://doi.org/10.1016/j.gloplacha.2020.103177, 2020.

Rigo, M., Preto, N., Roghi, G., Tateo, F., and Mietto, P.: A rise in the carbonate compensation depth of western Tethys in the Carnian (Late Triassic): deep-water evidence for the Carnian Pluvial Event, *Palaeogeogr. Palaeoclimatol. Palaeoecol.*, 246, 188–205, 2007.

Rigo, M., and Joachimski, M. M.: Palaeoecology of Late Triassic conodonts: Constraints from oxygen isotopes in biogenic apatite, *Acta Palaeontol. Pol.*, 55, 471–478, 2010.

Rigo, M., Jin, X., Godfrey, L., Katz, M. E., Sato, H., Tomimatsu, Y., and Onoue, T.: Unveiling a new oceanic anoxic event at the Norian/Rhaetian boundary (Late Triassic), *Sci. Rep.,* 14, 15574, 2024.

Riis, F., Lundschien, B. A., Høy, T., Mørk, A., and Mørk, M. B. E.: Evolution of the Triassic shelf in the northern Barents Sea region, *Polar Res.,* 27, 318–338, https://doi.org/10.1111/j.1751-8369.2008.00086.x, 2008.



Riis, F., and Fjeldskaar, W.: On the magnitude of the Late Tertiary and Quaternary erosion and its significance for the uplift of Scandinavia and the Barents Sea, in: *Structural and tectonic modelling and its application to petroleum geology, edited by: Larsen, R. M., Brekke, H., Larsen, B. T., and Talleraas, E., Elsevier, Amsterdam*, 163–185, 1992.

Rismyhr, B., Bjærke, T., Olaussen, S., Mulrooney, M. J., and Senger, K.: Facies, palynostratigraphy and sequence stratigraphy of the Wilhelmøya Subgroup (Upper Triassic–Middle Jurassic) in western central Spitsbergen, Svalbard, *Nor. Geol. Tidsskr.*, 99, 35–64, 2018.

Rizzo, R. E., Inskip, N. F., Fazeli, H., Betlem, P., Bisdom, K., Kampman, N., and Busch, A.: Modelling geological CO2 leakage: Integrating fracture permeability and fault zone outcrop analysis, *Int. J. Greenh. Gas* Control, 133, 104105, 2024.

Roberts, A. J., Engelschion, V. S., and Hurum, J. H.: First three-dimensional skull of the Middle Triassic mixosaurid ichthyosaur Phalarodon fraasi from Svalbard, Norway, *Acta Palaeontol. Pol.*, 67, 51–62, 2022.

Rocha, B. C., Davies, J. H., Janasi, V. A., Schaltegger, U., Nardy, A. J., Greber, N. D., and Lucchetti, A. C. F.: Rapid eruption of silicic magmas from the Parana magmatic province (Brazil) did not trigger the Valanginian event, *Geology*, 48, 1174–1178, 2020.

Rodriguez Blanco, L., Swart, P. K., Eberli, G. P., & Weger, R. J.: Negative $\delta^{13}C$ carb values at the Jurassic-Cretaceous boundary – Vaca Muerta Formation, Neuquén Basin, Argentina. *Palaeogeography, Palaeoclimatology, Palaeoecology*, **603**, 111208. https://doi.org/10.1016/j.palaeo.2022.111208, 2022.

Rodríguez-Tovar, F. J., Dorador, J., Zuchuat, V., Planke, S., and Hammer, Ø.: Response of macrobenthic trace maker community to the end-Permian mass extinction in Central Spitsbergen, Svalbard, *Palaeogeogr. Palaeoclimatol. Palaeoecol.*, 581, 2021.

Roest, W. R., and Srivastava, S. P.: Sea-floor spreading in the Labrador Sea: A new reconstruction, *Geology,* 17, 1000–1003, https://doi.org/10.1130/0091-7613(1989)017<1000:Sfsitl>2.3.Co;2, 1989.

Roghi, G.: Palynological investigations in the Carnian of the Cave del Predil area (Julian Alps, NE Italy), *Rev. Palaeobot. Palynol.*, 132, 1–35, 2004.

Rogov, M., and Zakharov, V.: Jurassic and Lower Cretaceous glendonite occurrences and their implication for Arctic paleoclimate reconstructions and stratigraphy, *Earth Sci. Front.,* 17, 345–347, 2010.

Rogov, M. A., Ershova, V. B., Shchepetova, E. V., Zakharov, V. A., Pokrovsky, B. G., and Khudoley, A. K.: Earliest Cretaceous (late Berriasian) glendonites from Northeast Siberia revise the timing of initiation of transient Early Cretaceous cooling in the high latitudes, *Cretac. Res.,* 71, 102–112, https://doi.org/10.1016/j.cretres.2016.11.011, 2017.



Rogov, M. A.: Ammonites and infrazonal stratigraphy of the Kimmeridgian and Volgian stages of Panboreal Superrealm. *Transactions of the Geological Institute*, 627, 1–732, https://doi.org/10.54896/00023272_2021_627_1, 2021.

Rogov, M. A., Zakharov, V., & Kiselev, D.: Refined ammonite and bivalve biostratigraphy of the Agardhfjellet and lowermost Rurikfjellet formations (Bathonian–Ryazanian) of the Longyearbyen area, Spitsbergen. *Neues Jahrbuch für Geologie und Paläontologie, Abhandlungen*, **309**(2), 169–198. https://doi.org/10.1127/njgpa/2023/1158, 2023.

Rooney, A. D., Strauss, J. V., Brandon, A. D., and Macdonald, F. A.: A Cryogenian chronology: Two long-lasting synchronous Neoproterozoic glaciations, *Geology,* 43, 459–462, 2015.

Royer, D. L., Berner, R. A., Montañez, I. P., Tabor, N. J., and Beerling, D. J.: CO2 as a primary driver of Phanerozoic climate, *GSA Today*, 14, 4–10, 2004.

Royer, D. L.: Linkages between CO2, climate, and evolution in deep time, *Proc. Natl. Acad. Sci. USA*, 105, 407–408, https://doi.org/10.1073/pnas.0710915105, 2008.

## S

Salamon, M.A., Hanken, N., Gorzelak, P., Riise, H.E. and Ferré, B., 2015. Crinoids from Svalbard in the aftermath of the end− Permian mass extinction. *Polish Polar Research*: 225-238-225-238,

Salamon, M. A., Hanken, N., Gorzelak, P., Riise, H. E., and Ferré, B.: Crinoids from Svalbard in the aftermath of the end-Permian mass extinction, *Pol. Polar Res.*, 36, 225–238, 2015.

Sangiorgi, F., Brumsack, H. J., Willard, D. A., Schouten, S., Stickley, C. E., O'Regan, M., Reichart, G. J., Sinninghe Damsté, J. S., and Brinkhuis, H.: A 26 million year gap in the central Arctic record at the greenhouse-icehouse transition: Looking for clues, *Paleoceanography*, 23, 2008.

Sanei, H., Grasby, S. E., & Beauchamp, B.: Latest Permian mercury anomalies. *Geology, 40*(1), 63-66, (2012.

Scheibner, C., Hartkopf-Fröder, C., Blomeier, D., and Forke, H.: The Mississippian (Lower Carboniferous) in northeast Spitsbergen (Svalbard) and a re-evaluation of the Billefjorden Group, *Z. Dtsch. Ges. Geowiss.*, 163, 293, 2012.

Schlanger, S. O., and Jenkyns, H.: Cretaceous oceanic anoxic events: causes and consequences, *Geol. Mijnb., 55, 1976.*

Schlegel, A., Lisker, F., Dörr, N., Jochmann, M., Schubert, K., and Spiegel, C.: Petrography and geochemistry of siliciclastic rocks from the Central Tertiary Basin of Svalbard–implications for provenance, tectonic setting and climate, *Z. Dtsch. Ges. Geowiss.,* 164, 173–186, 2013.



Schobben, M., Joachimski, M. M., Korn, D., Leda, L., and Korte, C.: Palaeotethys seawater temperature rise and an intensified hydrological cycle following the end-Permian mass extinction, *Gondwana Res.*, 26, 675–683, 2014.

Schobben, M., Foster, W. J., Sleveland, A. R., Zuchuat, V., Svensen, H. H., Planke, S., Bond, D. P., Marcelis, F., Newton, R. J., and Wignall, P. B.: A nutrient control on marine anoxia during the end-Permian mass extinction, *Nat. Geosci.,* 13, 640–646, 2020.

Schröder-Adams, C. J., Herrle, J. O., Embry, A. F., Haggart, J. W., Galloway, J. M., Pugh, A. T., and Harwood, D. M.: Aptian to Santonian foraminiferal biostratigraphy and paleoenvironmental change in the Sverdrup Basin as revealed at Glacier Fiord, Axel Heiberg Island, Canadian Arctic Archipelago, *Palaeogeogr. Palaeoclimatol. Palaeoecol.*, 413, 81–100, 2014.

Schweitzer, J., Paulsen, B., Antonovskaya, G. N., Fedorov, A. V., Konechnaya, Y. V., Asming, V. E., and Pirli, M.: A 24-Yr-Long Seismic Bulletin for the European Arctic, *Seismol. Res. Lett.,* 92, 2758–2767, https://doi.org/10.1785/0220210018, 2021.

Schaaf, N. W., Osmundsen, P. T., Van der Lelij, R., Schönenberger, J. L., OK, R. T., and Senger, K.: Tectono-sedimentary evolution of the eastern Forlandsundet Graben, Svalbard, *Nor. J. Geol.,* 100, https://doi.org/10.1785/njg100-4-4, 2021.

Schobben, Martin, William J. Foster, Arve RN Sleveland, Valentin Zuchuat, Henrik H. Svensen, Sverre Planke, David PG Bond et al. "A nutrient control on marine anoxia during the end-Permian mass extinction." *Nature Geoscience* 13, no. 9: 640-646. 2020.

Scotese, C. R., Bambach, R. K., Barton, C., Van der Voo, R., and Ziegler, A. M.: Paleozoic base maps, *J. Geol.,* 87, 217–277, https://doi.org/10.1086/628416, 1979.

Scotese, C. R., and Wright, N.: PALEOMAP Paleodigital Elevation Models (PaleoDEMs) for the Phanerozoic, PALEOMAP Project, https://www.earthbyte.org/paleodem-resource-scotese-and-wright-2018, 2018.

Scotese, C. R., Boucot, A. J., and McKerrow, W. S.: Gondwanan palaeogeography and palaeoclimatology, J. Afr. *Earth Sci.,* 28, 99–114, 1999.

Scotese, C. R., Song, H., Mills, B. J. W., and van der Meer, D. G.: Phanerozoic paleotemperatures: The earth's changing climate during the last 540 million years, *Earth-Sci. Rev.,* 215, https://doi.org/10.1016/j.earscirev.2021.103503, 2021.

Screen, J. A., and Simmonds, I.: The central role of diminishing sea ice in recent Arctic temperature amplification, *Nature,* 464, 1334–1337, https://doi.org/10.1038/nature09051, 2010.

Scrutton, C., Horsfield, W., and Harland, W.: Silurian fossils from western Spitsbergen, *Geol. Mag.,* 113, 519–523, 1976.

Sellwood, B., and Price, G.: Sedimentary facies as indicators of Mesozoic palaeoclimate, *Philos. Trans. R. Soc. Lond. B Biol. Sci.,* 341, 225–233, 1993.



Senger, K., Brugmans, P., Grundvåg, S.-A., Jochmann, M. M., Nøttvedt, A., Olaussen, S., Skotte, A., and Smyrak-Sikora, A.: Petroleum, coal and research drilling onshore Svalbard: a historical perspective, *Norwegian Journal of Geology,* Vol 99 Nr.3, 1-30, https://dx.doi.org/10.17850/njg99-3-1, 2019.

Senger, K., Planke, S., Polteau, S., Ogata, K., and Svensen, H.: Sill emplacement and contact metamorphism in a siliciclastic reservoir on Svalbard, Arctic Norway, *Nor. J. Geol.,* 94, 2014a.

Senger, K., Tveranger, J., Ogata, K., Braathen, A., and Planke, S.: Late Mesozoic magmatism in Svalbard: A review, *Earth-Sci. Rev.*, 139, 123–144, 2014b.

Senger, K., Tveranger, J., Braathen, A., Olaussen, S., Ogata, K., and Larsen, L.: CO2 storage resource estimates
in unconventional reservoirs: insights from a pilot-sized storage site in Svalbard, Arctic Norway, *Environ. Earth Sci.,* 73, 3987–4009, https://doi.org/10.1007/s12665-014-3684-9, 2015.

Senger, K., Betlem, P., Birchall, T., Gonzaga Jr, L., Grundvåg, S.-A., Horota, R. K., Laake, A., Kuckero, L., Mørk, A., and Planke, S.: Digitising Svalbard's geology: the Festningen digital outcrop model, *First Break*, 40, 47–55, 2022.

Senger, K., Betlem, P., Braathen, A., Olaussen, S., and Sand, G.: Longyearbyen CO2 Lab project - from a vision of a CO2-neutral Svalbard to a geoscience data eldorado, *Arct. Sci.*, https://doi.org/10.1139/as-2024-0019, 2024.

Sepkoski, J. J.: Patterns of Phanerozoic extinction: a perspective from global data bases, in: *Global events and event stratigraphy in the Phanerozoic, edited by: Walliser, O. H., Springer*, Berlin, Heidelberg, 35–
2865 51, https://doi.org/10.1007/978-3-642-79634-0_4, 1996.

Servais, T., and Harper, D. A.: The great Ordovician biodiversification event (GOBE): definition, concept and duration, *Lethaia*, 51, 151–164, 2018.

Sharma, M., Papanastassiou, D. A., and Wasserburg, G. J.: The concentration and isotopic composition of osmium in the oceans, *Geochim. Cosmochim. Acta,* 61, 3287–3299, https://doi.org/10.1016/S0016-
2870 7037(97)00210-X, 1997.

Silva, D., Lana, C., and de Souza Filho, C. R.: Petrographic and geochemical characterization of the granitic rocks of the Araguainha impact crater, Brazil, Meteorit. *Planet. Sci*., 51, 443–467, https://doi.org/10.1111/maps.12601, 2016.

Simms, M. J., and Ruffell, A. H.: Synchroneity of climatic change and extinctions in the Late Triassic, *Geology,* 17,
265–268, 1989.

Śliwińska, K. K., Coxall, H. K., Hutchinson, D. K., Liebrand, D., Schouten, S., and de Boer, A. M.: Sea surface temperature evolution of the North Atlantic Ocean across the Eocene–Oligocene transition, *Clim. Past*, 19, 123–140, 2023.



Śliwińska, K. K., and Head, M. J.: New species of the dinoflagellate cyst genus Svalbardella Manum, 1960, emend.
from the Paleogene and Neogene of the northern high to middle latitudes, *J. Micropalaeontol.,* 39, 139–
154, 2020.

Śliwińska, K. K., and Heilmann-Clausen, C.: Early Oligocene cooling reflected by the dinoflagellate cyst
Svalbardella cooksoniae, *Palaeogeogr. Palaeoclimatol. Palaeoecol.*, 305, 138–149, 2011.

Śliwińska, K. K., Jelby, M. E., Grundvåg, S.-A., Nøhr-Hansen, H., Alsen, P., and Olaussen, S.: Dinocyst stratigraphy
of the Valanginian–Aptian Rurikfjellet and Helvetiafjellet formations on Spitsbergen, Arctic Norway, *Geol.
Mag.*, 157, 1693–1714, https://doi.org/10.1017/S0016756819001249, 2020.

Śliwińska, K. K., Thomsen, E., Schouten, S., Schoon, P. L., and Heilmann-Clausen, C.: Climate-and gateway-
driven cooling of Late Eocene to earliest Oligocene sea surface temperatures in the North Sea Basin, *Sci.
Rep.,* 9, 4458, 2019.

Sluijs, A., Frieling, J., Inglis, G. N., Nierop, K. G., Peterse, F., Sangiorgi, F., and Schouten, S.: Late Paleocene–
early Eocene Arctic Ocean sea surface temperatures: reassessing biomarker paleothermometry at
Lomonosov Ridge, *Clim. Past,* 16, 2381–2400, 2020.

Smelror, M., Olaussen, S., Dumais, M. A., Grundvåg, S. A., and Abay, T. B.: Northern Svalbard Composite Tectono-
Sedimentary Element, *Geol. Soc. Lond. Mem.*, 57, M57-2023, 2024.

Smelror, M., Grenne, T., Gasser, D., and Bøe, R.: Deep-water trace fossils in the Ilfjellet rift basin (Middle
Ordovician), central Norwegian Caledonides, *Palaeoworld,* 32, 63–78, 2023.

Smelror, M., Olaussen, S., Dumais, M. A., Grundvåg, S. A., and Abay, T. B.: Northern Svalbard Composite Tectono-
Sedimentary Element, *Geol. Soc. Lond. Mem.*, 57, M57-2023, 2024.

Smyrak-Sikora, A., Johannessen, E. P., Olaussen, S., Sandal, G., and Braathen, A.: Sedimentary architecture
during Carboniferous rift initiation – the arid Billefjorden Trough, Svalbard, *J. Geol. Soc. Lond.,* 176, 225–
252, https://doi.org/10.1144/jgs2018-100, 2018.

Smyrak-Sikora, A., Nicolaisen, J. B., Braathen, A., Johannessen, E. P., Olaussen, S., and Stemmerik, L.: Impact
of growth faults on mixed siliciclastic-carbonate-evaporite deposits during rift climax and reorganisation—
Billefjorden Trough, Svalbard, Norway, *Basin Res.,* 33, 2643–2674, https://doi.org/10.1111/bre.12578,
2021.

Smyrak-Sikora, A., Engelschiøn, V., Foster, W., Jelby, M. E., Jones, M., Mosociova, T., Śliwińska, K., Vickers, M. L., &
Zuchuat, V.: Table: Review of Phanerozoic Paleoenvironmental and Paleoclimatic Evolution in Svalbard [Data set].
Zenodo. https://doi.org/10.5281/zenodo.14334261, 2024.

Solomon, S., Qin, D., Manning, M., Averyt, K., and Marquis, M.: Climate change 2007-the physical science basis:
Working group I contribution to the fourth assessment report of the IPCC, Cambridge University Press,
2007.



Song, H., Wignall, P. B., Tong, J., Bond, D. P., Song, H., Lai, X., and Chen, Y.: Geochemical evidence from bio-apatite for multiple oceanic anoxic events during Permian–Triassic transition and the link with end-Permian extinction and recovery, *Earth Planet. Sci. Lett.,* 353, 12–21, 2012.

Song, H., Wignall, P. B., Song, H., Dai, X., and Chu, D.: Seawater temperature and dissolved oxygen over the past 500 million years*, J. Earth Sci.*, 30, 236–243, 2019.

Soreghan, G. S.: Déjà-Vu All Over Again: Deep Time (Climate) Is Here To Stay, PALAIOS, 19, 1–2, https://doi.org/10.1669/0883-1351(2004)019<0001:Daoadt>2.0.Co;2, 2004.

Sorento, T., Olaussen, S., and Stemmerik, L.: Controls on deposition of shallow marine carbonates and evaporites

– lower Permian Gipshuken Formation, central Spitsbergen, Arctic Norway, *Sedimentology,* 67, 207–238, https://doi.org/10.1111/sed.12640, 2020.

Soua, M.: Early Carnian anoxic event as recorded in the southern Tethyan margin, Tunisia: an overview, Int. *Geol. Rev.*, 56, 1884–1905, 2014.

Speelman, E. N., van Kempen, M. M. L., Barke, J., Brinkhuis, H., Reichart, G. J., Smolders, A. J. P., Roelofs, J. G.

M., Sangiorgi, F., de Leeuw, J. W., Lotter, A. F., and Sinninghe Damsté, J. S.: The Eocene Arctic Azolla bloom: environmental conditions, productivity and carbon drawdown, *Geobiology*, 7, 155–170, https://doi.org/10.1111/j.1472-4669.2009.00195.x, 2009.

Spielhagen, R. F., and Tripati, A.: Evidence from Svalbard for near-freezing temperatures and climate oscillations in the Arctic during the Paleocene and Eocene, *Palaeogeogr. Palaeoclimatol. Palaeoecol.*, 278, 48–

56, https://doi.org/10.1016/j.palaeo.2009.04.012, 2009.

Stanley Jr, G. D.: The evolution of modern corals and their early history, Earth-Sci. Rev., 60, 195–225, 2003.

Stanley, S. M.: Estimates of the magnitudes of major marine mass extinctions in earth history, *Proc. Natl. Acad. Sci. USA*, 113, E6325–E6334, https://doi.org/10.1073/pnas.161309411, 2016.

Steel, R. J., Gjelberg, J., and Haarr, G.: Helvetiafjellet Formation (Barremian) at Festningen, Spitsbergen– a field

guide, *Nor. Polarinst. Årbok*, 111–128, 1978.

Steel, R. J., Dalland, A., Kalgraff, K., and Larsen, V.: The Central Tertiary Basin of Spitsbergen: sedimentary development of a sheared-margin basin, Geology of the North Atlantic Borderlands — Memoir 7, *CSPG Special Publications,* 647-664, 1981.

Steel, R., Gjelberg, J., Helland-Hansen, W., Kleinspehn, K., Nøttvedt, A., and Rye-Larsen, M.: The Tertiary strike-

slip basins and orogenic belt of Spitsbergen, *Special Publications of SEPM,* 1985.

Steel, R. J., and Worsley, D.: Svalbard's post-Caledonian strata—an atlas of sedimentational patterns and palaeogeographic evolution, in: Petroleum geology of the North European margin, edited by: Spencer, A. M., Springer, Dordrecht, 109–135, https://doi.org/10.1007/978-94-009-5626-1_9, 1984.

Stemmerik, L.: Late Palaeozoic evolution of the North Atlantic margin of Pangea*, Palaeogeogr. Palaeoclimatol.*

*Palaeoecol.*, 161, 95–126, 2000.



Stemmerik, L.: Influence of late Paleozoic Gondwana glaciations on the depositional evolution of the northern Pangean shelf, North Greenland, Svalbard, and the Barents Sea, *Geol. Soc. Am. Spec. Pap.*, 441, 2008.

Stemmerik, L., Bendix-Almgreen, S. E., and Piasecki, S.: The Permian–Triassic boundary in central East Greenland: past and present views, *Bull. Geol. Soc. Den.*, 48, 159–167, 2001.

Stemmerik, L., and Worsley, D.: 30 years on-Arctic Upper Palaeozoic stratigraphy, depositional evolution and hydrocarbon prospectivity, *Nor. J. Geol.,* 85, 2005.

Stewart, I.: Sustainable geoscience, *Nat. Geosci.,* 9, 262, 2016.

Stickley, C. E., Brinkhuis, H., Schellenberg, S. A., Sluijs, A., Röhl, U., Fuller, M., Grauert, M., Huber, M., Warnaar, J., and Williams, G. L.: Timing and nature of the deepening of the Tasmanian Gateway, *Paleoceanography,*
19, https://doi.org/10.1029/2004PA001022, 2004.

Stickley, C. E., St John, K., Koç, N., Jordan, R. W., Passchier, S., Pearce, R. B., and Kearns, L. E.: Evidence for middle Eocene Arctic sea ice from diatoms and ice-rafted debris, *Nature,* 460, 376–379, http://www.nature.com/doifinder/10.1038/nature08163, 2009.

Stordal, F., Svensen, H. H., Aarnes, I., and Roscher, M.: Global temperature response to century-scale degassing
from the Siberian Traps Large igneous province, *Palaeogeogr. Palaeoclimatol. Palaeoecol.,* 471, 96–107, https://doi.org/10.1016/j.palaeo.2017.01.045, 2017.

Storey, M., Duncan, R. A., and Tegner, C.: Timing and duration of volcanism in the North Atlantic Igneous Province: Implications for the geodynamics and links to the Iceland hotspot, *Chem. Geol.,* 241, 264–281, 2007a.

Storey, M., Duncan, R. A., and Swisher III, C. C.: Paleocene-Eocene Thermal Maximum and the opening of the
Northeast Atlantic, *Science*, 316, 587–589, 2007b.

Stouge, S., Christiansen, J. L., and Holmer, L. E.: Lower palaeozoic stratigraphy of Murchisonfjorden and Sparreneset, Nordaustlandet, Svalbard*, Geogr. Ann. Ser. A-Phys. Geogr.*, 93, 209–226, 2011.

Stouge, S., Harper, D. A., Boyce, W. D., and Christiansen, I. K. L.: Development of the lower Cambrian–Middle Ordovician carbonate platform: North Atlantic Region, 2012.

Straume, E. O., Gaina, C., Medvedev, S., and Nisancioglu, K. H.: Global Cenozoic paleobathymetry with a focus on the Northern Hemisphere oceanic gateways, *Gondwana Res.,* 86, 126–143, https://doi.org/10.1016/j.gr.2020.05.011, 2020.

Straume, E. O., Nummelin, A., Gaina, C., and Nisancioglu, K. H.: Climate transition at the Eocene–Oligocene influenced by bathymetric changes to the Atlantic–Arctic oceanic gateways, *Proc. Natl. Acad. Sci. USA*,
119, e2115346119, https://doi.org/10.1073/pnas.2115346119, 2022.

Suan, G., Popescu, S.-M., Suc, J.-P., Schnyder, J., Fauquette, S., Baudin, F., Yoon, D., Piepjohn, K., Sobolev, N. N., and Labrousse, L.: Subtropical climate conditions and mangrove growth in Arctic Siberia during the early Eocene, *Geology*, 45, 539–542, https://doi.org/10.1130/G38547.1, 2017.



Suc, J.-P., Fauquette, S., Popescu, S.-M., and Robin, C.: Subtropical mangrove and evergreen forest reveal Paleogene terrestrial climate and physiography at the North Pole, *Palaeogeogr. Palaeoclimatol. Palaeoecol.*, 551, 109755, 2020.

Sudermann, M., Galloway, J. M., Greenwood, D. R., West, C. K., and Reinhardt, L.: Palynostratigraphy of the lower Paleogene Margaret Formation at Stenkul Fiord, Ellesmere Island, Nunavut, Canada, *Palynology,* 45, 459–476, 2021.

Summerhayes, C. P.: Earth's climate evolution, John Wiley and Sons, 2015.

Summons, R. E., Welander, P. V., and Gold, D. A.: Lipid biomarkers: molecular tools for illuminating the history of microbial life, *Nat. Rev. Microbiol.*, 20, 174–185, 2022.

Sun, Y. D., Wignall, P. B., Joachimski, M. M., Bond, D. P., Grasby, S. E., Lai, X. L., and Sun, S.: Climate warming, euxinia and carbon isotope perturbations during the Carnian (Triassic) Crisis in South China, *Earth Planet. Sci. Lett.*, 444, 88–100, 2016.

Svensen, H., Planke, S., Malthe-Sørenssen, A., Jamtveit, B., Myklebust, R., Rasmussen Eidem, T., and Rey, S. S.: Release of methane from a volcanic basin as a mechanism for initial Eocene global warming, *Nature,* 429, 542–545, 2004.

Svensen, H., Planke, S., Polozov, A. G., Schmidbauer, N., Corfu, F., Podladchikov, Y. Y., and Jamtveit, B.: Siberian gas venting and the end-Permian environmental crisis, *Earth Planet. Sci. Lett.,* 277, 490–500, https://doi.org/10.1016/j.epsl.2008.11.015, 2009.

Svensen, H. H., Frolov, S., Akhmanov, G. G., Polozov, A. G., Jerram, D. A., Shiganova, O. V., Melnikov, N. V., Iyer, K., and Planke, S.: Sills and gas generation in the Siberian Traps, *Philos. Trans. R. Soc. A-Math. Phys. Eng. Sci.,* 376, 20170080, https://doi.org/10.1098/rsta.2017.0080, 2018.

Svensen, H. H., Jerram, D. A., Polozov, A. G., Planke, S., Neal, C. R., Augland, L. E., and Emeleus, H. C.: Thinking about LIPs: A brief history of ideas in Large igneous province research, *Tectonophysics*, 760, 229–251, 2019.

Swainson, I. P., and Hammond, R. P.: Ikaite, CaCO3·6H2O: Cold comfort for glendonites as paleothermometers, *Am. Mineral.*, 86, 1530–1533, https://doi.org/10.2138/am-2001-11-1223, 2001.

## T

Tarduno, J., Sliter, W., Kroenke, L., Leckie, M., Mayer, H., Mahoney, J., Musgrave, R., Storey, M., and Winterer, E.: Rapid formation of Ontong Java Plateau by Aptian mantle plume volcanism, *Science,* 254, 399–403, 1991.



3010 Tejada, M., Mahoney, J., Neal, C., Duncan, R., and Petterson, M.: Basement geochemistry and geochronology of Central Malaita, Solomon Islands, with implications for the origin and evolution of the Ontong Java Plateau, *J. Petrol.,* 43, 449–484, 2002.

Tejada, M. L. G., Suzuki, K., Kuroda, J., Coccioni, R., Mahoney, J. J., Ohkouchi, N., Sakamoto, T., and Tatsumi, Y.: Ontong Java Plateau eruption as a trigger for the early Aptian oceanic anoxic event, *Geology*, 37, 855–
3015 858, 2009.

Thordarson, T.: Accretionary-lapilli-bearing pyroclastic rocks at ODP Leg 192 Site 1184: a record of subaerial phreatomagmatic eruptions on the Ontong Java Plateau, *Geol. Soc. Lond. Spec. Publ.,* 229, 275–306, 2004.

Throndsen, T.: Vitrinite reflectance studies of coals and dispersed organic matter in Tertiary deposits in the
3020 Adventdalen area, Svalbard, *Polar Res.,* 2, 77–91, 1982.

Toggweiler, J., and Bjornsson, H.: Drake Passage and palaeoclimate, *J. Quat. Sci.,* 15, 319–328, 2000.

Tohver, E., Lana, C., Cawood, P. A., Fletcher, I. R., Jourdan, F., Sherlock, S., Rasmussen, B., Trindade, R. I. F., Yokoyama, E., Souza Filho, C. R., and Marangoni, Y.: Geochronological constraints on the age of a Permo–Triassic impact event: U–Pb and 40Ar/39Ar results for the 40 km Araguainha structure of central
3025 Brazil, *Geochim. Cosmochim. Acta,* 86, 214–227, https://doi.org/10.1016/j.gca.2012.03.005, 2012.

Tomimatsu, Y., Onoue, T., and Rigo, M.: Conodont and radiolarian biostratigraphic age constraints on Carnian (Upper Triassic) chert-hosted stratiform manganese deposits from Panthalassa: Formation of deep-sea mineral resources during the Carnian pluvial episode, *Mar. Micropaleontol*., 171, 102084, 2022.

Torsvik, T. H., Carlos, D., Mosar, J., Cocks, L. R. M., and Malme, T.: Global reconstructions and North Atlantic
3030 paleogeography 440 Ma to recent, in: BATLAS–Mid Norway plate reconstruction atlas with global and Atlantic perspectives, edited by: Eide, E. A., *Geol. Surv. Norw*., Trondheim, 18–39, 2002a.

Torsvik, T. H., Carlos, D., Mosar, J., Cocks, L. R. M., and Malme, T.: Global reconstructions and North Atlantic paleogeography 440 Ma to recent, in: BATLAS–Mid Norway plate reconstruction atlas with global and Atlantic perspectives, edited by: Eide, E. A., *Geol. Surv. Norw*., Trondheim, 18, 39, 2002b.

3035 Torsvik, T. H., and Cocks, L. R. M.: The integration of palaeomagnetism, the geological record and mantle tomography in the location of ancient continents, *Geol. Mag.,* 156, 242–260, 2019.

Torsvik, T. H., Van der Voo, R., Preeden, U., Mac Niocaill, C., Steinberger, B., Doubrovine, P. V., Van Hinsbergen, D. J., Domeier, M., Gaina, C., and Tohver, E.: Phanerozoic polar wander, palaeogeography and dynamics, *Earth-Sci. Rev*., 114, 325–368, 2012.

3040 Tozer, E. T., and Parker, J. R.: Notes on the Triassic biostratigraphy of Svalbard, *Geol. Mag.,* 105, 526–542, 1968.

Tozer, E. T.: A standard for Triassic Time, *Geol. Surv. Can. Bull.,* 156, 1–103, 1967.





Tremolada, F., Bornemann, A., Bralower, T. J., Koeberl, C., and van de Schootbrugge, B.: Paleoceanographic changes across the Jurassic/Cretaceous boundary: The calcareous phytoplankton response, *Earth Planet. Sci. Lett.*, 241, 361–371, https://doi.org/10.1016/j.epsl.2005.11.047, 2006.

Tripati, A., and Darby, D.: Evidence for ephemeral middle Eocene to early Oligocene Greenland glacial ice and pan-Arctic sea ice, *Nat. Commun.,* 9, 1038, 2018.

Twitchett, R. J., Looy, C. V., Morante, R., Visscher, H., and Wignall, P. B.: Rapid and synchronous collapse of marine and terrestrial ecosystems during the end-Permian biotic crisis, *Geology*, 29, 351–354, 2001.

## U

Uchman, A., Hanken, N.-M., Nielsen, J. K., Grundvåg, S.-A., and Piasecki, S.: Depositional environment, ichnological features and oxygenation of Permian to earliest Triassic marine sediments in central Spitsbergen, Svalbard, *Polar Res.,* 35, 24782, 2016.

Uhl, D., Traiser, C., Griesser, U., and Denk, T.: Fossil leaves as palaeoclimate proxies in the Palaeogene of Spitsbergen (Svalbard), *Acta Palaeobot.,* 47, 89, 2007

## V

Van Der Meer, D. G., Zeebe, R. E., van Hinsbergen, D. J., Sluijs, A., Spakman, W., and Torsvik, T. H.: Plate tectonic controls on atmospheric CO2 levels since the Triassic, *Proc. Natl. Acad. Sci. USA*, 111, 4380–

4385, 2014.

Van der Meer, D. G., van Saparoea, A. V. D. B., Van Hinsbergen, D. J. J., Van de Weg, R. M. B., Godderis, Y., Le Hir, G., and Donnadieu, Y.: Reconstructing first-order changes in sea level during the Phanerozoic and Neoproterozoic using strontium isotopes, *Gondwana Res.*, 44, 22–34, 2017.

van de Schootbrugge, B., Föllmi, K. B., Bulot, L. G., and Burns, S. J.: Paleoceanographic changes during the early

Cretaceous (Valanginian–Hauterivian): evidence from oxygen and carbon stable isotopes, *Earth Planet. Sci. Lett.*, 181, 15–31, 2000.

Veizer, J., Ala, D., Azmy, K., Bruckschen, P., Buhl, D., Bruhn, F., Carden, G. A. F., Diener, A., Ebneth, S., Godderis, Y., Jasper, T., Korte, C., Pawellek, F., Podlaha, O. G., and Strauss, H.: 87Sr/86Sr, δ13C and δ18O evolution of Phanerozoic seawater, *Chem. Geol.*, 161, 59–88, https://doi.org/10.1016/s0009-

2541(99)00081-9, 1999.

Veizer, J., and Prokoph, A.: Temperatures and oxygen isotopic composition of Phanerozoic oceans, *Earth-Sci. Rev.*, 146, 92–104, 2015.



Vérard, C., and Veizer, J.: On plate tectonics and ocean temperatures, *Geology*, 47, 881–885, 2019.

Verba, M.: Kollektornye svoystva porod osadochnogo chekhla arkhipelaga Shpitsbergen [Sedimentary cover reservoir of Svalbard archipelago], *Neftegaz. Geol. Teor. Prakt.*, 8, 1–45, 2013.

Vickers, M. L., Price, G. D., Jerrett, R. M., and Watkinson, M.: Stratigraphic and geochemical expression of Barremian–Aptian global climate change in Arctic Svalbard, *Geosphere*, 12, 1594–1605, https://doi.org/10.1130/ges01344.1, 2016.

Vickers, M. L., Price, G. D., Jerrett, R. M., Sutton, P., Watkinson, M. P., and FitzPatrick, M.: The duration and magnitude of Cretaceous cool events: Evidence from the northern high latitudes, *GSA Bull.,* 131, 1979–1994, https://doi.org/10.1130/b35074.1, 2019a.

Vickers, M. L., Bajnai, D., Price, G. D., Linckens, J., and Fiebig, J.: Southern high-latitude warmth during the Jurassic–Cretaceous: New evidence from clumped isotope thermometry, *Geology,* 47, 724–728, 2019b.

Vickers, M. L., Fernandez, A., Hesselbo, S. P., Price, G. D., Bernasconi, S. M., Lode, S., Ullmann, C. V., Thibault, N., Hougaard, I. W., and Korte, C.: Unravelling Middle to Late Jurassic palaeoceanographic and palaeoclimatic signals in the Hebrides Basin using belemnite clumped isotope thermometry, *Earth Planet. Sci. Lett.*, 546, 116401, 2020.

Vickers, M. L., Bernasconi, S. M., Ullmann, C. V., Lode, S., Looser, N., Morales, L. G., Price, G. D., Wilby, P. R., Hougård, I. W., Hesselbo, S. P., and Korte, C.: Marine temperatures underestimated for past greenhouse climate, *Sci. Rep.,* 11, 19109, 2021.

Vickers, M. L., Jelby, M. E., Śliwińska, K. K., Percival, L. M., Wang, F., Sanei, H., Price, G. D., Ullmann, C. V., Grasby, S. E., and Reinhardt, L.: Volcanism and carbon cycle perturbations in the High Arctic during the Late Jurassic–Early Cretaceous, *Palaeogeogr. Palaeoclimatol. Palaeoecol.,* 613, 111412, 2023.

Vigran, J. O.: Spores from Devonian deposits, Mimerdalen, Spitsbergen, *Norsk Polarinstitutt Skrifter*, v.132, p.1–33, 1964.

Vigran, J. O., Mangerud, G., Mørk, A., Worsley, D., and Hochuli, P. A.: Palynology and geology of the Triassic succession of Svalbard and the Barents Sea, *Nor. Geol. Unders.,* 14, 2014.

Vogt, T.: Geology of a Middle Devonian cannel coal from Spitsbergen, *Nor. Geol. Tidsskr.,* 21, 1–12, 1941.

von Appen, W.-J., Schauer, U., Somavilla, R., Bauerfeind, E., and Beszczynska-Möller, A.: Exchange of warming deep waters across Fram Strait, *Deep-Sea Res.* Pt. I, 103, 86–100, https://doi.org/10.1016/j.dsr.2015.06.003, 2015.



# W

Wagner, M., Hendy, I. L., & Lai, B.: Characterizing Ag uptake and storage in the marine diatom *Thalassiosira*
*pseudonana*: Implications for Ag biogeochemical cycling. *Marine Chemistry*, **247**,
    104175. https://doi.org/10.1016/j.marchem.2022.1041, 2022.

Wappler, T. and Denk, T.: Herbivory in early Tertiary Arctic forests. *Palaeogeography, Palaeoclimatology,*
    *Palaeoecology*, 310(3-4): 283-295, 2011.

Ware, D., Jenks, J. F., Hautmann, M., and Bucher, H.: Dienerian (Early Triassic) ammonoids from the Candelaria
Hills (Nevada, USA) and their significance for palaeobiogeography and palaeoceanography. *Swiss Journal*
    *of Geosciences,* 104, 161-181, 2011.

Watson, R., Baste, I., Larigauderie, A., Leadley, P., Pascual, U., Baptiste, B., ... and Mooney, H.: Summary for
    policymakers of the global assessment report on biodiversity and ecosystem services of the
    Intergovernmental Science-Policy Platform on Biodiversity and Ecosystem Services. *IPBES Secretariat:*
*Bonn, Germany*, 22-47, 2019.

Webby, B.D., Paris, F., Droser, M.L. and Percival, I.G.: The great Ordovician biodiversification event. Columbia
    University Press, 2004.

Weber, M.: Paleoenvironments during the Paleogene in the High Arctic of Spitsbergen – Evidence from
    Sedimentology and Palynology, Master thesis, Institute of Applied Geoscience, Technical University of
Darmstadt University Centre in Svalbard, 100 pp., 2019.

Weger, R.J., Eberli, G.P., Rodriguez Blanco, L., Tenaglia, M. and Swart, P.K.: Finding a VOICE in the Southern
    Hemisphere: A new record of global organic carbon? *Geological Society of America bulletin*,
    10.1130/B36405.1, 2022.

Weijers, J.W., Schouten, S., Sluijs, A., Brinkhuis, H. and Damsté, J.S.S.: Warm arctic continents during the
Palaeocene–Eocene thermal maximum. *Earth and Planetary Science Letters,* 261(1-2): 230-238,
    https://doi.org/10.1016/j.epsl.2007.06.033, 2007.

Weissert, H. and Channell, J.E.T.: Tethyan carbonate carbon isotope stratigraphy across the Jurassic-Cretaceous
    boundary: An indicator of decelerated global carbon cycling? *Paleoceanography,* 4(4): 483-494,
    https://doi.org/10.1029/PA004i004p00483, 1989.

Weissert, H. and Erba, E.: Volcanism, CO2 and palaeoclimate: a Late Jurassic–Early Cretaceous carbon and
    oxygen isotope record. *Journal of the Geological Society*, 161(4): 695-702, 10.1144/0016-764903-087,
    2004.

Weissert, H., Lini, A., Föllmi, K.B. and Kuhn, O.: Correlation of Early Cretaceous carbon isotope stratigraphy and
    platform drowning events: a possible link? *Palaeogeography, Palaeoclimatology, Palaeoecology,* 137(3):
189-203, https://doi.org/10.1016/S0031-0182(97)00109-0, 1998.



Wesenlund, F., Grundvåg, S.-A., Engelschiøn, V.S., Thießen, O. and Pedersen, J.H.: Linking facies variations, organic carbon richness and bulk bitumen content–A case study of the organic-rich Middle Triassic shales from eastern Svalbard. *Marine and Petroleum Geology*, 132: 105168, 2021.

Wesenlund, F., Grundvåg, S.-A., Engelschiøn, V.S., Thießen, O. and Pedersen, J.H.: Multi-elemental chemostratigraphy of Triassic mudstones in eastern Svalbard: Implications for source rock formation in front of the World's largest delta plain. *The Depositional Record,* 8(2): 718-753, https://doi.org/10.1002/dep2.182, 2022a.

Wesenlund, F., Grundvåg, S.A., Engelschiøn, V.S., Thießen, O. and Pedersen, J.H.: Multi-elemental chemostratigraphy of Triassic mudstones in eastern Svalbard: Implications for source rock formation in front of the World's largest delta plain. *The Depositional Record,* 8(2): 718-753, 2022b.

West, C.K., Greenwood, D.R. and Basinger, J.F.: Was the Arctic Eocene 'rainforest' monsoonal? Estimates of seasonal precipitation from early Eocene megafloras from Ellesmere Island, Nunavut. *Earth and Planetary Science Letters*, 427: 18-30, https://doi.org/10.1016/j.epsl.2015.06.036, 2015.

Westbury, M., Baleka, S., Barlow, A., Hartmann, S., Paijmans, J.L., Kramarz, A., Forasiepi, A.M., Bond, M., Gelfo, J.N. and Reguero, M.A.: A mitogenomic timetree for Darwin's enigmatic South American mammal Macrauchenia patachonica. *Nature Communications*, 8(1): 15951, 2017.

Westerhold, T., Marwan, N., Drury, A.J., Liebrand, D., Agnini, C., Anagnostou, E., Barnet, J.S., Bohaty, S.M., De Vleeschouwer, D. and Florindo, F.: An astronomically dated record of Earth's climate and its predictability over the last 66 million years. *Science*, 369(6509): 1383-1387, 10.1126/science.aba6853, 2020.

Wieczorek, R., Fantle, M.S., Kump, L.R. and Ravizza, G.: Geochemical evidence for volcanic activity prior to and enhanced terrestrial weathering during the Paleocene Eocene Thermal Maximum. *Geochimica et Cosmochimica Acta,* 119: 391-410, https://doi.org/10.1016/j.gca.2013.06.005, 2013.

Wierzbowski, H., Bajnai, D., Wacker, U., Rogov, M.A., Fiebig, J. and Tesakova, E.M.: Clumped isotope record of salinity variations in the Subboreal Province at the Middle–Late Jurassic transition. *Glob. Planet. Change,* 167, 172-189, 2018.

Wignall, P.B. and Twitchett, R.J.: Oceanic Anoxia and the End Permian Mass Extinction. Science, 272(5265): 1155-1158, doi:10.1126/science.272.5265.1155, 1996.

Wignall, P., Morante, R. and Newton, R.: The Permo-Triassic transition in Spitsbergen: $\delta^{13}C_{org}$ chemostratigraphy, Fe and S geochemistry, facies, fauna and trace fossils. *Geological Magazine,* 135(1): 47-62, https://doi.org/10.1017/S0016756897008121, 1998.

Wignall, P.B., Bond, D.P.G., Sun, Y., Grasby, S.E., Beauchamp, B., Joachimski, M.M. and Blomeier, D.P.G.: Ultra-shallow-marine anoxia in an Early Triassic shallow-marine clastic ramp (Spitsbergen) and the suppression of benthic radiation. *Geological Magazine,* 153(2): 316-331, 10.1017/s0016756815000588, 2016.



Wilkinson, C.M., Ganerød, M., Hendriks, B.W. and Eide, E.A.: Compilation and appraisal of geochronological data from the North Atlantic Igneous Province (NAIP). Geological Society, London, Special Publications, 447(1): 69-103 https://doi.org/10.1144/SP447.10, 2017.

Willard, D.A., Donders, T.H., Reichgelt, T., Greenwood, D.R., Sangiorgi, F., Peterse, F., Nierop, K.G., Frieling, J., Schouten, S. and Sluijs, A.: Arctic vegetation, temperature, and hydrology during Early Eocene transient global warming events. Global and Planetary Change, 178: 139-152, https://doi.org/10.1016/j.gloplacha.2019.04.012, 2019.

Wiman, C.: Ichthyosaurier aus der Trias Spitzbergens. *Bulletin of the Geological Institution of the University of Upsala*, *10*, 124-148, 1910.

Wiman, C.: Eine neue marine Reptilien-Ordnung aus der Trias Spitzbergens. *Bulletin of the Geological Institute of the University of Uppsala*, *22*, 183-196, 1928.

Wisshak, M., Volohonsky, E., and Blomeier, D.: Acanthodian fish trace fossils from the Early Devonian of Spitsbergen, Acta Palaeontol. Pol., 49, 2004a.

Wisshak, M., Volohonsky, E., Seilacher, A., and Freiwald, A.: A trace fossil assemblage from fluvial Old Red deposits (Wood Bay Formation; Lower to Middle Devonian) of NW-Spitsbergen, Svalbard, Lethaia, 37, 149–163, 2004b.

Woods, A. D.: Paleoceanographic and paleoclimatic context of Early Triassic time, C. R. Palevol, 4, 463–472, https://doi.org/10.1016/j.crpv.2005.07.003, 2005.

Worsley, D.: Sedimentological observations on the Grey Hoek Formation of northern Andrée Land, Spitsbergen, Nor. Polarinst. Årbok, 1970, 102–111, 1972.

Worsley, D., Agdestein, T., Gjelberg, J., Kirkemo, K., Mørk, A., Nilsson, I., Olaussen, S., Steel, R. J., and Stemmerik, L.: The geological evolution of Bjørnøya, Arctic Norway: implications for the Barents shelf, Nor. J. Geol., 81, 195–234, 2001.

Worsley, D.: The post-Caledonian development of Svalbard and the western Barents Sea, Polar Res., 27, 298–317, https://doi.org/10.1111/j.1751-8369.2008.00085.x, 2008.

Wu, Y., Chu, D., Tong, J., Song, H., Dal Corso, J., Wignall, P. B., Song, H., Du, Y., and Cui, Y.: Six-fold increase of atmospheric pCO2 during the Permian–Triassic mass extinction, Nat. Commun., 12, 2137, https://doi.org/10.1038/s41467-021-22298-7, 2021.

## Z

Zachos, J., Pagani, M., Sloan, L., Thomas, E. and Billups, K., 2001. Trends, Rhythms, and Aberrations in Global Climate 65 Ma to Present. *Science,* 292(5517): 686-693, doi:10.1126/science.1059412.



Zachos, J., Pagani, M., Sloan, L., Thomas, E., and Billups, K.: Trends, Rhythms, and Aberrations in Global Climate 65 Ma to Present, *Science,* 292, 686–693, https://doi.org/10.1126/science.1059412, 2001.

Zachos, J. C., Bohaty, S. M., John, C. M., McCarren, H., Kelly, D. C., and Nielsen, T.: The Palaeocene–Eocene carbon isotope excursion: constraints from individual shell planktonic foraminifer records, *Philos. Trans. R. Soc. A-Math. Phys. Eng. Sci.*, 365, 1829–1842, 2007.

Zachos, J. C., Dickens, G. R., and Zeebe, R. E.: An early Cenozoic perspective on greenhouse warming and carbon-cycle dynamics, *Nature,* 451, 279–283, https://doi.org/10.1038/nature06588, 2008.

Žák, K., Košťák, M., Man, O., Zakharov, V. A., Rogov, M. A., Pruner, P., Rohovec, J., Dzyuba, O. S., and Mazuch, M.: Comparison of carbonate C and O stable isotope records across the Jurassic/Cretaceous boundary in the Tethyan and Boreal Realms, *Palaeogeogr. Palaeoclimatol. Palaeoecol.*, 299, 83–96, https://doi.org/10.1016/j.palaeo.2010.10.038, 2011.

Zakharov, V. A., Rogov, M. A., Dzyuba, O. S., Žák, K., Košťák, M., Pruner, P., Skupien, P., Chadima, M., Mazuch, M., and Nikitenko, B. L.: Palaeoenvironments and palaeoceanography changes across the Jurassic/Cretaceous boundary in the Arctic realm: case study of the Nordvik section (north Siberia, Russia), *Polar Res.,* 33, 19714, https://doi.org/10.3402/polar.v33.19714, 2014.

Zanin, Y. N., Zamirailova, A. G., & Eder, V. G.: Chalcophile elements in black shales of the Bazhenov Formation, West Siberian sea basin. *Russian Geology and Geophysics*, **57**(4), 608–616. https://doi.org/10.1016/j.rgg.2015.03.018, 2016.

Zeebe, R. E., Ridgwell, A., and Zachos, J. C.: Anthropogenic carbon release rate unprecedented during the past 66 million years, *Nat. Geosci.*, 9, 325–329, http://dx.doi.org/10.1038/ngeo2681, 2016.

Zhang, H., Zhang, F., Chen, J.-B., Erwin, D. H., Syverson, D. D., Ni, P., Rampino, M., Chi, Z., Cai, Y.-F., Xiang, L., Li, W.-Q., Liu, S.-A., Wang, R.-C., Wang, X.-D., Feng, Z., Li, H.-M., Zhang, T., Cai, H.-M., Zheng, W., Cui, Y., Zhu, X.-K., Hou, Z.-Q., Wu, F.-Y., Xu, Y.-G., Planavsky, N., and Shen, S.-Z.: Felsic volcanism as a factor driving the end-Permian mass extinction, *Sci. Adv.,* 7, eabh1390, https://doi.org/10.1126/sciadv.abh1390, 2021a.

Zhang, L., Wang, C., Li, X., Cao, K., Song, Y., Hu, B., Lu, D., Wang, Q., Du, X., and Cao, S.: A new paleoclimate classification for deep time, *Palaeogeogr. Palaeoclimatol. Palaeoecol.,* 443, 98–106, https://doi.org/10.1016/j.palaeo.2015.11.041, 2016.

Zhang, Y., Ogg, J. G., Minguez, D., Hounslow, M. W., Olaussen, S., Gradstein, F. M., and Esmeray-Senlet, S.: Magnetostratigraphy of U-Pb–dated boreholes in Svalbard, Norway, implies that magnetochron M0r (a proposed Barremian-Aptian boundary marker) begins at 121.2 ± 0.4 Ma, *Geology*, 49, 733–737, https://doi.org/10.1130/g48591.1, 2021b.

Zhao, Y., Wei, W., Li, S., Yang, T., Zhang, R., Somerville, I., Santosh, M., Wei, H., Wu, J., Yang, J., Chen, W., and Tang, Z.: Rare earth element geochemistry of carbonates as a proxy for deep-time environmental



reconstruction, *Palaeogeogr. Palaeoclimatol. Palaeoecol.,* 574, 110443, https://doi.org/10.1016/j.palaeo.2021.110443, 2021.

Zuchuat, V., A Sedimentary Investigation of the Lower Triassic Formations and Their Underlying Permo-Carboniferous Units Across Spitsbergen, Svalbard. Institutt for geologi og bergteknikk, Norwegian University of Science and Technology. 165 pages, 2014.

Zuchuat, V., Sleveland, A. R. N., Twitchett, R. J., Svensen, H. H., Turner, H., Augland, L. E., Jones, M. T., Hammer, Ø., Hauksson, B. T., Haflidason, H., Midtkandal, I., and Planke, S.: A new high-resolution stratigraphic and palaeoenvironmental record spanning the End-Permian Mass Extinction and its aftermath in central Spitsbergen, Svalbard, *Palaeogeogr. Palaeoclimatol. Palaeoecol.*, 554, 109732, https://doi.org/10.1016/j.palaeo.2020.109732, 2020.

Økland, I. H., Delsett, L. L., Roberts, A. J., and Hurum, J. H.: A Phalarodon fraasi (Ichthyosauria: Mixosauridae) from the Middle Triassic of Svalbard, Norwegian Journal of Geology, 98, 267–288. https://doi.org/10.17850/njg98-2-06, 2018.
