# Peer review of "Phanerozoic paleoenvironmental and paleoclimatic evolution in Svalbard"

_EGUsphere, 2024_

## Referee Comment (RC1)

Smyrak-Sikora et al., Phanerozoic paleoenvironmental and paleoclimatic evolution in Svalbard

This is an admirably ambitious review of the literature re Svalbard's paleoenvironmental and paleoclimatic history. It will serve as a great resource for anyone interested in the long-term history and data available for Svalbard. It is not intended to presented new data with substantial new ideas, but to review the record of this region that-- like the Indian subcontinent-- has traversed across latitudes through the Phanerozoic.

It's nicely written, albeit can become a little dense at times— e.g. the Cenozoic is nearly all in one gargantuan paragraph. Breaking this up a bit might make for easier reading.
[It's possible that the Intro could be shortened a bit, and parts of it put into Geologic Setting or such?]
The figures are nicely done summaries of substantial data, although the font and symbol sizes are quite tiny in places, and thus difficult to decipher.

Other than that, I have only a few comments and suggestions, listed by line number below.

143— "twice to nearly four times AS FAST AS…"

144-149— the point about using Svalbard as an interesting case study, so to speak, for the phenomenon of polar amplification is interesting, but it has really only been situated in an arctic locale since the Cretaceous (from Fig. 3). So perhaps this statement should be qualified a bit.

146— a bit unclear (to me) what is meant by model "gradients"

506— Some might argue that it started in the latest Devonian (Fammenian)— with the late Devonian glaciation— well documented in South America, eastern North America, and other regions (albeit with minimal evidence for ice in the early Mississippian).

593— the rockS

597— check author spelling here (Beauchamp?)

689— space needed

809— consider replacing "modest" with "low-resolution"

826— space needed

1144— Consider qualifying the statement re "remarkable continuous stratigraphic succession" — yes, it is indeed a remarkable record with significant representation through much of the Phanerozoic, but it is not without unconformities, so is not, strictly speaking, continuous.

Figure 13 is a very interesting figure (reminds me of the path of India from polar latitudes to low latitudes— the opposite of Svalbard's path).

---

## Author Response (AR1)

Dear Editor Shiling Yang,

Thank you for your assistance with the revision process.

We are pleased to submit a revised version of our manuscript, in which we have addressed all comments and suggestions provided by the three reviewers. For clarity, our responses to Reviewer 1 are marked in blue, Reviewer 2 in green, and Reviewer 3 in orange. Individual response files to each reviewer are also provided.

In addition, the manuscript has undergone an internal review. As a result, several minor modifications and additional references have been included. These internal changes are marked in purple within the manuscript for transparency.

In response to Reviewer 3's comment, we have added a new plot (in Figure 4) presenting the accessible data for  $\delta^{13}C_{org}$  and total organic carbon (TOC). An interactive version of this plot, along with the corresponding dataset, is available on Zenodo.

Additional changes to the figures include:

Figure 4: plot with climate has been adjusted for Cambrian-Ordovician (internal review).

Figure 3: we have changed yellow to black circles to improve visibility of Svalbard's position (internal review).

Figure 6: Colours of the time scale were adjusted to match the style of other figures (internal review).

Figure 8 colours in the background of the formation have been remover (reviewer 2).

Figure 9: additional data have been plotted (reviewer3).

Figure 13 the plot with climate has been changed to match figure 4.

We hope the revised manuscript meets the journal's expectations and look forward to your feedback.

Sincerely,

Aleksandra Smyrak-Sikora

**Response to the comments by the Reviewer 1 Gerilyn (Lynn) Soreghan**

**Citation**: https://doi.org/10.5194/egusphere-2024-3912-RC1

**Dear Professor Gerilyn Soreghan**

Thank you very much for your constructive review. We will incorporate all suggestions, and we have provided responses to your comments, which are highlighted in blue.

This is an admirably ambitious review of the literature re Svalbard's paleoenvironmental and paleoclimatic history. It will serve as a great resource for anyone interested in the long-term history and data available for Svalbard. It is not intended to presented new data with substantial new ideas, but to review the record of this region that—like the Indian subcontinent—has traversed across latitudes through the Phanerozoic.

Thank you for this kind review.

It's nicely written, albeit can become a little dense at times—e.g. the Cenozoic is nearly all in one gargantuan paragraph. Breaking this up a bit might make for easier reading.

We have introduced a new paragraph break in the line 993.

[It's possible that the Intro could be shortened a bit, and parts of it put into Geologic Setting or such?]

Ok, the lines 129-138 of the introduction are moved to the geological settings (lines 181-190).

The figures are nicely done summaries of substantial data, although the font and symbol sizes are quite tiny in places, and thus difficult to decipher.

We have revised our figures and increase font and symbol sizes in Fig 9.

Figures 4, 10, 11 and 13 should be full-page figures, which will improve the readability.

Other than that, I have only a few comments and suggestions, listed by line number below.

143— "twice to nearly four times AS FAST AS..."

Ok, fixed.

144-149— the point about using Svalbard as an interesting case study, so to speak, for the phenomenon of polar amplification is interesting, but it has really only been situated in an arctic locale since the Cretaceous (from Fig. 3). So perhaps this statement should be qualified a bit.

We have included a sentence in the introduction (Lines 148-151): Since the Cretaceous, Svalbard has occupied an Arctic position, following its earlier location within more northerly boreal zones of the paleocontinents.

146— a bit unclear (to me) what is meant by model "gradients"

Rephrased to: *Polar amplification is evident in the geological past as extreme climates that are that are inconsistent with temperature distributions predicted by current models.*

506— Some might argue that it started in the latest Devonian (Fammenian)— with the late Devonian glaciation— well documented in South America, eastern North America, and other regions (albeit with minimal evidence for ice in the early Mississippian).

We have included it in lines 528-529 (after Rosa, and Isbell, 2021).

593— the rockS

Ok, fixed.

597— check author spelling here (Beauchamp?)

Ok, fixed.

689 - space needed

Ok, fixed.

809— consider replacing "modest" with "low-resolution"

Ok, fixed.

826— space needed

Ok, fixed.

1144— Consider qualifying the statement re "remarkable continuous stratigraphic succession"— yes, it is indeed a remarkable record with significant representation through much of the Phanerozoic, but it is not without unconformities, so is not, strictly speaking, continuous.

Ok, fixed.

Figure 13 is a very interesting figure (reminds me of the path of India from polar latitudes to low latitudes— the opposite of Svalbard's path).

Thank you.

With best regards,

Aleksandra Smyrak-Sikora

**Response to the comments by reviewer 2, Helmut Weissert**

Citation: https://doi.org/10.5194/egusphere-2024-3912-RC2

**Dear Prof. Helmut Weissert**

We would like to thank you and acknowledge your efforts in reviewing our manuscript. We incorporated all suggestions. Below, we provide responses to your comments, highlighted in green.

A review paper typically includes a substantial number of citations. Maintaining an overview of the vast number of publications concerning Phanerozoic climate in relation to Svalbard is definitely challenging. Including early literature on a topic is advisable as it provides a time dimension to our research and shows how ideas have evolved over time. Consequently, several recommendations for additional references will be added to reflect this context.

We are glad to include all suggested additional references.

Climate evolution - I am little surprised that the authors never mentioned/discussed the establishment of the Pangea-Megamonsoonal climate during the Permian and Triassic. Commenting on the establishment of a monsoonal climate during the Permian and Triassic and discussing any evidence (or mentioning "no-evidence") in the climate records that suggests a non-zonal climate is definitely beneficial. (see Parrish, 1991; Mutti and Weissert, 1995; or Preto et al., 2010, among others). The history of Earth provides excellent examples of "different climate worlds," which aid in the understanding of the coevolution of life and climate.

We have included information highlighting the global Mega-monsoonal climate during the Triassic. Including suggested references in the following sections:

4.3.1,

4.3.3,

4.3.4

6

6.2

However, the monsoonal climate has not been discussed in any of the previous work conducted in Svalbard. For this reason, we have not developed this aspect further. Since several global climate regimes are not represented in the Svalbard record—and we do not elaborate on these missing events—we have also kept the discussion of the monsoonal climate brief.

Evaluation of controversial data: with your rich expertise you are in a position to critically comment on controversial data, this can be of help for the reader. E.g. line 555-560: Deceasing or increasing O-18 values? Can you possibly comment on these controversial interpretations?

Explanation is added in lines 580-584.

**Comments**

Introduction

Citations: line 68 add early literature on this topic: e.g. Seibold, E., 1990 Engineering Geology (or others)

Seibold, E., 1990 is included in line 70

line 88 citations: e.g. Wignall 2001

Included

line 90 e.g. Weissert and Erba 2004

Fixed

line 91 Erba and Larson, 1999

Included Larson, R. L., & Erba, E. (1999)

line 95 OAE's, Schlanger and Jenkyns 1976

Included

line 195 Fra**sn**ian

Fixed

line 243 ...consists...

Fixed.

3.1.5. (Key stratigraphic sections) > you may shift climate description in this paragraph to discussion to 4.2.1.

Fixed. Climate descriptions are removed.

line 450 ....limestone/dolomite rocks

Fixed.

4.1. You show C-isotope data in Fig 7 – I did not find any discussion of these data in paragraph 4.1.

Additional information regarding the plot is included.

line 482 citations: Add e.g. Berner, R., 1993, 2005

Fixed.

Fig. 8 what are the meanings of the green-blue colors in the simplified lithological profile?

The colours have no meaning in context of this manuscript and are be removed.

line 555-560 decrease or increase in oxygen isotope values?

Explanation is added in lines 580-584

line 615 C-isotope anomaly and, also, oxygen isotope anomaly: e.g. Sun et al., 2012

Fixed.

line 617 see also Sanson-Barrera et al., 2016, for an east Greenland data set

Fixed

line 655 ....reflect...

Fixed

line 660 circulation pattern in a megamonsoonal world? see also Y. Hu et al 2022 ("emergence of the modern global monsoon...")

Fixed and also included further down in section 4.3.4

line 671 citation e.g. Sun et al., 2012

**Fixed**

line 680 you may also cite Leung, Zhang, Connell, 2022 for a stimulating discussion Reference included.

line 735 >> followed by prominent positive C-isotope excursion across Smithian-Spathian boundary (e.g. Galfetti et al., 2007, EPSL, negative spike followed by positive excursion).

Reference: Fixed

4.3.4 A possible additional reference: Preto et al., 2010

Fixed.

4.4.2. Time of collapsing Megamonsoonal climate (Sellwood and Valdes, 2010), time of ice buildup (Frakes and Francis, 1988)

Reference to: Sellwood and Valdes, 2008 (Jurassic climates. Proceedings of the Geologists Association, 119, 5-17) is included.

Frakes and Francis, 1988 manuscript is on Early Cretaceous cooling episodes and ice rafting, therefore this reference is included in the following section: 4.3.4. Jurassic-Cretaceous: A greenhouse with cold snaps.

line 826 ...the latter results....

Fixed

line 844 the record from the restricted Arabian carbonate platform is not very reliable if compared with eastern Tethyan pelagic records from the Hawasina Basin (Oman). There, the pattern of the J-C curve resembles the western Tethys pattern (see Celestino, Wohlwend et al, 2017).

Fixed, with reference to the western Tethys (Celestino et al., 2017). Carbon isotope stratigraphy, biostratigraphy and sedimentology of the Upper Jurassic–Lower Cretaceous Rayda Formation, Central Oman Mountains. *Newsletters on Stratigraphy*, 50(1), 91-109.

line 914. See early citation of Parana volcanism in Weissert et al., 1998

**Included**

line 1004 ...excellent....excellent.... (revise wording)

Fixed

line 1040 ....effects have been....

**Fixed**

line 2046 ....coupled...

Fixed

line 1095. ...models suggest...

Fixed.

line 1166 ...OAE1a.

Fixed

line 1196 >> add comment on Monsoonal climate

Fixed

6.1. Monsoonal climate (e.g. Parrish 1993; Mutti and Weissert, 1995, Preto et al., 2010 etc)
References are included in section 6.2.

With best regards,

Aleksandra Smyrak-Sikora

**Response to the review by anonymous reviewer 3**

Citation: https://doi.org/10.5194/egusphere-2024-3912-RC3

**Dear reviewer**

We thank you for your constructive response on the manuscript. We agree with your suggestions and will address these in a revised manuscript. Please see our responses in brown:

Lines 89-90: Saturation states of biologically important elements "in the ocean"

**Fixed.**

Lines 155-160 and figure 4: The authors describe five major climate types (A: Tropical, B: Dry, C: Temperate, D: Continental, and E: Polar) and state that these classifications are based on climatically sensitive deposits, paleontological evidence, and geochemical proxies. To enhance clarity, I recommend including a table summarizing the diagnostic criteria for each climate type. For example: Tropical (A): Presence of coal deposits, coral reefs, and warmwater fossils. This table would help readers better understand the basis for the climate classifications used in the study.

We have provided a clarification in the Supplementary material A1 outlining the main indicators used to create the plot.

Figure 4: I suggest that if it is possible the authors include direct data records (e.g., a long-term carbon isotope curve for Svalbard) in Figure 4, rather than simply showing data coverage. This would provide readers with a more direct and insightful understanding of the carbon cycle history in Svalbard.

The request for a Phanerozoic-scale plot with carbon isotope data presents challenges related to converting thickness into a timescale. Most of the data are collected along stratigraphic profiles in terms of thickness, and the time represented by the layers is highly uncertain. We have attempted this once before but were not entirely satisfied with the result; therefore, it was removed from the manuscript.

To address this request, the Carbon isotope values ( $\delta^{13}$  Corg) and total organic carbon (TOC) plotted in Figure 4. Stratigraphic age is based on a proxy conversion from sediment thickness to time. There is also an interactive plot and file with all plotted data as supplementary material A2.

Line 581, the reported  $\delta^{18}$ O values range from +2% to +30%, which appears unusually large. Could the authors verify these values? Please also specify whether these values are reported relative to VPDB or VSMOW.

This section is removed as it has considered the isotope composition of the cement with no meaning for deep-time climate.

Lines 659-670: In these lines, the authors discuss multiple geochemical datasets that indicate significant environmental changes during the End-Permian Mass Extinction (EPME). Are these data derived from the same drill core shown in Figure 9? If so, I recommend plotting these geochemical data in Figure 9 to visually illustrate the catastrophic environmental fluctuations during the EPME. This would make this data more accessible to readers.

We have expanded Fig. 9 with Fe/ K data from Zuchuat et al., 2020 that we refer to in the text.

in Figure 9, "VPBD" should be corrected to "VPDB," and some font sizes in the figure are too small to read comfortably. Please adjust the font sizes for better readability.

The readability of Fig 9 is improved, as well as the misspelling.

In Figure 12, "VPBD" should be corrected to "VPDB"

Fixed.

With kind regards,

Aleksandra Smyrak-Sikora